# CAUSAL GRAPH TRANSFORMER FOR TREATMENT EFFECT ESTIMATION UNDER UNKNOWN INTERFERENCE

**Anpeng Wu**[1,2]  **Haiyi Qiu**[1]  **Zhengming Chen**[2,3]  **Zijian Li**[2]  **Ruoxuan Xiong**[4]
**Fei Wu**[1*]  **Kun Zhang**[2,5*]

[1]Zhejiang University  [2]MBZUAI  [3]Guangdong University of Technology
[4]Emory University  [5]Carnegie Mellon University

{anpwu, haiyiqiu, wufei}@zju.edu.cn, {chenzhengming1103,leizigin}@gmail.com,
ruoxuan.xiong@emory.edu, kunz1@cmu.edu

## ABSTRACT

Networked interference, also known as the peer effect in social science and spillover effect in economics, has drawn increasing interest across various domains. This phenomenon arises when a unit's treatment and outcome are influenced by the actions of its peers, posing significant challenges to causal inference, particularly in treatment assignment and effect estimation in real applications, due to the violation of the SUTVA assumption. While extensive graph models have been developed to identify treatment effects, these models often rely on structural assumptions about networked interference, assuming it to be identical to the social network, which can lead to misspecification issues in real applications. To address these challenges, we propose an Interference-Agnostic **Cau**sal **Gra**ph Transfor**mer** (**CauGramer**), which aggregates peers information via $L$-order Graph Transformer and employs cross-attention to infer aggregation function for learning interference representations. By integrating confounder balancing and minimax moment constraints, CauGramer fully incorporates peer information, enabling robust treatment effect estimation. Extensive experiments on two widely-used benchmarks demonstrate the effectiveness and superiority of CauGramer. The code is available at `https://github.com/anpwu/CauGramer`.

## 1 INTRODUCTION

Causal inference has drawn increasing interest across networked scenarios, such as epidemiology (Barkley et al., 2020; Benjamin-Chung et al., 2018), marketing (Chae et al., 2017; Krishna & Rajan, 2009), economics (Mas & Moretti, 2009), and social science (Aronow & Samii, 2017; Cornelissen et al., 2017; Wang et al., 2018). However, one primary problem in causal inference on networked data stems from networked *interference*[1], in which each unit's treatment and outcome are influenced by the actions of its peers (interference nodes). This violation of the Stable Unit Treatment Value Assumption (SUTVA) poses significant challenges for treatment assignment and effect estimation (Cai et al., 2023; Jiang & Sun, 2022). For example, in COVID studies (Fisher, 2020; Latkin et al., 2021; Matrajt et al., 2021), individuals' decisions to get vaccinated (treatment) not only protect themselves from COVID but also reduce infection risk (outcome) within their social networks, thereby influencing the outcomes of others in the community. In this case, both individual and community immunity responses are influenced by the overall vaccine distribution, making individual-level information insufficient to identify causal effects. Similar to Jiang & Sun (2022) and Matrajt et al. (2021), given observational data, we aim to identify treatment effects and answer the counterfactual question: Would community immunity be stronger if treatments were assigned to a different group of people?

To estimate causal effects under networked interference, conventional network studying typically assumes that the structure of *interference graph*[2], is known and identical to the social network (Cai

---

*Corresponding Authors

[1]Interference refers to interactions between individuals where one person's actions can affect others.

[2]The interference graph represents interactions between individuals, with edges indicating these interactions. It may have edges not present in the social network, as interference can occur between non-connected individuals.

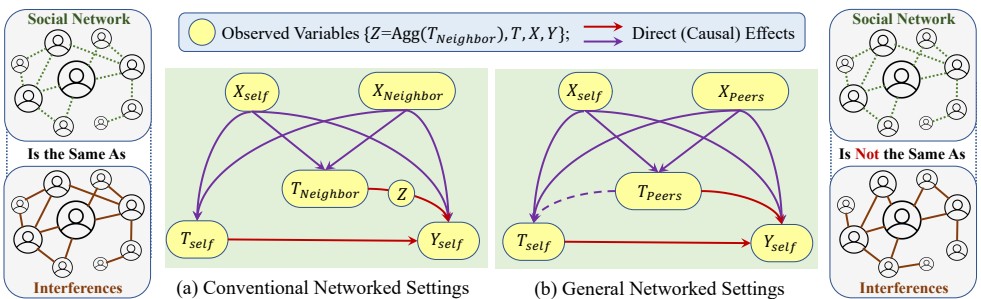

Figure 1: Causal Diagrams in Networked Data. (a) Conventional networked settings: interference graph equals social network with known peer-treatment effect on outcomes; (b) General networked settings: interference differs from social network with unknown peer-treatment effect.

et al., 2023; Forastiere et al., 2021; Qu et al., 2021; Liu et al., 2023; Ma et al., 2021; Ma & Tresp, 2021). As shown in Figure 1(a), given a known interference graph and assuming conditional independence between a unit's treatment and its neighbors' treatments, some semi-parametric estimators propose networked propensity score methods to infer causal effects (marked in red line) of both the unit's treatment and its neighbors' treatments (Chin, 2019; Ma & Tresp, 2021). Moreover, inspired by representation learning Shalit et al. (2017), recent works develop representation balancing methods and targeted maximum likelihood estimators for better effect estimation on networked data (Jiang & Sun, 2022; Cai et al., 2023; Chen et al., 2024). However, these works still follow the structural assumption that the interference graph is identical to the social network, which induces estimation bias in real-world networks, like how influenza viruses do not spread strictly along social networks.

Additionally, most existing works assume that the interference of neighbors' treatments on a unit's outcome is captured through a specified summary function, commonly defined as neighborhood exposure $Z = \text{Agg}(T_{Neighbor})$ (Forastiere et al., 2021; Qu et al., 2021; Jiang & Sun, 2022; Ma & Tresp, 2021; Cai et al., 2023; Chen et al., 2024).Typically, this aggregation is performed using a sum or average function, which mismatches the true data generation process (DGP) in real-world applications, leading to model misspecification and additional bias in causal effect estimation.

Therefore, in this paper, we study causal inference problems under an unknown interference graph and an unknown summary function, as illustrated in Figure 1(b). In such scenarios, a unit's treatment may be influenced by peers' treatments (marked in dashed line). To achieve robust causal estimation under the unknown interference setting, we develop an interference-agnostic **Cau**sal **Gra**ph Transfor**mer** (**CauGramer**) framework. First, although the interference graph is unknown, all interference nodes must exist within the $L$-order neighbor network ($L$ is sufficiently large). Thus, we can aggregate peer information (interference) using an $L$-layer Graph Transformer. Second, the complex interaction between two nodes is unknown. Unlike traditional methods that consider only self-attention, we model the interference function by treating self and peer features as linear-query and GCN-key pairs, designing a cross-attention mechanism to capture these interactions, which also broader the receptive field to learn complex sequential interference representations. Finally, by integrating confounder balancing and minimax moment constraints, our CauGramer fully incorporates peer information, enabling robust treatment effect estimation. Extensive experiments on two widely-used benchmarks demonstrate the superior performance of CauGramer. The contributions are summarized as follows:

- We formalize the treatment effects estimation problem under an unknown interference graph, which is not the same as the social network, and the structural function is unknown, noting that the interference graph may have edges not present in social network.

- To achieve robust causal inference under unknown interference, we propose an interference-agnostic CauGramer, which aggregates peer information through $L$-order networks and designs cross-attention to capture complex-sequential intervention functions. Moreover, balancing and bridge moment constraints are developed to ensure robust causal prediction.

- To illustrate the effectiveness and superiority of CauGramer, we conduct extensive experiments on two real-world networks with pseudo-real in scenarios involving unknown interference, latent variables, and limited budgets on treatments.

Table 1: Characteristics of Representative Algorithms on Observational Networked Data.

| Method | Settings | | Effects | | | Prior | | Sub-Modules | | |
|---|---|---|---|---|---|---|---|---|---|---|
| | Interference | Unmeasured | Main | Peer | Total | Graph | Summary | Reweighting | Representation | Attention |
| CNE (Veitch et al., 2019) | | P | ✓ | | | ✓ | | ✓ | | |
| NetDeconf (Guo et al., 2020) | | P | ✓ | | | ✓ | | | ✓ | |
| DRLearner (Leung & Loupos, 2022) | ✓ | | ✓ | ✓ | ✓ | ✓ | | ✓ | | |
| Net-TMLE (Ogburn et al., 2024) | ✓ | | ✓ | | | ✓ | ✓ | ✓ | | |
| GDML (Khatami et al., 2024) | ✓ | | ✓ | ✓ | ✓ | ✓ | ✓ | ✓ | | |
| G-HSIC (Ma & Tresp, 2021) | ✓ | | ✓ | | | ✓ | ✓ | | ✓ | |
| SPNet (Huang et al., 2023) | ✓ | P | ✓ | | | ✓ | | ✓ | ✓ | ✓ |
| RRNet (Cai et al., 2023) | ✓ | | ✓ | ✓ | ✓ | ✓ | ✓ | ✓ | ✓ | |
| NetEst (Jiang & Sun, 2022) | ✓ | | ✓ | ✓ | ✓ | ✓ | ✓ | | ✓ | |
| Uncertainty (Bhattacharya et al., 2020) | ✓ | | | | | | ✓ | | | |
| UNITE (Lin et al., 2024) | ✓ | | ✓ | | | | | | ✓ | |
| CauGramer (Ours) | ✓ | ✓ | ✓ | ✓ | ✓ | | | | ✓ | ✓ |

In observational networked data, we consider two settings with networked **Interference** or **Unmeasured** Confounders, three causal effects (**Main**, **Peer**, and **Total** effects), two forms of prior knowledge (interference **Graph** and **Summary** function), and three sub-modules (**Reweighting, Representation, Attention**), where ✓ indicates that the methods consider the corresponding term. For the unmeasured confounders setting, "P" denotes that CNE, NetDeconf, and SPNet only adjust partial unmeasured confounders embedded in the structural network and unit information.

## 2 RELATED WORK

In the causal inference on i.i.d. data, conventional methods typically utilized prognostic or propensity scores for matching, stratification, reweighting, and doubly robust techniques to control confounding effects (Bang & Robins, 2005; Hansen, 2008; Lee & Lee, 2022; Li et al., 2016; Rosenbaum, 1987; Rosenbaum & Rubin, 1983; Stuart et al., 2013; Yao et al., 2021a). With the advances in deep learning, CFRNet (Johansson et al., 2016; Shalit et al., 2017) first propose learning balanced representations by directly minimizing the distribution distance between treated and control groups using the Integral Probability Metric (IPM). Similar to Inverse Propensity Weighting (IPW) (Cai et al., 2023; Forastiere et al., 2021; Lee & Lee, 2022; Liu et al., 2023; 2016; Qu et al., 2021; Veitch et al., 2019; Zhang et al., 2023), representation with IPM has inspired numerous representation-based methods in both i.i.d. data (Hassanpour & Greiner, 2019; 2020; Wu et al., 2022; Yao et al., 2018; 2021b) and networked data (Cai et al., 2023; Guo et al., 2020; Zhao et al., 2024; Jiang & Sun, 2022; Ma & Tresp, 2021). Following this, we summarize representative algorithms on observational networked data in Table 1.

Under the assumption of no interference effects, CNE (Veitch et al., 2019) uses semi-supervised prediction to learn node embeddings for main effects estimation, and NetDeconf (Guo et al., 2020) leverages graph convolutional networks (GCNs) to captures the hidden confounders for main effect estimation. However, both approaches overlook interference and peer effects. To address this, DRLearner (Leung & Loupos, 2022) proposes a doubly robust estimator and explicitly models interference within a social network. Using targeted maximum likelihood estimation and double machine learning techniques, Net-TMLE (Ogburn et al., 2024) and GDML (Khatami et al., 2024) rely on pre-specified exposure mappings and assume interference is limited to direct neighbors. These strong assumptions often fail to hold in real-world applications. In addition, methods such as G-HSIC (Ma & Tresp, 2021), SPNet (Huang et al., 2023), RRNet (Cai et al., 2023), and NetEst (Jiang & Sun, 2022) explore causal effects under networked interference but either focus on main causal effects or require prior knowledge of a summary function. A common limitation across these approaches is the assumption that the interference graph is known and identical to the social network—a condition that is difficult to satisfy in practical scenarios.

Under unknown interference graph, Sävje et al. (2021) and Cortez et al. (2022) rely on randomized controlled trials (RCTs) to study causal effects. Hoshino & Yanagi (2023) assume that the interference graph is a subgraph of the social network and relies on the random assignment of instrumental variables (IVs) to ensure valid causal inference. In observational data, Bhattacharya et al. (2020) assumes that the true data-generating process perfectly corresponds (satisfying Markov and faithfulness conditions) to some unknown chain graph with two restrictions. However, in practical scenarios, structure learning algorithms often face implementation issues and produce unreliable results due to the size of the graph and unmet assumptions. Although Lin et al. (2024) proposes the UNITE framework, which uses L0-norm regularization and GCNs for accurate treatment effect estimation, it only supports the estimation of AME and IME. Besides, they do not consider that peer treatments can influence the treatment choices of others in real applications. **To address these issues and estimate causal effects under an unknown interference graph, we propose incorporating GCNs with cross-attention in the $L$-layer CauGramer embedding to aggregate $L$-order neighbor information.**

Notably, our CauGramer differs from Graphormer (Ying et al., 2021), CT (Melnychuk et al., 2022), CAL (Sui et al., 2022), and SPNet (Huang et al., 2023). Graphormer (Ying et al., 2021) just integrates the encoder of Transformer (Vaswani et al., 2017) into node, edge, and graph embedding learning for graph classification, without considering the Transformer decoder. CT (Melnychuk et al., 2022) is specifically designed for estimating counterfactual outcomes over time not for networked data. CAL (Sui et al., 2022) designs two soft mask graphs for node attention and edge attention and conducts implicit backdoor adjustment for graph classification, which suffers from networked interference and unmeasured confounders. For SPNet (Huang et al., 2023), it utilizes an adaptive learning weight function based on concatenation of neighbor features and self-features for reweighting, which fundamentally is not a Transformer framework.

## 3 TREATMENT EFFECT ESTIMATION UNDER UNKNOWN INTERFERENCE

### 3.1 PROBLEM SETUP AND NOTATIONS

We first describe our setting by taking the COVID vaccination as a case study (Fisher, 2020; Latkin et al., 2021; Matrajt et al., 2021). Let $X, T, Y$ denote a unit's features (e.g., physical fitness), treatment (e.g., receiving vaccination), and observed outcome (e.g., risk of infection), respectively. As shown in Figure 1(b), in the presence of networked interference, a unit's COVID infection risk is not only influenced by their own vaccination status but also by that of their peers. Therefore, when we study the causal effects of different vaccine assignments (i.e., treatment) on the community immunity level (i.e., outcome), we have to take the interference (i.e., peer effects) into consideration. However, in real applications, the interference graph typically is unknown and is not the same as the social network: COVID transmission is not strictly limited to social connections, as individuals without social ties can still interact and potentially spread the virus.

To model this problem, we consider the networked data as $\mathcal{D} = (\{\boldsymbol{x}_i, t_i, y_i\}_{i=1}^N, \boldsymbol{A}, \boldsymbol{E})$, where $\boldsymbol{A}$ and $\boldsymbol{E}$ represent the adjacency matrices of the observed social network and unknown interference graph, respectively, and $N$ denotes the total number of units. For each unit $i$, we have $d$-dimensional features $\boldsymbol{x}_i \in \mathcal{X}$ where $\mathcal{X} \subset \mathbb{R}^d$, binary treatment assignment $t_i \in \mathcal{T}$ where $\mathcal{T} = \{0, 1\}$, and the corresponding outcome $y_i \in \mathcal{Y}$ where $\mathcal{Y} \subset \mathbb{R}$. To distinguish between social networks and interference graphs, we denote the first-order 'neighbors' of $i$ in the social network $\boldsymbol{A}$ as observed $\mathcal{N}_i$ and the interacting nodes of $i$ in the interference graph $\boldsymbol{E}$ as unknown 'peers' $\mathcal{P}_i$. Then, we denote the treatment and feature vectors received by unit $i$'s peers as $\boldsymbol{t}_{\mathcal{P}_i}$ and $\boldsymbol{x}_{\mathcal{P}_i}$. Similarly, we define $\boldsymbol{t}_{\mathcal{N}_i}$ and $\boldsymbol{x}_{\mathcal{N}_i}$. Under networked interference, the potential outcome of a unit is influenced not only by its treatment but also by its peers' treatments, denoted by $y(\boldsymbol{x}_i, \boldsymbol{x}_{\mathcal{P}_i}, t, \boldsymbol{t}_{\mathcal{P}})$, where $\{t, \boldsymbol{t}_{\mathcal{P}}\}$ without subscript $i$ represents the manipulated interventions on the treatment. Then, the causal estimand of interest can be defined through three parameters: main effects (ME), peer effects (PE), and total effects (TE) (Cai et al., 2023; Jiang & Sun, 2022). We denote $\boldsymbol{t}_{\mathcal{P}} = \boldsymbol{0}_{\mathcal{P}}$ as the full-zero treatments.

**Definition 1** (Individual Main Effects (IME)). *IME denotes the effects of self-treatment, i.e.,* $\tau_{IME}(\boldsymbol{x}_i) = y(\boldsymbol{x}_i, \boldsymbol{x}_{\mathcal{P}_i}, 1, \boldsymbol{0}_{\mathcal{P}}) - y(\boldsymbol{x}_i, \boldsymbol{x}_{\mathcal{P}_i}, 0, \boldsymbol{0}_{\mathcal{P}}).$

**Definition 2** (Individual Peer Effects (IPE)). *IPE denotes the effects of peers' treatments, i.e.,* $\tau_{IPE}(\boldsymbol{x}_i, \boldsymbol{t}_{\mathcal{P}}) = y(\boldsymbol{x}_i, \boldsymbol{x}_{\mathcal{P}_i}, 0, \boldsymbol{t}_{\mathcal{P}}) - y(\boldsymbol{x}_i, \boldsymbol{x}_{\mathcal{P}_i}, 0, \boldsymbol{0}_{\mathcal{P}})$ *for any* $\boldsymbol{t}_{\mathcal{P}} \in \mathcal{T}^{|\mathcal{P}_i|}.$

**Definition 3** (Individual Total Effects (ITE)). *ITE denotes the combination of main and peer effects, i.e.,* $\tau_{ITE}(\boldsymbol{x}_i, \boldsymbol{t}_{\mathcal{P}}) = y(\boldsymbol{x}_i, \boldsymbol{x}_{\mathcal{P}_i}, 1, \boldsymbol{t}_{\mathcal{P}}) - y(\boldsymbol{x}_i, \boldsymbol{x}_{\mathcal{P}_i}, 0, \boldsymbol{0}_{\mathcal{P}})$ *for any* $\boldsymbol{t}_{\mathcal{P}} \in \mathcal{T}^{|\mathcal{P}_i|}.$

Similarly, average main effects (AME), average peer effects (APE), and average total effects (ATE) of specific treatment distribution $T \sim \pi_T$ can be defined, e.g., ATE $= \mathbb{E}_i^N[\tau_{\text{ITE}}(\boldsymbol{x}_i, \boldsymbol{t}_{\mathcal{P}})]$. Given the observational $\mathcal{D} = (\{\boldsymbol{x}_i, t_i, y_i\}_{i=1}^N, A)$ with unknown interference $\boldsymbol{E}$, we aim to identify three treatment effects and seek to answer the counterfactual question "would the community immunity level be stronger if we assigned treatments to a different group of people"?

### 3.2 CAUSAL IDENTIFICATION ON UNKNOWN INTERFERENCE GRAPH

To precisely estimate the three treatment effects, i.e., ME, PE, and TE, we first discuss causal identification under the standard causal assumptions on networked data (Jiang & Sun, 2022).

**Assumption 1** (Positivity). *The probability of a unit with their peers to receive any treatment pair* $(t, \boldsymbol{t}_{\mathcal{P}})$ *is always positive, i.e.,* $0 < \mathbb{P}(t_i = t, \boldsymbol{t}_{\mathcal{P}_i} = \boldsymbol{t}_{\mathcal{P}} \mid \boldsymbol{x}_i, \boldsymbol{x}_{\mathcal{P}_i}) < 1$ *for any* $\boldsymbol{x}_i.$

**Assumption 2** (Consistency). *The potential outcome is the same as the observed outcome under the same self-treatment and peer-treatments, i.e., $y_i = y(\boldsymbol{x}_i, \boldsymbol{x}_{\mathcal{P}_i}, t_i, \boldsymbol{t}_{\mathcal{P}_i})$ for treatment pair $(t_i, \boldsymbol{t}_{\mathcal{P}_i})$.*

**Assumption 3** (Unconfoundedness). *The self-treatment and peer-treatments are independent of the potential outcome given self and peer' features, i.e., $y(\boldsymbol{x}_i, \boldsymbol{x}_{\mathcal{P}_i}, t, \boldsymbol{t}_{\mathcal{P}}) \perp\!\!\!\perp (t, \boldsymbol{t}_{\mathcal{P}}) \mid (\boldsymbol{x}_i, \boldsymbol{x}_{\mathcal{P}_i})$.*

**Theorem 1** (Causal Identification). *Given these assumptions, while the Stable Unit Treatment Value Assumption (SUTVA) does not hold under networked interference, the treatment effects are identified as long as we can control the confounders $\boldsymbol{x}_i$ and the peers $\boldsymbol{x}_{\mathcal{P}_i}$.*

*Proof.* As shown in Figure 1, when we consider the (unknown) peer interference graph, all common causes $(\boldsymbol{x}_i, \boldsymbol{x}_{\mathcal{P}_i})$ of treatment pair $(t_i, \boldsymbol{t}_{\mathcal{P}_i})$ and outcomes $y_i$ have been discovered. Thus, we have:

$$
\begin{aligned}
\tau_{\text{TTE}}(\boldsymbol{x}_i, \boldsymbol{t}_{\mathcal{P}}) &= \mathbb{E}[y(\boldsymbol{x}_i, \boldsymbol{x}_{\mathcal{P}_i}, 1, \boldsymbol{t}_{\mathcal{P}}) \mid \boldsymbol{x}_i, \boldsymbol{x}_{\mathcal{P}_i}] - \mathbb{E}[y(\boldsymbol{x}_i, \boldsymbol{x}_{\mathcal{P}_i}, 0, \boldsymbol{0}_{\mathcal{P}}) \mid \boldsymbol{x}_i, \boldsymbol{x}_{\mathcal{P}_i}] \\
&= \mathbb{E}[y(\boldsymbol{x}_i, \boldsymbol{x}_{\mathcal{P}_i}, 1, \boldsymbol{t}_{\mathcal{P}}) \mid \boldsymbol{x}_i, \boldsymbol{x}_{\mathcal{P}_i}, t, \boldsymbol{t}_{\mathcal{P}}] - \mathbb{E}[y(\boldsymbol{x}_i, \boldsymbol{x}_{\mathcal{P}_i}, 0, \boldsymbol{0}_{\mathcal{P}}) \mid \boldsymbol{x}_i, \boldsymbol{x}_{\mathcal{P}_i}, 0, \boldsymbol{0}_{\mathcal{P}}] \quad (1) \\
&= \mathbb{E}[y_{1, \boldsymbol{t}_{\mathcal{P}}} \mid \boldsymbol{x}_i, \boldsymbol{x}_{\mathcal{P}_i}, t, \boldsymbol{t}_{\mathcal{P}}] - \mathbb{E}[y_{0, \boldsymbol{0}_{\mathcal{P}}} \mid \boldsymbol{x}_i, \boldsymbol{x}_{\mathcal{P}_i}, 0, \boldsymbol{0}_{\mathcal{P}}], \quad (2)
\end{aligned}
$$

where $y_{t, \boldsymbol{t}_{\mathcal{P}}}$ is the observed outcome when the unit and its peers have features $\boldsymbol{x}_i, \boldsymbol{x}_{\mathcal{P}_i}$ and receive the treatment pair $(t, \boldsymbol{t}_{\mathcal{P}})$. Eq. (1) holds under the Uncounfoundedness, i.e., $y(\boldsymbol{x}_i, \boldsymbol{x}_{\mathcal{P}_i}, t, \boldsymbol{t}_{\mathcal{P}}) \perp (t, \boldsymbol{t}_{\mathcal{P}}) \mid (\boldsymbol{x}_i, \boldsymbol{x}_{\mathcal{P}_i})$. Eq. (2) holds under the Consistency Assumption, i.e., $y_{t, \boldsymbol{t}_{\mathcal{P}}} = y(\boldsymbol{x}_i, \boldsymbol{x}_{\mathcal{P}_i}, t, \boldsymbol{t}_{\mathcal{P}})$. Theorem 1 holds for any treatment pair $(t, \boldsymbol{t}_{\mathcal{P}})$ under Positivity Assumption, i.e., $\mathbb{P}(t_i = t, \boldsymbol{t}_{\mathcal{P}_i} = \boldsymbol{t}_{\mathcal{P}}) > 0$. $\square$

However, since the interference graph is unknown, we cannot directly model the expectation function $\mathbb{E}[y_{t, \boldsymbol{t}_{\mathcal{P}}} \mid \boldsymbol{x}_i, \boldsymbol{x}_{\mathcal{P}_i}, t, \boldsymbol{t}_{\mathcal{P}}]$. Nevertheless, under the Unconfoundednes assumption, we know that $y(\boldsymbol{x}_i, \boldsymbol{x}_{\mathcal{P}_i}, t, \boldsymbol{t}_{\mathcal{P}}) \perp \{t_i, \boldsymbol{t}_{\mathcal{P}_i}\} \mid \{\boldsymbol{x}_i, \boldsymbol{x}_{\mathcal{P}_i}\}$. Therefore, if we control for all observed confounders $\{\boldsymbol{x}_i\}_{i=1}^n$, we can infer to $y(\boldsymbol{x}_i, \boldsymbol{x}_{\mathcal{P}_i}, t, \boldsymbol{t}_{\mathcal{P}}) \perp \{t_i, \boldsymbol{t}_{\mathcal{P}_i}\} \mid \{\boldsymbol{x}_i\}_{i=1}^n$. The interaction information in the interference graph is embedded within the node and network features. To capture this, we propose using two functions, $g_x$ and $g_t$, to represent peer feature and peer treatment information, respectively.

**Assumption 4** (Representation). *The treatment vector $\boldsymbol{t}_{\mathcal{P}_i}$ of peer nodes can be captured by a peer treatment function $g_t$, and the confounder vector $\boldsymbol{x}_{\mathcal{P}_i}$ by a peer confounder function $g_x$.*

**Proposition 1** (Interference). *If the unknown interference graph $\boldsymbol{E}$ is latent in full graph information $\{\boldsymbol{x}, \boldsymbol{t}, \boldsymbol{A}\}$, then, the outcome $\mathbb{E}[y_{t, \boldsymbol{t}_{\mathcal{P}}} \mid \boldsymbol{x}_i, \boldsymbol{x}_{\mathcal{P}_i}, t, \boldsymbol{t}_{\mathcal{P}}]$ is identified.*

*Proof.* Denote the other non-peer nodes of unit $i$ in the interference graph $\boldsymbol{E}$ as $\mathcal{O}_i$, i.e., all units in the community can be represented by $\{i, \mathcal{P}_i, \mathcal{O}_i\}$, i.e., $\boldsymbol{x} = \{\boldsymbol{x}_i, \boldsymbol{x}_{\mathcal{P}_i}, \boldsymbol{x}_{\mathcal{O}_i}\}$ for any $i$. Then, we have:

$$
\mathbb{E}[y_{t, \boldsymbol{t}_{\mathcal{P}}} \mid \boldsymbol{x}_i, \boldsymbol{x}_{\mathcal{P}_i}, t, \boldsymbol{t}_{\mathcal{P}}] \overset{(a)}{=} \mathbb{E}[y_{t, \boldsymbol{t}_{\mathcal{P}}} \mid \boldsymbol{x}_i, \boldsymbol{x}_{\mathcal{P}_i}, \boldsymbol{x}_{\mathcal{O}_i}, t, \boldsymbol{t}_{\mathcal{P}}, \boldsymbol{t}_{\mathcal{O}}] \overset{(b)}{=} \mathbb{E}[y_{t, \boldsymbol{t}_{\mathcal{P}}} \mid g_x(\boldsymbol{x}, \boldsymbol{A}), g_t(\boldsymbol{t}, \boldsymbol{A})], \quad (3)
$$

where $g_x(\boldsymbol{x}, \boldsymbol{A})$ a covariate embedding which captures peer features, and $g_t(\boldsymbol{t}, \boldsymbol{A})$ is a treatment embedding which captures peer treatments. Eq. (a) holds because $(\boldsymbol{x}_{\mathcal{O}_i}, \boldsymbol{t}_{\mathcal{O}_i})$ do not interfere with the outcomes of unit $i$. Eq. (b) holds when $g_x(\boldsymbol{x}, \boldsymbol{A})$ and $g_t(\boldsymbol{t}, \boldsymbol{A})$ can fully embed peer information. $\square$

In this paper, we do not directly recover the interference graph. Instead, we aim to learn two representation functions $g_x$ and $g_t$ to aggregate interference information. To avoid over-learning, we will use a balancing decomposing constraint to remove as much irrelevant information as possible from $g_x$ and $g_t$ that does not belong to peer interference nodes.

## 4 CAUGRAMER: CAUSAL GRAPH TRANSFORMER WITH CROSS-ATTENTION

Following the identification results from the previous sections, under unknown interference graph $\boldsymbol{E}$, we propose an interference-agnostic framework, called Causal Graph Transformer (CauGramer), to learn interference representations $\boldsymbol{R}_x = g_x(\boldsymbol{x}, \boldsymbol{A})$ and $\boldsymbol{R}_t = g_t(\boldsymbol{t}, \boldsymbol{A})$ from observed social network $\boldsymbol{A}$ and units features $(\boldsymbol{x}, \boldsymbol{t})$. Despite this advancement, there remain several questions and challenges in accurately estimating the main effects, peer effects, and total effects of treatment on outcomes:

- What are the characteristics of the interference representations $\boldsymbol{R}_x = g_x(\boldsymbol{x}, \boldsymbol{A})$ and $\boldsymbol{R}_t = g_t(\boldsymbol{t}, \boldsymbol{A})$, and how can we learn them?
- What is the difference between self-treatment and peer-treatment? How do we balance confounders for self-treatment and outcomes, as well as for peer-treatment and outcomes?
- How do we unbiasedly estimate the counterfactual regression outcomes?

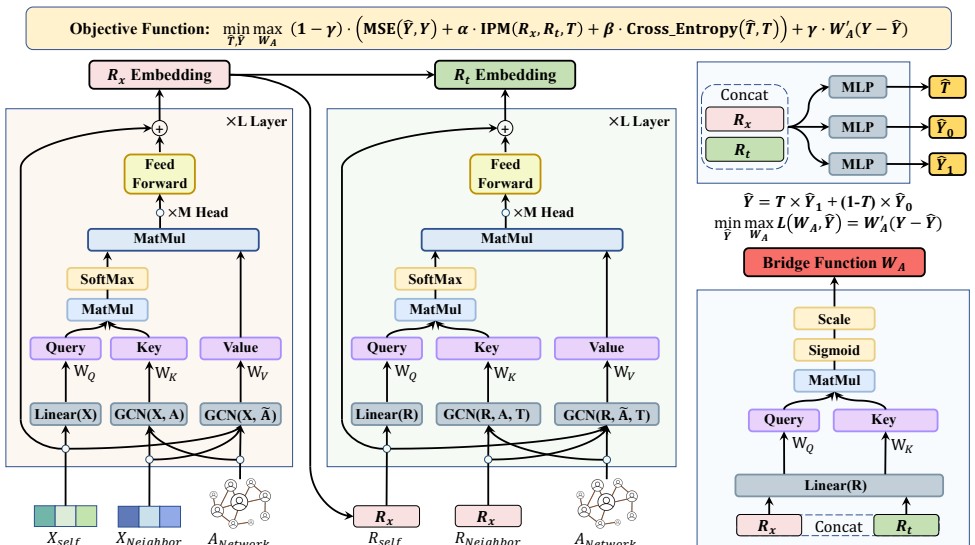

Figure 2: An illustration of the proposed CauGramer with cross-attention. Since the interference graph $E$ is unknown, we use $L$-layer CauGramer embedding to learn interference representations $R_x$ and $R_t$ from $L$-order neighbors in network $A$ with features $X$. Then we design a bridge function as minimax constraints to estimate potential outcomes. $\tilde{A}$ denotes $A$ with diagonal elements as 1.

These questions motivate the development of our CauGramer framework. Next, we will introduce CauGramer and provide insights into addressing these challenges. Notably, in this paper, we focus on estimating treatment effects through the two interference representations but do not reconstruct the interference graph, as identifying it is pretty hard and unnecessary for identifying treatment effects.

## 4.1 GCNs with Cross-Attention for Interference Representation Learning

**Motivation for Cross-Attention**: In networked data, traditional GCNs often struggle with limited receptive fields and information homogenization, making it challenging to capture long-distance interference. Consequently, conventional studies often only account for interference among first-order neighbors (Cai et al., 2023; Forastiere et al., 2021; Guo et al., 2020; Jiang & Sun, 2022; Qu et al., 2021). While Transformer's variants provide a broader perspective in graph learning (Rampášek et al., 2022; Sui et al., 2022; Ying et al., 2021), their use of GCN embeddings as input and attention modeling confined to self-embedding still limits the receptive field, hindering long-distance interference learning in the topology. To address this, we extend the Transformer to an $L$-order neighbor network and use cross-attention to broaden the receptive field for GCNs, enabling them to learn complex-sequential interference representations. In our CauGramer, as shown in Figure 2, we use GCNs with different adjacency matrices to capture topological information and construct local cross-attention for longer interaction sequences, allowing exploration of a larger receptive field and learning heterogeneous information.

Under networked interference, each unit's action and outcome are influenced not only by their own features but also by actions and features of (unknown) peers. For learning interference representation $R_x$, as shown in Figure 2, we model cross-attention (QKV pairs $(q_i, k_i, v_i)$) on self-features ($x_i$), neighbor-features ($x_{\mathcal{P}_i}$), and combined features ($x_i$, $x_{\mathcal{P}_i}$) to capture interactions between nodes within and beyond the local neighborhood. Let $r_x^{(1)} = x$ denote the input of the first layer, and $\tilde{A}$ represents the adjacency matrix $A$ with diagonal elements as 1. For each attention head in layer $h$,

$$r_x^{(h+1)} = \text{Attention}_x^{(h)} \cdot V_x^{(h)} = \text{Softmax}\left(\frac{Q_x^{(h)} \cdot K_x^{(h)\prime}}{\sqrt{d}}\right) \cdot V_x^{(h)}, \qquad (4)$$

$$Q_x^{(h)} = \text{Linear}_x^{(h)}(r_x^{(h)})W_Q^{(h)\prime}, \quad K_x^{(h)} = \text{GCN}_{xk}^{(h)}(r_x^{(h)}, A)W_K^{(h)\prime}, \quad V_x^{(h)} = \text{GCN}_{xv}^{(h)}(r_x^{(h)}, \tilde{A})W_V^{(h)\prime}. \quad (5)$$

In the $L$-layer CauGramer embedding with $M$-head attention, we use GCNs to aggregate $L$-order neighbor information and topology structure, while cross-attention provides broader receptive field that allows GCNs to embed complex-sequential interference representations, i.e., $R_x = g_x(x, A)$.

Then, we use the final output $\boldsymbol{r}_x^{(L)}$ as the input of the first layer of the peer-treatment representation network, i.e., $\boldsymbol{r}_t^{(1)} = \boldsymbol{r}_x^{(L)}$. As shown in Figure 2, we can similarly construct attention heads in layer $h$ of the treatment representation by considering neighbors' treatments into GCNs.

$$\boldsymbol{r}_t^{(h+1)} = \text{Attention}_t^{(h)} \cdot \text{V}_t^{(h)} = \text{Softmax}\left(\frac{\text{Q}_t^{(h)} \cdot \text{K}_t^{(h)\prime}}{\sqrt{d}}\right) \cdot \text{V}_t^{(h)}, \tag{6}$$

$$\text{Q}_t^{(h)} = \text{Linear}_t^{(h)}(\boldsymbol{r}_t^{(h)})W_Q^{(h)\prime}, \quad \text{K}_t^{(h)} = \text{GCN}_{tk}^{(h)}(\boldsymbol{r}_t^{(h)}, \boldsymbol{A}, \boldsymbol{t})W_K^{(h)\prime}, \quad \text{V}_t^{(h)} = \text{GCN}_{tv}^{(h)}(\boldsymbol{r}_t^{(h)}, \tilde{\boldsymbol{A}}, \boldsymbol{t})W_V^{(h)\prime}. \tag{7}$$

Then, we learn peer-treatment representations through $L$ layers with $M$-head cross-attentions, i.e., $\boldsymbol{R}_t = g_t(\boldsymbol{t}, \boldsymbol{A})$. As the network's depth $L$ increases, the scope of our GCNs with cross-attention expands, enabling the embedding of all potential interference peers into representations $\boldsymbol{R}_x$ and $\boldsymbol{R}_t$.

## 4.2 Representation Balancing for Adjusting Confounding Bias

**Motivation for Representation Balancing**: The interference representations $\boldsymbol{R}_x$ and $\boldsymbol{R}_t$, embedding the information of self-features $\boldsymbol{x}_i$, peer features $\boldsymbol{x}_{\mathcal{P}_i}$ and peer treatments $\boldsymbol{t}_{\mathcal{P}_i}$, are common cases of self-treatment and outcome of interest, which would confound the causal effect estimation. To estimate main effects and peer effects, inspired by Cai et al. (2023); Guo et al. (2020); Huang et al. (2023); Jiang & Sun (2022); Ma & Tresp (2021), we also learn balanced representations to adjust confounders by minimizing IPM of treated and control groups (Johansson et al., 2016; Shalit et al., 2017). Let $\boldsymbol{r} = \boldsymbol{r}_x \oplus \boldsymbol{r}_t$ be the concatenation of $\boldsymbol{r}_x$ and $\boldsymbol{r}_t$, we use Wasserstein (Arjovsky et al., 2017; Cuturi & Doucet, 2014) to measure the discrepancy of distributions:

$$\text{IPM}(\boldsymbol{r}, \boldsymbol{t}) = \text{Wass}(\{\boldsymbol{r}_{i:t_i=0}\}, \{\boldsymbol{r}_{j:t_j=1}\}), \tag{8}$$

where $\{\boldsymbol{r}_{i:t_i=t}\}$ refers to the concatenated representation distribution of $\boldsymbol{r} = \boldsymbol{r}_x \oplus \boldsymbol{r}_t$ on the samples with $t \in \{0, 1\}$. By minimizing $\text{IPM}(\boldsymbol{r}_x, \boldsymbol{r}_t, \boldsymbol{t})$, we can learn a balanced representation $\boldsymbol{r}_i$ and use it to identify the main effects for self-treatment $t_i$. *However, the relationship between peer-treatment $\boldsymbol{t}_{\mathcal{P}_i}$ and outcome $y_i$ remains confounded by the common causes $\boldsymbol{r}_i$, leading to biased estimates of peer and total effects. Moreover, the peer-treatment $\boldsymbol{t}_{\mathcal{P}_i}$ is unknown. Thus, we reformulate the balancing constraint $\boldsymbol{r}_i \perp\!\!\!\perp \boldsymbol{t}_{\mathcal{P}_i}$ to $\{\boldsymbol{r}_{j:j\in\mathcal{P}_i}\} \perp\!\!\!\perp t_i$. Recall the GCN structure, where the representation $\boldsymbol{r}_i$ serves as a weighted proxy of peer representations $\{\boldsymbol{r}_{j:j\in\mathcal{P}_i}\}$, hence, we maximize predictive power of $\{\boldsymbol{r}_{j:j\in\mathcal{P}_i}\}$ on self-treatment $t_i$, while minimizing $\text{IPM}(\boldsymbol{r}_x, \boldsymbol{r}_t, \boldsymbol{t})$ to ensure $\{\boldsymbol{r}_{j:j\in\mathcal{P}_i}\} \perp\!\!\!\perp t_i$:*

$$\text{Cross\_Entropy}(\hat{\boldsymbol{t}}, \boldsymbol{t}) = -\frac{1}{N}\sum_{i=1}^{N}\left[t_i \log(p_i) + (1-t_i)\log(1-p_i)\right], \tag{9}$$

where $p_i = \sigma(f_t(\{\boldsymbol{r}_{j:j\in\mathcal{P}_i}\}))$ is feed forward neural networks to estimate self-treatment $t_i$, and $\sigma$ is the Sigmoid function. With the balanced $\boldsymbol{r}_i$, the objective function of potential outcome estimation:

$$\mathcal{L}_y = \text{MSE}(\hat{\boldsymbol{y}}, \boldsymbol{y}) + \alpha \cdot \text{IPM}(\boldsymbol{r}, \boldsymbol{t}) + \beta \cdot \text{Cross\_Entropy}(\hat{\boldsymbol{t}}, \boldsymbol{t}), \tag{10}$$

where $\hat{y}_i = t_i \cdot f_y^1(\boldsymbol{r}_i) + (1-t_i) \cdot f_y^0(\boldsymbol{r}_i)$, $f_y^0(\boldsymbol{r}_i)$ and $f_y^1(\boldsymbol{r}_i)$ are feed-forward neural networks estimating potential control and treated outcomes. $\alpha$ and $\beta$ are hyper-parameters to trade off the representation balancing, treatment, and outcome regression, discussed in Appendix D and E.

We embed interference information into representations but do not need to identify or reconstruct it. For representation networks $\boldsymbol{r}(\boldsymbol{x}, \boldsymbol{t}, \boldsymbol{A})$ and outcome regression network $\hat{\boldsymbol{y}}(\boldsymbol{r}, \boldsymbol{t})$, we input the observed variables and use the latent representation $\boldsymbol{r}(\boldsymbol{x}, \boldsymbol{t}, \boldsymbol{A})$ to predict the outcome, i.e. $\mathbb{E}[\boldsymbol{y}|\boldsymbol{x}, \boldsymbol{t}, \boldsymbol{A}] = \hat{\boldsymbol{y}}(\boldsymbol{r}, \boldsymbol{t})$. By maximizing the predictive power of $\boldsymbol{r}$ on $\boldsymbol{y}$, (i.e., minimizing the estimation error ($\text{MSE}(\hat{\boldsymbol{y}}, \boldsymbol{y})$)), we can force the interference information is embed to $\boldsymbol{r}(\boldsymbol{x}, \boldsymbol{t}, \boldsymbol{A})$.

## 4.3 Bridge Moment Constraints for Potential Outcome Prediction

**Motivation for Bridge Moment Constraints**: There exists a trade-off between representation balancing and treatment regression. To avoid model misspecification and ensure unbiased estimation, we construct conditional moment constraints $\mathbb{E}[\boldsymbol{y} - \hat{\boldsymbol{y}} \mid \boldsymbol{x}, \boldsymbol{t}] = 0$, ensuring the residual approaches zero given $\boldsymbol{x}$ and $\boldsymbol{t}$. Furthermore, we reformulate it as an unconditional maximum moment problem:

$$\boldsymbol{y}^* = \arg\min_{\hat{\boldsymbol{y}}}\max_{q\in\mathbb{Q}}\mathbb{E}[(\boldsymbol{y} - \hat{\boldsymbol{y}})q(\boldsymbol{r}, \boldsymbol{t})], \quad q(\boldsymbol{r}, \boldsymbol{t}) = \text{Sigmoid}\left(\text{Q} \cdot \text{K}'\right), \tag{11}$$

where $\text{Q} = \text{Linear}(\boldsymbol{r}) \cdot W_Q'$, and $\text{K} = \text{Linear}(\boldsymbol{r}) \cdot W_K'$. Then, $q(\cdot)$ is termed as a "**bridge function**", a transformation links to eliminate confounding effects from the confounders in causal identification.

Table 2: Results of Constant Treatment Effects Estimation on BlogCatalog (BC) and Flickr Datasets.

| BC | Effects | CFRNet | DRLearner | NetDeconf | G-HSIC | SPNet | CAL | Graphormer | RRNet | NetEst | CauGramer |
|---|---|---|---|---|---|---|---|---|---|---|---|
| $\epsilon_{AVE}$ | AME | $\underline{0.058}_{\pm0.03}$ | $0.210_{\pm0.03}$ | $0.075_{\pm0.03}$ | $0.076_{\pm0.04}$ | $0.066_{\pm0.04}$ | $0.083_{\pm0.03}$ | $0.086_{\pm0.06}$ | $0.105_{\pm0.05}$ | $0.076_{\pm0.02}$ | $\mathbf{0.054}_{\pm\mathbf{0.03}}$ |
| | APE | $0.117_{\pm0.06}$ | $0.207_{\pm0.02}$ | $0.351_{\pm0.09}$ | $0.387_{\pm0.02}$ | $0.223_{\pm0.10}$ | $0.370_{\pm0.07}$ | $0.436_{\pm0.03}$ | $0.229_{\pm0.06}$ | $\underline{0.078}_{\pm0.02}$ | $\mathbf{0.034}_{\pm\mathbf{0.03}}$ |
| | ATE | $0.123_{\pm0.07}$ | $\underline{0.064}_{\pm0.05}$ | $0.337_{\pm0.09}$ | $0.351_{\pm0.06}$ | $0.203_{\pm0.10}$ | $0.355_{\pm0.08}$ | $0.349_{\pm0.07}$ | $0.296_{\pm0.06}$ | $0.065_{\pm0.03}$ | $\mathbf{0.039}_{\pm\mathbf{0.04}}$ |
| $\sqrt{\epsilon_{PEHE}}$ | IME | $\underline{0.096}_{\pm0.03}$ | $0.545_{\pm0.01}$ | $0.119_{\pm0.05}$ | $0.132_{\pm0.01}$ | $0.100_{\pm0.05}$ | $0.131_{\pm0.03}$ | $0.211_{\pm0.09}$ | $0.152_{\pm0.07}$ | $0.099_{\pm0.03}$ | $\mathbf{0.075}_{\pm\mathbf{0.07}}$ |
| | IPE | $0.122_{\pm0.06}$ | $0.220_{\pm0.02}$ | $0.365_{\pm0.09}$ | $0.410_{\pm0.03}$ | $0.230_{\pm0.10}$ | $0.384_{\pm0.07}$ | $0.457_{\pm0.03}$ | $0.238_{\pm0.06}$ | $\underline{0.092}_{\pm0.01}$ | $\mathbf{0.044}_{\pm\mathbf{0.02}}$ |
| | ITE | $0.147_{\pm0.07}$ | $0.514_{\pm0.01}$ | $0.351_{\pm0.09}$ | $0.384_{\pm0.06}$ | $0.213_{\pm0.09}$ | $0.370_{\pm0.08}$ | $0.429_{\pm0.04}$ | $0.311_{\pm0.06}$ | $\underline{0.117}_{\pm0.01}$ | $\mathbf{0.063}_{\pm\mathbf{0.04}}$ |
| **Flickr** | Effects | CFRNet | DRLearner | NetDeconf | G-HSIC | SPNet | CAL | Graphormer | RRNet | NetEst | CauGramer |
| $\epsilon_{AVE}$ | AME | $0.066_{\pm0.04}$ | $0.110_{\pm0.05}$ | $0.088_{\pm0.03}$ | $0.096_{\pm0.03}$ | $0.054_{\pm0.03}$ | $0.090_{\pm0.05}$ | $\underline{0.030}_{\pm0.03}$ | $0.160_{\pm0.03}$ | $0.063_{\pm0.04}$ | $\mathbf{0.028}_{\pm\mathbf{0.03}}$ |
| | APE | $0.115_{\pm0.04}$ | $0.211_{\pm0.03}$ | $0.345_{\pm0.06}$ | $0.354_{\pm0.04}$ | $0.121_{\pm0.05}$ | $0.302_{\pm0.04}$ | $0.409_{\pm0.03}$ | $0.268_{\pm0.09}$ | $\underline{0.055}_{\pm0.03}$ | $\mathbf{0.019}_{\pm\mathbf{0.01}}$ |
| | ATE | $0.144_{\pm0.06}$ | $0.117_{\pm0.07}$ | $0.351_{\pm0.05}$ | $0.300_{\pm0.09}$ | $0.131_{\pm0.05}$ | $0.310_{\pm0.06}$ | $0.382_{\pm0.04}$ | $0.374_{\pm0.11}$ | $\underline{0.073}_{\pm0.05}$ | $\mathbf{0.032}_{\pm\mathbf{0.02}}$ |
| $\sqrt{\epsilon_{PEHE}}$ | IME | $0.119_{\pm0.04}$ | $0.517_{\pm0.01}$ | $0.134_{\pm0.05}$ | $0.129_{\pm0.02}$ | $\underline{0.079}_{\pm0.05}$ | $0.137_{\pm0.07}$ | $0.086_{\pm0.03}$ | $0.229_{\pm0.04}$ | $0.095_{\pm0.05}$ | $\mathbf{0.047}_{\pm\mathbf{0.04}}$ |
| | IPE | $0.128_{\pm0.04}$ | $0.240_{\pm0.03}$ | $0.382_{\pm0.07}$ | $0.405_{\pm0.04}$ | $0.131_{\pm0.05}$ | $0.332_{\pm0.05}$ | $0.459_{\pm0.03}$ | $0.294_{\pm0.10}$ | $\underline{0.070}_{\pm0.03}$ | $\mathbf{0.032}_{\pm\mathbf{0.01}}$ |
| | ITE | $0.177_{\pm0.05}$ | $0.531_{\pm0.02}$ | $0.380_{\pm0.05}$ | $0.363_{\pm0.09}$ | $0.142_{\pm0.05}$ | $0.334_{\pm0.06}$ | $0.443_{\pm0.04}$ | $0.405_{\pm0.12}$ | $\underline{0.105}_{\pm0.05}$ | $\mathbf{0.051}_{\pm\mathbf{0.02}}$ |

Table 3: Results of Heterogeneous Treatment Effects Estimation with/without Unconfoundedness on BlogCatalog (BC) and Flickr Datasets. The best is **boldface** while the second best is underlined.

| BC | Effects | CFRNet | DRLearner | GDML | SPNet | NetEst | CEVAE | CNE | UNITE | C.G.(UC) | C.G.(NC) | CauGramer |
|---|---|---|---|---|---|---|---|---|---|---|---|---|
| $\epsilon_{AVE}$ | AME | $0.106_{\pm0.03}$ | $0.196_{\pm0.07}$ | $0.166_{\pm0.06}$ | $0.083_{\pm0.06}$ | $0.089_{\pm0.03}$ | $0.081_{\pm0.03}$ | $0.186_{\pm0.04}$ | $\mathbf{0.069}_{\pm\mathbf{0.00}}$ | $0.109_{\pm0.07}$ | $0.092_{\pm0.07}$ | $\underline{0.073}_{\pm0.06}$ |
| | APE | $0.092_{\pm0.05}$ | $0.183_{\pm0.03}$ | $0.371_{\pm0.06}$ | $0.212_{\pm0.11}$ | $0.077_{\pm0.02}$ | $0.403_{\pm0.01}$ | $0.519_{\pm0.05}$ | - | $0.067_{\pm0.04}$ | $\mathbf{0.055}_{\pm\mathbf{0.03}}$ | $\underline{0.057}_{\pm0.04}$ |
| | ATE | $0.116_{\pm0.07}$ | $0.099_{\pm0.02}$ | $0.537_{\pm0.09}$ | $0.243_{\pm0.11}$ | $0.109_{\pm0.03}$ | $0.349_{\pm0.04}$ | $1.127_{\pm0.06}$ | - | $0.077_{\pm0.05}$ | $\underline{0.047}_{\pm0.03}$ | $\mathbf{0.045}_{\pm\mathbf{0.04}}$ |
| $\sqrt{\epsilon_{PEHE}}$ | IME | $0.150_{\pm0.03}$ | $0.544_{\pm0.03}$ | $0.265_{\pm0.08}$ | $0.131_{\pm0.07}$ | $0.147_{\pm0.03}$ | $\underline{0.120}_{\pm0.03}$ | $0.288_{\pm0.05}$ | $0.194_{\pm0.00}$ | $0.159_{\pm0.10}$ | $0.139_{\pm0.09}$ | $\mathbf{0.090}_{\pm\mathbf{0.05}}$ |
| | IPE | $0.229_{\pm0.04}$ | $0.216_{\pm0.03}$ | $0.411_{\pm0.06}$ | $0.246_{\pm0.10}$ | $0.145_{\pm0.01}$ | $0.439_{\pm0.01}$ | $0.552_{\pm0.05}$ | - | $0.125_{\pm0.02}$ | $\mathbf{0.117}_{\pm\mathbf{0.02}}$ | $\underline{0.118}_{\pm0.02}$ |
| | ITE | $0.218_{\pm0.04}$ | $0.529_{\pm0.02}$ | $0.607_{\pm0.10}$ | $0.279_{\pm0.10}$ | $0.173_{\pm0.02}$ | $0.398_{\pm0.03}$ | $0.610_{\pm0.06}$ | - | $0.149_{\pm0.03}$ | $\underline{0.129}_{\pm0.02}$ | $\mathbf{0.125}_{\pm\mathbf{0.01}}$ |
| **Flickr** | Effects | CFRNet | DRLearner | GDML | SPNet | NetEst | CEVAE | CNE | UNITE | C.G.(UC) | C.G.(NC) | CauGramer |
| $\epsilon_{AVE}$ | AME | $0.091_{\pm0.05}$ | $0.135_{\pm0.08}$ | $0.239_{\pm0.08}$ | $0.096_{\pm0.06}$ | $0.069_{\pm0.05}$ | $0.063_{\pm0.04}$ | $0.168_{\pm0.05}$ | $\mathbf{0.043}_{\pm\mathbf{0.00}}$ | $0.091_{\pm0.07}$ | $0.073_{\pm0.05}$ | $\underline{0.058}_{\pm0.05}$ |
| | APE | $0.160_{\pm0.08}$ | $0.216_{\pm0.02}$ | $0.381_{\pm0.02}$ | $0.166_{\pm0.07}$ | $0.067_{\pm0.05}$ | $0.432_{\pm0.05}$ | $0.562_{\pm0.06}$ | - | $0.067_{\pm0.02}$ | $\underline{0.038}_{\pm0.02}$ | $\mathbf{0.030}_{\pm\mathbf{0.03}}$ |
| | ATE | $0.201_{\pm0.10}$ | $0.131_{\pm0.07}$ | $0.620_{\pm0.08}$ | $0.203_{\pm0.08}$ | $0.123_{\pm0.04}$ | $0.389_{\pm0.05}$ | $1.055_{\pm0.02}$ | - | $0.056_{\pm0.04}$ | $\underline{0.032}_{\pm0.03}$ | $\mathbf{0.025}_{\pm\mathbf{0.03}}$ |
| $\sqrt{\epsilon_{PEHE}}$ | IME | $0.174_{\pm0.06}$ | $0.529_{\pm0.02}$ | $0.359_{\pm0.10}$ | $0.147_{\pm0.07}$ | $0.136_{\pm0.06}$ | $0.139_{\pm0.04}$ | $0.261_{\pm0.06}$ | $0.212_{\pm0.00}$ | $0.141_{\pm0.08}$ | $\underline{0.107}_{\pm0.06}$ | $\mathbf{0.096}_{\pm\mathbf{0.06}}$ |
| | IPE | $0.207_{\pm0.07}$ | $0.264_{\pm0.02}$ | $0.448_{\pm0.02}$ | $0.213_{\pm0.06}$ | $0.156_{\pm0.03}$ | $0.505_{\pm0.05}$ | $0.637_{\pm0.06}$ | - | $0.128_{\pm0.01}$ | $\underline{0.112}_{\pm0.02}$ | $\mathbf{0.111}_{\pm\mathbf{0.01}}$ |
| | ITE | $0.274_{\pm0.08}$ | $0.547_{\pm0.02}$ | $0.716_{\pm0.09}$ | $0.252_{\pm0.07}$ | $0.185_{\pm0.05}$ | $0.478_{\pm0.04}$ | $0.669_{\pm0.07}$ | - | $0.150_{\pm0.02}$ | $\underline{0.127}_{\pm0.01}$ | $\mathbf{0.125}_{\pm\mathbf{0.01}}$ |

The detailed explanations of bridge functions are deferred to Appendix D.4. If the model is correctly specified and properly optimized, the residual $\epsilon = \boldsymbol{y} - \hat{\boldsymbol{y}}$ should be independent of covariates $\boldsymbol{x}$, i.e., $\epsilon \perp \{\boldsymbol{x}, \boldsymbol{t}\}$. We have $\mathbb{E}[q(\boldsymbol{x}, \boldsymbol{t}) \cdot \epsilon] = \mathbb{E}[q(\boldsymbol{x}, \boldsymbol{t})] \cdot \mathbb{E}[\epsilon] = 0$ for any bridge functions $q(\boldsymbol{x}, \boldsymbol{t})$. The minimax bridge moment constraint in Eq. (11) serves as a residual correction term, ensuring the residual $\mathbb{E}[\boldsymbol{y} - \hat{\boldsymbol{y}} \mid q(\boldsymbol{x}, \boldsymbol{t})] = 0$ for any $q(\boldsymbol{x}, \boldsymbol{t})$.

By incorporating this constraint, the objective function for potential outcome estimation becomes:

$$\arg\min_{\hat{\boldsymbol{t}}, \hat{\boldsymbol{y}}} \max_{q \in \mathbb{Q}} \mathcal{L} = (1 - \gamma)\left(\text{MSE}(\hat{\boldsymbol{y}}, \boldsymbol{y}) + \alpha \cdot \text{IPM}(\boldsymbol{r}, \boldsymbol{t}) + \beta \cdot \text{Cross\_Entropy}(\hat{\boldsymbol{t}}, \boldsymbol{t})\right) + \gamma \cdot \mathbb{E}[q(\boldsymbol{r}, \boldsymbol{t})(\boldsymbol{y} - \hat{\boldsymbol{y}})], \quad (12)$$

where $\gamma$ is a hyper-parameter serving as an interpolation coefficient. Under unconfoundedness assumption, when $\gamma = 0$, the model degrades to trade-off regression, and when $\gamma = 1$, it degrades to a Conditional Moment model. In the presence of unmeasured confounders, we will collect negative control variables and embed them into the Eqs. (11) and (12), generalizing CauGramer to CauGramer(NC) to address unmeasured confounding bias. The details are deferred to Appendix D.4.

The **implementation details**, **pseudo-code**, and **advances** of CauGramer are provided in Appendix D. Discussion and optimization of **Hyper-Parameters** $\{\alpha, \beta, \gamma\}$ can be found in Appendix E.

## 5 EMPIRICAL EXPERIMENTS

To evaluate the performance of our CauGramer on networked data, we conduct extensive experiments across eight dimensions: (1) Constant Treatment Effects (Table 2), (2) Heterogeneous Treatment Effects (Table 3), (3) Unknown Interference Graph (Figure 3), (4) Limited Budgets (Figure 4), and (5) Ablation Study (Table 4) in main text; (6) Hyper-Parameter Optimization (Figure 6 and Table 5), (7) Time Cost (Table 6), and (8) Various Interference Scenarios, including Interference Strength (Figure 7), Interference Functions (Table 7), and Unmeasured Confounding (Table 8). We include comparisons against more baselines in Tables 9&10 in Appendix E.

**Datasets.** In real-world applications, the ground-truth causal relations are unknown, and thus it is impossible to evaluate the causal estimation performance of CauGramer directly. Therefore, following existing works (Cai et al., 2023; Guo et al., 2021; 2020; Jiang & Sun, 2022; Ma et al.,

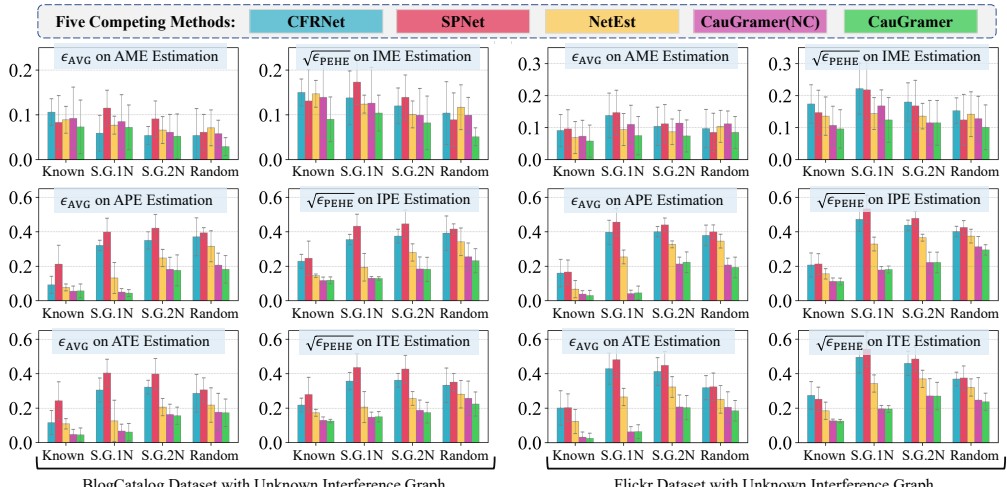

Figure 3: Results on General Networked Data with Unknown Interference Graph and Unknown Summary Functions: 'Known' graph is the same as social networks; 'S.G.1N' and 'S.G.2N' are sampled SubGraphs from 1st and 2nd-order neighbors graph; 'Random' denotes random graphs.

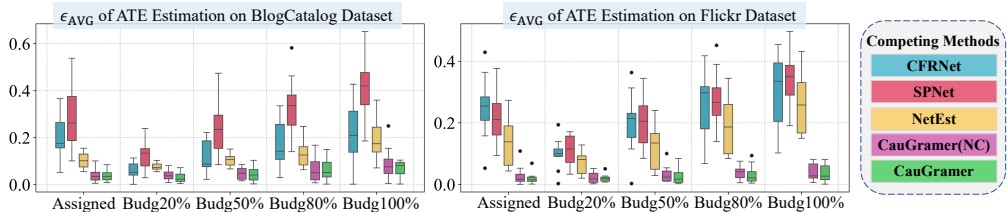

Figure 4: Results ($\epsilon_{\text{AVG}}$) of ATE Estimation on Limited Budget for Treatment Distribution: 'Assigned' denotes observed treatment, 'BudgX%' indicates varying budget allocations for treatment assignment.

2021; Veitch et al., 2019), we use pseudo-real datasets from BlogCatalog (BC) and Flickr, where the features ($\boldsymbol{x}$) and social networks ($\boldsymbol{A}$) are real, while treatments ($\boldsymbol{t}$), outcomes ($\boldsymbol{y}$), and interference ($\boldsymbol{E}$) are simulated. We use the same data simulation by Jiang & Sun (2022) to evaluate CauGramer's performance on original data with known intervention graphs and summary functions for constant treatment effects. Then, we design a series of generalized heterogeneous treatment effects datasets with unknown summary functions across various settings. Furthermore, we simulate unmeasured confounding by removing features. The generation details are deferred to Appendix E.

**Baselines.** We summarize the characteristics of all competing baselines in Table 1 in Section 2. In the traditional data (Jiang & Sun, 2022), we compare our methods CauGramer with causal methods (CFRNet (Shalit et al., 2017), DRLearner (Leung & Loupos, 2022), NetDeconf (Guo et al., 2020), G-HSIC (Ma & Tresp, 2021), SPNet (Huang et al., 2023), RRNet (Cai et al., 2023), and NetEst (Jiang & Sun, 2022)) and two Transformer methods (CAL (Sui et al., 2022) and Graphormer (Ying et al., 2021)). Then, we select top-4 baselines and add four models (GDML (Khatami et al., 2024), CEVAE (Louizos et al., 2017), CNE (Veitch et al., 2019) and UNITE (Lin et al., 2024)) in general network settings. For the sake of fairness, we modify all non-interference methods by incorporating neighbors' treatment and social networks as additional input. For unmeasured confounders, we provide two variants: C.G.(UC) randomly selects and removes two features from the input as unmeasured confounders and C.G.(NC) incorporates NCs as proxies for address unmeasured confounding effects.

**Evaluations.** In the experiments, we aim to estimate the ME, PE, and TE of treatments on individual and population level. In population, we use the Mean Absolute Error (MAE) on AME, APE, and ATE as metric, i.e., $\epsilon_{\text{AVG}} = \|\hat{\tau} - \tau\|$, where $\tau$ and $\hat{\tau}$ are the true and estimated average effects. In individual, we use rooted Precision in Estimation of Heterogeneous Effect (PEHE) on IME, IPE, and ITE, i.e., $\sqrt{\epsilon_{\text{PEHE}}} = \sqrt{\frac{1}{N}\sum_{i=1}^{N}(\hat{\tau}_i - \tau_i)^2}$. We conduct 10 replications to report (mean$_{\pm\text{std}}$) results.

**Results.** We compare our method with the baselines for estimating treatment effects with known interference graphs in Tables 2 and 3. First, the proposed CauGramer consistently outperforms all

Table 4: Ablation studies on CauGramer ($\alpha$, $\beta$, $\gamma$) by setting $\alpha = 0$ (no balancing constraints), $\beta = 0$ (no peer treatment constraints), and $\gamma = 0$ (no moment constraints). 'w/o CA' indicates no cross-attention. We report $\epsilon_{\text{AVE}}$ on AME, APE, ATE, and $\sqrt{\epsilon_{\text{PEHE}}}$ on IME, IPE, ITE.

| BlogCatalog(BC) | AME | APE | ATE | IME | IPE | ITE |
|---|---|---|---|---|---|---|
| CauGramer (0.0,0.0,0.0) | $0.093_{\pm 0.01}$ | $0.072_{\pm 0.02}$ | $0.089_{\pm 0.06}$ | $0.147_{\pm 0.05}$ | $0.139_{\pm 0.01}$ | $0.189_{\pm 0.04}$ |
| CauGramer (0.0,0.4,0.2) | $0.118_{\pm 0.10}$ | $\underline{0.061}_{\pm 0.03}$ | $0.116_{\pm 0.07}$ | $0.209_{\pm 0.13}$ | $0.133_{\pm 0.02}$ | $0.216_{\pm 0.06}$ |
| CauGramer (0.8,0.0,0.2) | $0.085_{\pm 0.08}$ | $\underline{0.061}_{\pm 0.04}$ | $\underline{0.066}_{\pm 0.06}$ | $0.126_{\pm 0.11}$ | $\underline{0.120}_{\pm 0.02}$ | $\underline{0.132}_{\pm 0.03}$ |
| CauGramer (0.8,0.4,0.0) | $\underline{0.080}_{\pm 0.09}$ | $0.064_{\pm 0.04}$ | $0.069_{\pm 0.05}$ | $\underline{0.123}_{\pm 0.12}$ | $0.121_{\pm 0.02}$ | $0.142_{\pm 0.03}$ |
| CauGramer (0.8,0.4,0.2) w/o CA | $0.104_{\pm 0.06}$ | $0.067_{\pm 0.02}$ | $0.072_{\pm 0.05}$ | $0.154_{\pm 0.08}$ | $0.125_{\pm 0.01}$ | $0.142_{\pm 0.03}$ |
| CauGramer (0.8,0.4,0.2) | $\mathbf{0.073}_{\pm \mathbf{0.06}}$ | $\mathbf{0.057}_{\pm \mathbf{0.04}}$ | $\mathbf{0.045}_{\pm \mathbf{0.04}}$ | $\mathbf{0.090}_{\pm \mathbf{0.05}}$ | $\mathbf{0.118}_{\pm \mathbf{0.02}}$ | $\mathbf{0.125}_{\pm \mathbf{0.01}}$ |

| Flickr | AME | APE | ATE | IME | IPE | ITE |
|---|---|---|---|---|---|---|
| CauGramer (0.0,0.0,0.0) | $0.080_{\pm 0.07}$ | $0.050_{\pm 0.05}$ | $0.131_{\pm 0.06}$ | $0.127_{\pm 0.09}$ | $0.130_{\pm 0.02}$ | $0.209_{\pm 0.04}$ |
| CauGramer (0.0,0.4,0.2) | $0.087_{\pm 0.07}$ | $0.213_{\pm 0.15}$ | $0.309_{\pm 0.03}$ | $0.146_{\pm 0.08}$ | $0.303_{\pm 0.13}$ | $0.384_{\pm 0.16}$ |
| CauGramer (0.8,0.0,0.2) | $\underline{0.066}_{\pm 0.03}$ | $\mathbf{0.028}_{\pm \mathbf{0.01}}$ | $\underline{0.028}_{\pm 0.03}$ | $\underline{0.118}_{\pm 0.04}$ | $\underline{0.112}_{\pm 0.01}$ | $\underline{0.128}_{\pm 0.01}$ |
| CauGramer (0.8,0.4,0.0) | $0.078_{\pm 0.03}$ | $0.041_{\pm 0.02}$ | $0.051_{\pm 0.03}$ | $0.149_{\pm 0.04}$ | $0.141_{\pm 0.01}$ | $0.160_{\pm 0.01}$ |
| CauGramer (0.8,0.4,0.2) w/o CA | $0.100_{\pm 0.11}$ | $0.135_{\pm 0.16}$ | $0.174_{\pm 0.19}$ | $0.190_{\pm 0.13}$ | $0.221_{\pm 0.13}$ | $0.265_{\pm 0.15}$ |
| CauGramer (0.8,0.4,0.2) | $\mathbf{0.058}_{\pm \mathbf{0.05}}$ | $\underline{0.030}_{\pm 0.03}$ | $\mathbf{0.025}_{\pm \mathbf{0.03}}$ | $\mathbf{0.096}_{\pm \mathbf{0.06}}$ | $\mathbf{0.111}_{\pm \mathbf{0.01}}$ | $\mathbf{0.125}_{\pm \mathbf{0.01}}$ |

baselines, because our Causal Graph Transformer with Minimax Moment Constraints improves the computation graph with Cross-Attention and provides double guarantees to address confounding bias. Second, when NCs are available and latent variables exit, while C.G.(UC) suffers from unmeasured confounding, C.G.(NC) significantly improves performance, even approaching that of CauGramer. In this case, the moment constraint in C.G.(NC) acts as a bridge function. By comparing our C.G.(NC) and CauGramer method to the optimal baseline in Tables 2 and 3, when the interference graph is known, we observe a decrease in $\epsilon_{\text{AVG}}$ and $\epsilon_{\text{PEHE}}$ of $\{39\%, 56\%\}$ and $\{46\%, 51\%\}$ on constant total effect estimation for BlogCatalog and Flickr datasets, a decrease in $\epsilon_{\text{AVG}}$ and $\epsilon_{\text{PEHE}}$ of $\{55\%, 80\%\}$ and $\{28\%, 32\%\}$ on heterogeneous total effect estimation, highlighting the superiority of our method.

**Abalation Studies.** As shown in Table 4, we denote different ablation versions as CauGramer($\alpha$, $\beta$, $\gamma$), where hyperparameters $\alpha$ controls balancing constraints, $\beta$ handles peer treatments constraints and $\gamma$ manages moment constraints. 'w/o CA' indicates CauGramer without the cross-attention module. The results in Table 4 demonstrate that each component contributes effectively to causal effect estimation. In Appendix E.2, we further discuss the hyperparameter optimization for CauGramer.

**Scaling to Unknown Interference Graph and Limited Budgets.** Figure 3 shows the performance of our methods on unknown interference across BlogCatalog and Flickr datasets. Across 1st and 2nd-order SubGraphs ('S.G.1N' and 'S.G.2N') and 'Random' Graphs, CauGramers consistently outperform all baselines. Especially on 'S.G.1N', our method shows a TE estimation improvement of over 35%. In Figure 4, across treatment assignment budgets from 20% to 100%, we observe robust performance with ATE estimation errors below 0.1 on both datasets. Conversely, other methods display notably increasing estimation errors as the budget changes.

**Time Complexity Analysis.** CauGramer suffers from the same limitation as other graph transformers, namely the quadratic complexity of the attention computation. However, unlike Graphormer and its variants, which rely on computationally expensive full graph attention mechanisms through self-attention, CauGramer computes cross-attention only from its 1-hop neighbors' embeddings. Its time cost decreases as node degree decreases. Empirically, we report the average training times of various methods in a single execution on both datasets in Table 6 in Appendix E.3. CauGramer significantly reduces single execution training time compared to most baselines, keeping it under 180 seconds.

## 6 CONCLUSION

In networked interference settings, many graph models developed to identify treatment effects rely on the structural assumption of a known interference graph and aggregation function, which may not hold true in real-world applications and can result in model misspecification. To address this issue, we study causal inference problems under unknown interference and propose a **CauGramer**, which learns interference representations and infers causal effects without relying on structural assumptions. By integrating confounder balancing and minimax moment constraints, CauGramer fully incorporates peer information, enabling robust treatment effect estimation.

## ACKNOWLEDGEMENTS

This work was supported by the National Natural Science Foundation of China (62441605, 62441617, 62376243, 62037001), and the Starry Night Science Fund at Shanghai Institute for Advanced Study (Zhejiang University).

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

## A  BROADER IMPACTS

This paper proposes a novel **Cau**sal **Gra**ph Transfor**mer** algorithm, named **CauGramer**, which extends the computation graph of Transformer to networked data and uses GCN with cross-attention to learn complex intervention representations. By fully embedding peer information, the balancing constraints and minimax moment constraints provide double guarantees for causal prediction. This advancement in Transformer improves the interpretability and generalization performance of the model across various networked scenarios, such as epidemiology, marketing, and social science. For instance, in COVID vaccination, when vaccine resources are limited, CauGramer can identify the causal effects of different treatment distributions on groups and then allocate vaccinations precisely, maximizing community immunity with limited resources. Additionally, it supports evaluating the impact of community skills training on the labor market and estimating the effects of public policies on social groups, such as housing distributions on community stability and economic development.

However, in real social problems, treatment distributions that maximize treatment effects on the population might not be the optimal choice for certain individuals. When individual treatment effects and community treatment effects conflict, we should make decisions based on the specific situation rather than relying solely on model results. Overall, CauGramer advances the Transformer framework for networked data, providing new lines for further research on causal Transformers in the machine learning field. We will also explore its application to more ML tasks, using it to accelerate large models with Transformers and eliminate large model illusions through causal prediction.

## B  LIMITATIONS AND FUTURE DIRECTIONS

**Unknown Social Network:** CauGramer relies on a known social network for information aggregation. In practice, however, such networks may not always be accessible due to privacy concerns or application-specific constraints. Nevertheless, as demonstrated in Figure 3, CauGramer remains robust even when the provided network is entirely unrelated to the 'Random' interference graph. It still learns causal effects effectively and outperforms existing methods, suggesting that any connected graph—even one that is randomly generated and independent of the interference graph—can help aggregate interference information. In future work, we aim to address this limitation by developing strategies to construct efficient connected graphs directly from the data. This would further reduce dependence on a predefined social network and enhance the method's applicability to scenarios without network availability.

**Negative Controls Availability**: In the presence of latent confounding variables, although the proposed CauGramer can readily generalize to address unmeasured confounding by constructing bridge functions of negative controls, identifying negative controls remains a challenging task, requiring expert knowledge and extensive experimental data support. Fortunately, with increasing attention from practitioners to this technique, negative controls are becoming more accessible through web technologies, user profiles, and digitized records (Hu et al., 2023). To further study unmeasured confounding, we plan to explore techniques like instrumental variable (IV) regression, variational autoencoders (VAEs), and sensitivity analysis in future work.

## C  THE DIFFERENCE OF OUR ALGORITHMS WITH TRADITIONAL METHODS

Our CauGramer is different with the following algorithms:

1. Leung & Loupos (2022): This paper introduces a non-parametric approach for estimating causal effects, specifically treatment and spillover effects, using observational data from a known social network. It proposes a doubly robust learner to model causal effect where interference decreases with network distance, accounting for peer influences on both outcomes and treatment selection. Unlike prior studies that rely on low-dimensional representations of confounding, the authors introduce a high-dimensional network unconfoundedness condition, utilizing graph neural networks (GNNs) to estimate these confounders effectively.

2. Ogburn et al. (2024): This paper introduces a semiparametric approach for handling dependencies caused by information transmission and latent similarities between nodes. It assumes a known interference graph with interference limited to direct neighbors and relies on prior knowledge of two structural functions, $s_C$ and $s_X$ , to summarize covariate and

treatment interferences. Given these strong assumptions, Net-TMLE leverages structural equation modeling and targeted maximum likelihood estimation to estimate causal effects, supporting inference for static, dynamic, and stochastic interventions in social networks.

3. Khatami et al. (2024): Assuming the interference graph and exposure mapping are known, this paper proposes constructing a focal set, where all units' neighborhoods are non-overlapping, to ensure independence among analyzed units and enable consistent estimation of causal effects in social networks. By combining graph machine learning approaches with the Double Machine Learning (GDML) framework, this method represents a significant advancement in causal inference. It integrates Graph Neural Networks (GNNs) to estimate propensity scores and outcome models while addressing challenges like interference, high-dimensional confounding, and model misspecification.

4. Sävje et al. (2021): This paper study average treatment effects estimation in randomized experiments with unknown and arbitrary interference among units. The authors propose the Expected Average Treatment Effect (EATE) as a generalized estimand that incorporates potential spillover effects by marginalizing all possible treatment assignments. While their study provides a theoretical and experimental analysis using existing methods, it does not introduce new algorithms to address unknown interference. Furthermore, their analysis is based on randomized controlled trials (RCTs). In contrast, our paper focuses on addressing the challenge of unknown interference in observational data.

5. Cortez et al. (2022): Under unknown network structures, Cortez et al. (2022) extend the work of Yu et al. (2022) by generalizing causal inference methods from linear to polynomial models, enabling the estimation of total treatment effects (TTE) under more complex and non-linear network interactions. They introduce a staggered rollout experimental design, which incrementally applies treatments and collects observations, eliminating the need for prior knowledge of the network structure. However, a limitation of their approach is its reliance on multiple randomized trials (i.e., the staggered rollout design) to gather observations at different stages.

6. Hoshino & Yanagi (2023): This paper studies causal inference models in which individuals interact within a social network and may not comply with the assigned treatments. It potentially assumes that the interference arises through the social network, where the structure of interference may depend on local or approximate neighborhood interactions. Based on this, the paper introduces Instrumental Exposure Mapping (IEM), a low-dimensional representation of potentially complex spillover effects, to estimate Intention-to-Treat (ITT) and Local Average Treatment Effects (LATE) for compliers, accounting for both direct and indirect effects while explicitly handling the challenges of noncompliance and unknown interference. However, this paper assumes that the interference graph is a subgraph of the social network and relies on the random assignment of instrumental variables (IVs), as well as the independence of IVs across individuals, to ensure valid causal inference.

7. Bhattacharya et al. (2020): Under interference and network uncertainty, this paper proposes methods that integrate structure learning and causal inference techniques to estimate causal effects. Their algorithms rely on structure learning to first learn graphical models and then estimate the Population Average Overall Effect (PAOE), a variant of the Average Total Effect. They address network uncertainty under partial interference through structure learning and then use the auto-g-computation algorithm to estimate the PAOE. This method rely on the assumption that the true data-generating process corresponds perfectly (satisfying Markov and faithfulness conditions) to some unknown chain graph, with two restrictions. This paper does not release its code.

8. Lin et al. (2024): Under unknown interference, the UNITE algorithm uses a Graph Structure Learner (GSL) to infer the hidden interference structure by constructing a complete graph and imposing L0-norm regularization to identify significant connections. With the inferred structure, Graph Convolutional Networks (GCNs) are employed to learn an aggregation function and model the interference from neighboring units, effectively capturing the interference patterns and enabling accurate treatment effect estimation.

DRLearner (Leung & Loupos, 2022) proposes a doubly robust estimator and explicitly models interference within a social network. Using targeted maximum likelihood estimation and double machine learning techniques, Net-TMLE (Ogburn et al., 2024) and GDML (Khatami et al., 2024) rely

on pre-specified exposure mappings and assume interference is limited to direct neighbors. Under the unknown interference graph, Sävje et al. (2021) and Cortez et al. (2022) rely on randomized controlled trials (RCTs) to study causal effects. Hoshino & Yanagi (2023) assume that the interference graph is a subgraph of the social network and relies on the random assignment of instrumental variables (IVs) to ensure valid causal inference. In observational data, Bhattacharya et al. (2020) assumes that the true data-generating process perfectly corresponds (satisfying Markov and faithfulness conditions) to some unknown chain graph with two restrictions. However, in practical scenarios, structure learning algorithms often face implementation issues and produce unreliable results due to the size of the graph and unmet assumptions. Although Lin et al. (2024) proposes the UNITE framework, which uses L0-norm regularization and GCNs for accurate treatment effect estimation, it only supports the estimation of AME and IME. Besides, they do not consider that peer treatments can influence the treatment choices of others in real applications. To address these issues and estimate causal effects under an unknown interference graph, we propose incorporating GCNs with cross-attention in the $L$-layer CauGramer embedding to aggregate $L$-order neighbor information.

## D  Implementation Details, Pseudo-Code, and Advances

In this paper, inspired by the powerful Transformer (Vaswani et al., 2017), we propose an interference-agnostic **Cau**sal **Gra**ph Transfor**mer** (**CauGramer**) framework, which extends the computation graph of the Transformer to an $L$-order neighbor network and uses cross-attention to provide a broader receptive field for GCNs (Kipf & Welling, 2016), enabling them to learn complex-sequential interference representations. Under the generalized networked setting with Unconfoundedness (Figure 5(a)), by fully embedding peer information, the proposed balancing constraints and minimax moment constraints can provide double guarantees for causal prediction to estimate the main, peer, and total effects. However, when the Unconfoundedness Assumption is violated, the outcome regression (Eq. (1)) would be confounded by unmeasured confounders (denoted by ($U$)). To generalize CauGramer for settings with unmeasured confounders (UCs), we propose incorporating negative control variables (NCs) into the unconditional maximum moment optimization and integrating this into the objective function (Eq. (12)). This adaptation directly extends CauGramer to CauGramer(NC) for such settings. Appendix D.1 describes the key definitions and assumptions of CauGramer; Appendix D.2 details its implementation; Appendix D.4 introduces the negative controls and bridge functions for CauGramer(NC); and we discuss CauGramer's advances and limitations in Appendices D.5 and B.

### D.1  Key Definitions and Assumptions of CauGramer

In this subsection, we will clarify the definitions of confounders and potential outcomes, as well as the required assumptions in CauGramer. We start by reviewing the causal diagrams of the proposed generalized networked setting. As shown in Figure 1, in social networked data, both the self-treatment and peer-treatment of a unit influence the self-outcome. We define the effect of self-treatment on self-outcome as main effect, denoted as $\tau_M(\boldsymbol{x}_i, \boldsymbol{x}_{\mathcal{P}_i}) : \{\boldsymbol{x}_i, \boldsymbol{x}_{\mathcal{P}_i}\} \to y_i$, which is a function of self-features and peer-features. Next, we define the peer-treatment effect of peer $j$ on unit $i$'s outcome as the heterogeneous peer effect, denoted as $\tau_P(\boldsymbol{x}_i, \boldsymbol{x}_j) : \{\boldsymbol{x}_i, \boldsymbol{x}_j\} \to y_i$ where $j \in \mathcal{P}_i$, which is a function of the features of unit $i$ and peer $j$. Finally, we further define the potential control outcome under $(t_i, \boldsymbol{t}_{\mathcal{P}_i}) = (0, \boldsymbol{0}_{\mathcal{P}_i})$ as $\mu_0(\boldsymbol{x}_i, \boldsymbol{x}_j) : \{\boldsymbol{x}_i, \boldsymbol{x}_{\mathcal{P}_i}\} \to y_i$. The formal definitions are as follows:

**Definition** (Potential Control Outcomes). $y(\boldsymbol{x}_i, \boldsymbol{x}_{\mathcal{P}_i}, 0, \boldsymbol{0}_{\mathcal{P}}) = \mu_0(\boldsymbol{x}_i, \boldsymbol{x}_j)$.

**Definition** (Potential Treated Outcomes). $y(\boldsymbol{x}_i, \boldsymbol{x}_{\mathcal{P}_i}, t_i, \boldsymbol{t}_{\mathcal{P}_i}) = \mu_0(\boldsymbol{x}_i, \boldsymbol{x}_j) + t_i \cdot \tau_M(\boldsymbol{x}_i, \boldsymbol{x}_{\mathcal{P}_i}) + \sum_{j \in \mathcal{P}_i} [t_j \cdot \tau_P(\boldsymbol{x}_i, \boldsymbol{x}_j)]$.

**Definition** (Observed Outcomes). $y_i = t_i \cdot y(\boldsymbol{x}_i, \boldsymbol{x}_{\mathcal{P}_i}, t_i, \boldsymbol{t}_{\mathcal{P}_i}) + (1 - t_i) \cdot y(\boldsymbol{x}_i, \boldsymbol{x}_{\mathcal{P}_i}, 0, \boldsymbol{0}_{\mathcal{P}})$.

In networked interference, the challenge is that we can only observe the outcome for the treatment assignment that was actually applied, and not for all potential treatment assignments simultaneously. This means we cannot directly observe what would have happened under different treatment distribution. Besides, in observational data, as shown in Figure 1(b), some variables are common causes of both the treatment and the outcome of interest. These variables are confounders—extraneous variables whose presence affects the relationship between the variables being studied and introduces confounding bias (Pearl, 2009) when imbalanced confounders exist.

**Definition** (Confounders to Main Effect). Confounders to Main Effect are variables that influence both self-treatment and self-outcome, potentially creating a spurious association that can distort the main effect estimation. In this paper, Confounders of $t_i$ and $y_i$ are $\{\boldsymbol{x}_i, \boldsymbol{x}_{\mathcal{P}_i}, \boldsymbol{t}_{\mathcal{P}_i}\}$.

**Definition** (Confounders to Peer Effect). Confounders to Peer Effect are variables that influence both peer-treatment and self-outcome, potentially creating a spurious association that can distort the peer effect estimation. In this paper, Confounders of $\boldsymbol{t}_{\mathcal{P}_i}$ and $y_i$ are $\{\boldsymbol{x}_i, \boldsymbol{x}_{\mathcal{P}_i}, \boldsymbol{t}_i\}$.

To eliminate confounding bias for estimating main effects and peer effects, inspired by Cai et al. (2023); Guo et al. (2020); Huang et al. (2023); Jiang & Sun (2022); Ma & Tresp (2021), we propose representation balancing to adjust confounders by minimizing IPM (Eq. (10)) of treated and control groups (Johansson et al., 2016; Shalit et al., 2017):

$$\mathcal{L}_y = \text{MSE}(\hat{\boldsymbol{y}}, \boldsymbol{y}) + \alpha \cdot \text{IPM}(\boldsymbol{r}, \boldsymbol{t}) + \beta \cdot \text{Cross\_Entropy}(\hat{\boldsymbol{t}}, \boldsymbol{t}),$$

where $\alpha$ and $\beta$ are hyper-parameters to trade off the representation balancing and outcome regression.

To precisely estimate treatment effects, i.e., ME, PE, and TE, we first make the standard causal assumptions on networked data (Forastiere et al., 2021; Jiang & Sun, 2022).

**Assumption** (Positivity). The probability of a unit with their peers to receive any treatment pair $(t, \boldsymbol{t}_{\mathcal{P}})$ is always positive, i.e., $0 < \mathbb{P}(t_i = t, \boldsymbol{t}_{\mathcal{P}_i} = \boldsymbol{t}_{\mathcal{P}} \mid \boldsymbol{x}_i, \boldsymbol{x}_{\mathcal{P}_i}) < 1$ for any $\boldsymbol{x}_i$.

**Assumption** (Consistency). The potential outcome is the same as the observed outcome under the same self-treatment and peer-treatments, i.e., $y_i = y(\boldsymbol{x}_i, \boldsymbol{x}_{\mathcal{P}_i}, t_i, \boldsymbol{t}_{\mathcal{P}_i})$ for treatment pair $(t_i, \boldsymbol{t}_{\mathcal{P}_i})$.

**Assumption** (Unconfoundedness). The self-treatment and peer-treatments are independent of the potential outcome given self and peer' features, i.e., $y(\boldsymbol{x}_i, \boldsymbol{x}_{\mathcal{P}_i}, t, \boldsymbol{t}_{\mathcal{P}}) \perp\!\!\!\perp (t, \boldsymbol{t}_{\mathcal{P}}) \mid (\boldsymbol{x}_i, \boldsymbol{x}_{\mathcal{P}_i})$.

**Assumption** (Representation). The treatment vector $\boldsymbol{t}_{\mathcal{P}_i}$ of peer nodes can be captured by a peer treatment function $g_t$, and the confounder vector $\boldsymbol{x}_{\mathcal{P}_i}$ by a peer confounder function $g_x$.

As shown in Figure 1, under unconfoundedness assumption, all confounders have been observed. When we control $\{\boldsymbol{x}_i, \boldsymbol{x}_{\mathcal{P}_i}, \boldsymbol{t}_{\mathcal{P}_i}\}$, the main effect of $t_i$ on $y_i$ is identified; when we control $\{\boldsymbol{x}_i, \boldsymbol{x}_{\mathcal{P}_i}, \boldsymbol{t}_i\}$, the peer effect of $\boldsymbol{t}_{\mathcal{P}_i}$ on $y_i$ is identified.

Positivity assumption requires that the probability of receiving any treatment among a unit and their peers is always positive, i.e., there are no circumstances under which a treatment cannot be applied. It's fundamental to causal inference because if there are conditions where a treatment is completely unavailable, it becomes impossible to draw any inference about its effects from the data. Consistency assumption states that under the same self-treatment and peer-treatments, the observed outcome is consistent with the potential outcome. This assumption is also crucial in causal inference as it provides the basis for inferring causal effects from observational data.

### D.2 Implementation Details and Pseudo-Code

As shown in Figure 2, the overall architecture of our model consists of the following components:

- The interference representation $\boldsymbol{R}_x = g_x(\boldsymbol{x}, \boldsymbol{A})$ of self-features and peer-features. In this paper, we propose a $L$-layers (default: 2) $M$-heads (default: 3) cross-attention GCN to learn the representation $\boldsymbol{R}_x = g_x(\boldsymbol{x}, \boldsymbol{A})$. In each attention head, all neural networks consist of one layer comprising 32 hidden units. We then perform cross-attention computation, which yields the concatenation of M head embeddings, followed by a feed-forward network to output a 32-dimensional representation.

- The interference representation $\boldsymbol{R}_t = g_t(\boldsymbol{t}, \boldsymbol{A})$ of self-treatments and peer-treatments. In this paper, we propose a $L$-layers (default: $L = 1$ for main results and $L = 2$ for unknown interference graph in Figure 3) $M$-heads (default: 3) cross-attention GCN to learn the representation $\boldsymbol{R}_t = g_t(\boldsymbol{t}, \boldsymbol{A})$. The process for learning the treatment interference representation $\boldsymbol{R}_t$ is similar to that of the feature interference representation $\boldsymbol{R}_x$, except that the GCN employs treatments $\boldsymbol{t}$ to mask node information in each attention. The feed-forward network then outputs a 32-dimensional representation.

- The treatments and outcomes regression networks $\{f_t, f_y^0, f_y^1\}$. First, we concatenate the representations $\boldsymbol{R} = \boldsymbol{R}_x \oplus \boldsymbol{R}_t$. Then, we use three two-layer linear networks, where each layer comprises 64 hidden units, to regress treatments $\boldsymbol{T}$ and potential outcomes $\{\boldsymbol{Y}_0, \boldsymbol{Y}_1\}$.

---

**Algorithm 1** CauGramer: Causal Graph Transformer

---

**Input:** The observed features $x_i$, treatment $t_i$, and outcomes $y_i$; Social network $A$; Hyper-parameters $\{\alpha, \beta, \gamma\}$; The num of training epochs I = 300; Steps $\{K, S\} = \{150, 2\}$.
**Output:** Representations $\{g_x, g_t\}$, one bridge function $q$, one treatment estimators $f_t$, and two potential outcome estimators $\{f_y^0, f_y^1\}$.
**Loss Function:** $\mathcal{L} = (1 - \gamma)\left(\text{MSE}(\hat{y}, y) + \alpha \cdot \text{IPM}(r, t) + \beta \cdot \text{Cross\_Entropy}(\hat{t}, t)\right) + \gamma \cdot \mathbb{E}[W_A(y - \hat{y})]$.

---

Initialize networks $\{g_x, g_t, q, f_t, f_y^0, f_y^1\}$.
**for** $i = 1, 2, \cdots, $ I **do**
 $r_x = g_x(x, A), r_t = g_t(t, A)$;
 $r = r_x \oplus r_t$;
 $W_A = q(r, t), \hat{t} = \sigma(f_t(\{r_{j:j \in \mathcal{P}_i}\})), \hat{y}_0 = f_y^0(r)$ and $\hat{y}_1 = f_y^1(r)$;
 $\hat{y} = t \cdot \hat{y}_1 + (1 - t) \cdot \hat{y}_0$;
 $\text{Cross\_Entropy}(\hat{t}, t) = -\frac{1}{N}\sum_{i=1}^{N}\left[t_i \log\left(\hat{t}_i\right) + (1 - t_i)\log\left(1 - \hat{t}_i\right)\right]$;
 $\text{IPM}(r, t) = \text{Wass}(\{r_{i:t_i=0}\}, \{r_{j:t_j=1}\}), \quad \text{MSE}(\hat{y}, y) = \frac{1}{N}\sum_{i=1}^{N}(\hat{y}_i - y_i)^2$;
 **if** $i < $ K or $i \% $ S $== 0$ **then**
  Update $\{g_x, g_t, f_t, f_y^0, f_y^1\} \leftarrow \text{Adam}(\mathcal{L})$;
 **else**
  Update $\{q\} \leftarrow \text{Adam}(-\frac{1}{N}\sum_{i=1}^{N}(q(r_i, t_i) \cdot (\hat{y}_i - y_i)))$;
 **end if**
**end for**
**return** Networks $\{g_x, g_t, q, f_t, f_y^0, f_y^1\}$.

---

- The bridge function $W_A = q(r, t)$. We input the concatenated representation $R$ into a single-layer linear network to obtain Query and Key separately. Then, they are fed into MatMul operations to obtain the attention-based bridge function.

In the whole architecture of CauGramer, we use the ReLU activation function and set the dropout rate to 0.1 to mitigate overfitting. Then, the objective function is:

$$\arg\min_{\hat{t}, \hat{y}} \max_{W_A} \mathcal{L} = (1 - \gamma)\left(\text{MSE}(\hat{y}, y) + \alpha \cdot \text{IPM}(r, t) + \beta \cdot \text{Cross\_Entropy}(\hat{t}, t)\right) + \gamma \cdot \mathbb{E}[W_A(y - \hat{y})]$$

where $W_A = q(r, t), \hat{t} = \sigma(f_t(\{r_{j:j \in \mathcal{P}_i}\})), \hat{y}_i = t_i \cdot f_y^1(r_i) + (1 - t_i) \cdot f_y^0(r_i)$. Then, we adopt Adam optimization with a learning rate of 0.01 and set epochs to 300 to alternately train $W_A$ and $\{\hat{T}, \hat{Y}_0, \hat{Y}_1\}$. During the optimization of $W_A$, we keep the parameters in the representation networks $\{g_x(\cdot), g_t(\cdot)\}$ and treatment and outcome regression networks $\{f_t(\cdot), f_y^0(\cdot), f_y^1(\cdot)\}$ fixed. Conversely, we keep the parameters in $q(\cdot)$ fixed while optimizing the representation networks and treatment and outcome regression networks. The pseudo-code is placed in Algorithm 1.

Hardware used: (1) MacBook Pro with Apple M2 Pro. (2) Ubuntu 16.04.3 LTS operating system with 2 * Intel Xeon E5-2660 v3 @ 2.60GHz CPU (40 CPU cores, 10 cores per physical CPU, 2 threads per core), 256 GB of RAM, and 4 * GeForce GTX TITAN X GPU with 12GB of VRAM.

Software used: Python 3.9 with numpy 1.26.4, scipy 1.13.0, pandas 2.2.2, torch 2.3.0, scikit-learn 1.4.2, openpyxl 3.1.2, torch_geometric 2.5.2, torch-scatter 1.1.0.

## D.3 BRIDGE FUNCTION: ENFORCING RESIDUAL INDEPENDENCE

The core intuition behind the bridge function is that it enforces independence between the residual $y - \hat{y}$ and the variables $\{x, t\}$:

1. Independence $(y - \hat{y}(x, t)) \perp \{x, t\}$ is equivalent to $(y - \hat{y}(x, t)) \perp q(x, t)$ for all $q(\cdot)$.

2. This equivalence implies that $Pr(y - \hat{y}(x, t), q(x, t)) = Pr(y - \hat{y}(x, t)) \times Pr(q(x, t))$ for all $q(\cdot)$.

3. As a result, we have: $E[(y - \hat{y}(x, t)) \cdot q(x, t)] = E[y - \hat{y}(x, t)] \cdot E[q(x, t)] = 0$ for all $q(\cdot)$.

4. This leads to the minimax formulation: $\min_{\hat{y}} \max_{q \in Q} E[(y - \hat{y}) \cdot q(x, t)]$, where:

- $\max_{q \in Q}$ detects any dependency between the residual and the covariates.
- $\min_{\hat{y}}$ adjusts the predictions $\hat{y}$ to eliminate this dependency.

"The model being correctly specified and properly optimized" is a sufficient condition for "the residual is independent of covariates $x$". Conversely, if the residual is not independent of $\{x, t\}$, this indicates that the model suffers from misspecification. The bridge function serves to mitigate such misspecification issues by enforcing the independence of the residual and the covariates.

### D.4 BRIDGE FUNCTIONS WITH NEGATIVE CONTROLS FOR CAUGRAMER(NC)

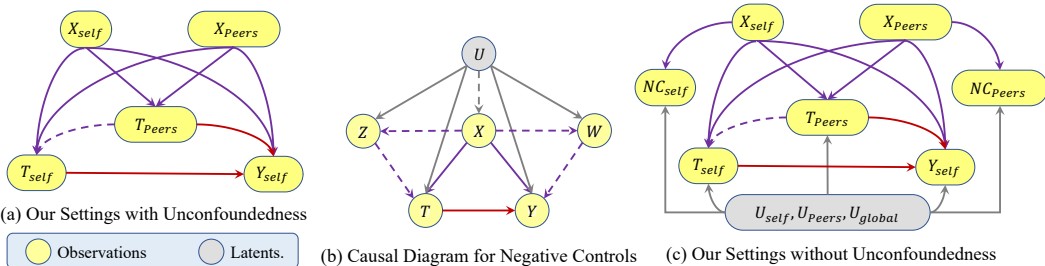

Figure 5: Causal Diagram for Negative Controls and Generalized Networked Settings.

However, in real applications, numerous latent variables may exist, leading to a violation of the Unconfoundedness Assumption. These unmeasured variables can influence both the treatment and the outcome, posing additional challenges for causal inference. When this assumption is violated, negative control variables (NCs) typically are used for detection, reduction, and correction of confounding bias (Kallus et al., 2021; Shi et al., 2020). As shown in the causal diagram in Figure 5(b), negative control variables are variables that are not direct causes of the outcome or are not caused by the treatment, and they are becoming more accessible through web technologies, user profiles, and digitized records (Hu et al., 2023).

**Definition** (Negative Controls (NCs)). Negative Controls comprise two types: NC exposures (NCE, denoted by $z$) and NC outcomes (NCO, denoted by $w$). NCE $z$ cannot directly affect the outcome $y$, and neither NCE $z$ nor the treatment $t$ can affect the NCO $w$, i.e. $z \perp\!\!\!\perp y \mid (x, u)$, $z \perp\!\!\!\perp w \mid (x, u)$, $t \perp\!\!\!\perp w \mid (x, u)$ and $(z, w) \not\perp\!\!\!\perp (x, u)$.

From the causal diagram and definition of negative controls, NCEs $Z$ are proxy variables that are linked to the treatment $T$ through the same unmeasured confounders $U$, and NCOs $W$ are influenced by the same unmeasured confounders $U$ as the outcome $Y$. Then, Miao et al. (2018) and Kallus et al. (2021) construct a bridge function $q(z, t, x)$ for the treatment regression using NCE $z$, and a bridge function $h(w, t, x)$ for the outcome regression using NCO $w$:

$$\frac{\mathbb{E}[t \mid x]}{\mathbb{E}[t \mid x, u]} = \mathbb{E}[\mathbb{E}[t \mid x]q(z, t, x) \mid (t, x, u)], \quad \mathbb{E}[y \mid t, x, u] = \mathbb{E}[h(w, t, x) \mid t, x, u]. \quad (13)$$

The main goal of bridge functions is to learn transformation links between observed variables and negative controls to eliminate confounding effects from the same unmeasured confounders in causal identification. In the presence of unmeasured confounders, we propose using bridge functions optimization to generalize the unconditional maximum moment optimization in Eq. (11). This directly extends our CauGramer to CauGramer(NC) for addressing unmeasured confounders.

**Theorem 2.** *When negative control variables are available, the treatment effects are identified from the bridge functions $q(z, t, x)$ and $h(w, t, x)$ (Kallus et al., 2021).*

$$\hat{h} = \arg\min_{h \in \mathbb{H}} \max_{q \in \mathbb{Q}'} \mathbb{E}[q(z, t, x)(h(w, t, x) - y)], \quad (14)$$

$$\hat{q} = \arg\min_{q \in \mathbb{Q}} \max_{h \in \mathbb{H}'} \mathbb{E}[\mathbb{E}[t \mid x]q(z, t, x)h(w, t, x) - \int h(w, t, x)\mathbb{E}[t \mid x]d\mu(t)]. \quad (15)$$

*where $\mu$ is the base measure associated with treatments $t$, and $\mathbb{H}$ and $\mathbb{Q}$ are hypothesis spaces.*

The proof can be found in Kallus et al. (2021). In our CauGramer, we design a cross-attention moment constraint through a minimax game to make residual independence from input features:

$$\boldsymbol{y}^* = \arg\min_{\hat{\boldsymbol{y}}} \max_{q \in \mathbb{Q}} \mathbb{E}[q(\boldsymbol{r}, \boldsymbol{t})(\boldsymbol{y} - \hat{\boldsymbol{y}})], \quad q(\boldsymbol{r}, \boldsymbol{t}) = \text{Sigmoid}\left(Q \cdot K'\right),$$

The negative control variables $(z, w)$ can be directly embedded into our maximum moment optimization in the above moment constraint and generalized it as follows:

$$\boldsymbol{y}^* = \arg\min_{\hat{\boldsymbol{y}}} \max_{q \in \mathbb{Q}} \mathbb{E}[q(\boldsymbol{r}(\boldsymbol{z}, \boldsymbol{x}), \boldsymbol{t})(\boldsymbol{y} - \hat{\boldsymbol{y}}(\boldsymbol{r}(\boldsymbol{w}, \boldsymbol{x}), \boldsymbol{t}))], q(\boldsymbol{r}(\boldsymbol{z}, \boldsymbol{x}), \boldsymbol{t}) = \text{Sigmoid}\left(Q \cdot K'\right), \quad (16)$$

This allows us to transform the moment optimization into a bridge optimization problem, thereby generalizing CauGramer to CauGramer(NC) and enabling it to address unmeasured confounders without incurring additional computational costs.

## D.5 ADVANCES OF CAUGRAMER

In networked data, traditional Graph neural network (GNN) and its variants (e.g., GCN) often face limitations related to the size of the receptive field and suffer from information homogenization.

- Receptive Field Size: In GNNs, the receptive field refers to the scope of neighboring nodes or edges that influence the representation of a unit. As the network deepens, the receptive field may not expand sufficiently to capture information from distant nodes or edges, leading to limited attention and limited receptive field.

- Information Homogenization: Deepening GNNs can lead to the problem of information homogenization, where the representations of different nodes become increasingly similar as they pass through multiple layers. This can cause loss of important structural or relational information present in the graph.

These constraints pose challenges for traditional GNNs and GCNs in capturing long-distance interference information within graph topologies. Consequently, conventional network interference studies often only account for interference among first-order neighbors (Cai et al., 2023; Forastiere et al., 2021; Guo et al., 2020; Huang et al., 2023; Jiang & Sun, 2022; Lee & Lee, 2022; Liu et al., 2016; 2023; Ma & Tresp, 2021; Qu et al., 2021; Veitch et al., 2019; Zhao et al., 2025; 2024).

Transformers provide a broader perspective to GNNs, overcoming the traditional GNN limitation of receptive field size and enabling the capture of distant node information through their attention mechanism. While recent Transformer variants have achieved superior performance in graph learning (Rampášek et al., 2022; Sui et al., 2022; Ying et al., 2021), they rely on modeling full graph attention through self-attention, which models attention (Query, Key and Value) within the self-embedding, which poses further issues:

- Using embedding from GNNs as input for Transformers, the self-attention mechanism may still struggle to capture the long-distance information of the graph.

- By modeling self-attention mechanisms only on the self-embedding of each unit, these models may fail to capture networked interference.

- Storing and processing the attention weights for every pair of nodes or edges in the graph requires substantial memory resources.

- Modeling full graph attention using self-attention mechanisms can incur substantial computational costs, particularly for large graphs.

These works still suffer from the limited receptive field and high memory and computation costs, and they cannot address the networked interference problem. To address these issues, we propose an interference-agnostic **Cau**sal **Gra**ph Transfor**mer** (**CauGramer**) framework, which extends the computation graph of the Transformer to an $L$-order neighbor network and uses cross-attention to provide a broader receptive field for GCNs (Kipf & Welling, 2016), enabling them to learn complex-sequential interference representations. As shown in Figure 2, in each embedding layer, we construct cross-attention to learn Query $\boldsymbol{q}_i$, Key $\boldsymbol{k}_i$, and Value $\boldsymbol{v}_i$ from self-features ($\boldsymbol{x}_i$), neighbor-features ($\boldsymbol{x}_{\mathcal{P}_i}$), and combined features ($\boldsymbol{x}_i, \boldsymbol{x}_{\mathcal{P}_i}$), respectively, rather than just modeling self-attention on only

each unit's own embedding. This allows us to capture interactions between nodes within and beyond the local neighborhood, rather than focusing solely on each unit's own embedding.

In the $L$-layer CauGramer embedding with $M$-head attention, we use GCNs to aggregate $L$-order neighbor information and topology structure, while cross-attention provides a broader receptive field in GCNs for learning complex-sequential interference representations. As the network's depth $L$ increases, the scope of our GCNs with cross-attention expands, enabling the embedding of all potential interference peers into representations $\boldsymbol{R}_x$ and $\boldsymbol{R}_t$. Furthermore, we reformulate the Transformer decoder with cross-attention minimax constraints to precisely estimate the main, peer, and total effects. When negative control variables are available, the minimax constraint in the objective function (Eq. (12)) of CauGramer acts as the bridge function in Theorem 2, thereby empowering CauGramer to generalize effectively and address the challenges posed by unmeasured confounders.

# E  DATASETS, SUPPLEMENTAL EXPERIMENTS, AND HYPER-PARAMETERS

In this section, we introduce the dataset details used in this paper, conduct hyper-parameter optimization analysis (Figure 6 and Table 5), and time-cost analysis (Table 6). Additionally, we present supplementary experiments to demonstrate the robustness of CauGramer under various scenarios, including interference strength (Figure 7), interference functions (Table 7), unmeasured confounding (Table 8). Furthermore, we include comparisons against additional baselines, including graph Transformers and conventional networked methods, in Tables 9 and 10.

## E.1  DATASETS DETAILS

In real-world applications, the ground-truth causal relations are unknown, and thus it is impossible to evaluate the causal estimation performance of CauGramer directly. Therefore, following existing works (Veitch et al., 2019; Jiang & Sun, 2022; Guo et al., 2020; 2021; Ma et al., 2021; Cai et al., 2023), we generate semi-synthetic data from BlogCatalog (BC) and Flickr datasets [3], where the features ($\boldsymbol{x}$) and social networks ($\boldsymbol{A}$) are real, while treatments ($\boldsymbol{t}$), outcomes ($\boldsymbol{y}$), and interference ($\boldsymbol{E}$) are simulated:

- BlogCatalog (BC) (Guo et al., 2020; Jiang & Sun, 2022). BlogCatalog (BC) is an online community that offers blog services, where each blogger represents a study unit within the dataset. The relationships between these units form the social network, with each edge denoting a social link. The features are bag-of-words representations of keywords in the bloggers' descriptions.

- Flickr (Guo et al., 2020; Jiang & Sun, 2022). Flickr is an online social network providing image and video sharing services. The dataset is constructed by forming links between images that share common metadata. In this dataset, each instance is a user, and each edge represents the social relationship between two users. The features of each user represent a list of tags indicating their interests.

We use the same linear discriminant analysis (LDA) and the same data simulation by Jiang & Sun (2022) on BlogCatalog and Flickr datasets. Moreover, Jiang & Sun (2022) use METIS (Karypis & Kumar, 1998) to partition the original networks into three sub-networks as train/valid/test data with 2482, 2461, and 2358 samples in Flickr, and 1784, 1716, and 1696 samples in BlogCatalog. In the testing data, we will randomly sample binary treatments from a Bernoulli distribution with a probability of 0.5 to simulate manipulation operations. Then, we use the same data simulation by Jiang & Sun (2022) to evaluate CauGramer's performance on original data with known intervention graphs and summary functions for constant treatment effects. Then, we design a series of generalized heterogeneous treatment effects datasets with unknown summary functions across various settings.

**For the conventional networked setting with known interference graph and known summary function (Table 2)**, similar to Jiang & Sun (2022), we generate the treatments and potential outcomes with known intervention graphs ($\boldsymbol{A} = \boldsymbol{E}$) and summary functions ($z_i = \frac{1}{|\mathcal{N}_i|} \sum_{j \in \mathcal{N}_i} t_j$), as follows. Let $p_i^t = \sigma(\boldsymbol{w}_x^{(1)} \cdot \boldsymbol{x}_i), p_{\mathcal{N}_i}^t = \frac{1}{|\mathcal{N}_i|} \sum_{j \in \mathcal{N}_i} p_j^t, p_i^y = \sigma(\boldsymbol{w}_x^{(2)} \cdot \boldsymbol{x}_i), p_{\mathcal{N}_i}^y = \frac{1}{|\mathcal{N}_i|} \sum_{j \in \mathcal{N}_i} p_j^y$, where

---

[3]The BlogCatalog and Flickr datasets are available at: `https://github.com/songjiang0909/Causal-Inference-on-Networked-Data`.

$\{\boldsymbol{w}_x^{(1)}, \boldsymbol{w}_x^{(2)}\}$ are random coefficient vectors sampled from uniform distribution $\mathcal{U}(-1.0, 1.0)$.

$$t_i = \mathbf{1}(p_i^t + p_{\mathcal{N}_i}^t > \overline{p_i^t} + \overline{p_{\mathcal{N}_i}^t}), \ z_i = \frac{1}{|\mathcal{N}_i|} \sum_{j \in \mathcal{N}_i} t_j, \ y_i = t_i + z_i + p_i^y + 0.5 p_{\mathcal{N}_i}^y + \epsilon_i, \qquad (17)$$

where $\mathbf{1}$ is an indicator function, $\overline{p_i^t}$ and $\overline{p_{\mathcal{N}_i}^t}$ denote the average value of $p_i^t$ and $p_{\mathcal{N}_i}^t$. The variable $\epsilon_i \sim \mathcal{N}(0, 0.1)$ denotes a measurement error.

**For the generalized networked setting with known interference graph and unknown summary function (Table 3)**, under interference graph $\boldsymbol{E}$, we use the same functions by Cai et al. (2023); Jiang & Sun (2022) to define:

$$p_i^t = \sigma(\boldsymbol{w}_x^{(1)} \cdot \boldsymbol{x}_i), \quad p_{\mathcal{P}_i}^t = \frac{1}{|\mathcal{P}_i|} \sum_{j \in \mathcal{P}_i} p_j^t, \quad p_i^y = \sigma(\boldsymbol{w}_x^{(2)} \cdot \boldsymbol{x}_i), \quad p_{\mathcal{P}_i}^y = \frac{1}{|\mathcal{P}_i|} \sum_{j \in \mathcal{P}_i} p_j^y,$$

$$z_i^{NCE} = \sigma(\boldsymbol{w}_x^{(3)} \cdot \boldsymbol{x}_i) - \frac{1}{|\mathcal{P}_i|} \sum_{j \in \mathcal{P}_i} \sigma(\boldsymbol{w}_x^{(3)} \cdot \boldsymbol{x}_j), \ w_i^{NCO} = \sigma(\boldsymbol{w}_x^{(4)} \cdot \boldsymbol{x}_i) - \frac{1}{|\mathcal{P}_i|} \sum_{j \in \mathcal{P}_i} \sigma(\boldsymbol{w}_x^{(4)} \cdot \boldsymbol{x}_j),$$

where $\{\boldsymbol{w}_x^{(1)}, \boldsymbol{w}_x^{(2)}, \boldsymbol{w}_x^{(3)}, \boldsymbol{w}_x^{(4)}\}$ are random coefficient vectors sampled from uniform distribution $\mathcal{U}(-1.0, 1.0)$, $z_i^{NCE}$ and $w_i^{NCO}$ are NCs. Then, we generate treatments based on the priority level $s_i = x_i^{(1)} + x_i^{(2)}$, where $x_i^{(1)}$ and $x_i^{(2)}$ denotes the 1st and 2nd variable in the feature of unit $i$.

$$m_i = p_i^t + p_{\mathcal{P}_i}^t, \quad \tilde{t}_i = m_i + \frac{1}{|\mathcal{P}_i|} \sum_{j \in \mathcal{P}_i} t_j * \mathbf{1}(s_j > s_i), \quad t_i = \mathbf{1}\left(\sigma(\tilde{t}_i) > \overline{m_i}\right). \qquad (18)$$

In the treatment generation process, we simulate treatment interference based on the priority level $s_i$ to avoid cycles. Then, we simulate outcomes with heterogeneous effects under networked interference.

$$y_i = t_i \cdot \tau_i^{IME} + \tau \cdot \tau_i^{IPE} + p_i^y + 0.5 p_{\mathcal{N}_i}^y + \epsilon_i, \qquad (19)$$

$$\tau_i^{IME} = \sqrt{1 + p_i^y - p_{\mathcal{N}_i}^y}, \quad \tau_i^{IPE} = \frac{1}{|\mathcal{P}_i|} \sum_{j \in \mathcal{P}_i} t_j * \|\boldsymbol{x}_i - \boldsymbol{x}_j\|_2^2, \qquad (20)$$

where $\|\boldsymbol{x}_i - \boldsymbol{x}_j\|_2^2$ denotes the squared Euclidean distance, and $\rho \in \{0.2, 0.4, 0.6, 0.8, 1.0, 1.5, 2.0\}$ represents the different interference level in the Figure 7.

**Moreover, to simulate an unmeasured confounder setting (Table 3),** we will randomly select two variables as latent variables and remove them in the feature of unit $i$. In Table 3, CauGramer(UC) directly employs models with missing variables to estimate treatment effects, while CauGramer(NC) uses $\{z_i^{NCE}, w_i^{NCO}\}$ as NCs to identify treatment effects.

**In the generalized networked data with unknown interference graph and unknown summary functions (Figure 3),** we design three interference graphs $\boldsymbol{E}$ which are not the same as the social network $\boldsymbol{A}$. 'S.G.1N' and 'S.G.2N' represent subgraphs from graphs composed of 1st and 2nd-order neighbors marked by Euclidean distance $(0.1 < \|\boldsymbol{x}_i^{(:5)} - \boldsymbol{x}_j^{(:5)}\|_2^2 < 0.2)$, respectively, while 'Random' denotes randomly generated graphs by Euclidean distance $(0.20 < \|\boldsymbol{x}_i^{(:5)} - \boldsymbol{x}_j^{(:5)}\|_2^2 < 0.25)$. Additionally, for comparison, we denote the 'Known' graph as the social network. Then, we use these interference graphs to simulate data by the above simulation process (Eqs. (18) and (19)).

**Then, in the limited budget experiments (Figure 4),** treatment sources like COVID vaccinations are limited, requiring treatment assignment to adhere to varying budget constraints ('BudgX%') rather than predefined treatment assignment policies ('Assigned'). Given the training data generated by equations (18) and (19), in testing data generation, we randomly sample binary treatments from a Bernoulli distribution with a probability of $\{20\%, 50\%, 80\%, 100\%\}$ to simulate different budget on treatment sources. This enables us to evaluate the performance of the proposed CauGramer in estimating treatment effects under varying budget constraints.

**For different interference levels (Figure 7)),** we adjust the coefficient $\rho$ ranging from $\{0.2, 0.4, 0.6, 0.8, 1.0, 1.5, 2.0\}$ in equation (19) to simulate various levels of interference.

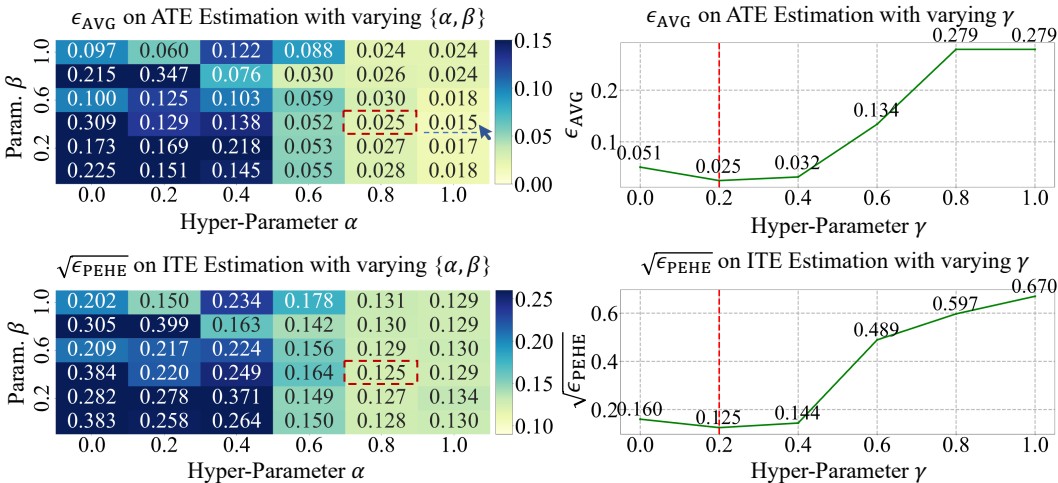

Figure 6: Hyper-Parameter Optimization on Flickr Dataset: CauGramer optimizes three hyper-parameters, $\{\alpha, \beta, \gamma\} \in \{0.0, 0.2, 0.4, 0.6, 0.8, 1.0\}$. The minimum regression error on the validation data indicates the optimal hyper-parameters ($\{\alpha, \beta, \gamma\} = \{0.8, 0.4, 0.2\}$) for ITE estimation, marked by the red dashed box and lines. A higher value on $\alpha$ might improve ATE estimation performance.

### E.2 HYPER-PARAMETERS OPTIMIZATION

We first review the objective function (Eq. (12)) of the proposed CauGramer:

$$\arg \min_{\hat{t}, \hat{y}} \max_{q \in \mathbb{Q}} \mathcal{L} = (1 - \gamma) \left( \text{MSE}(\hat{y}, y) + \alpha \cdot \text{IPM}(r, t) + \beta \cdot \text{Cross\_Entropy}(\hat{t}, t) \right) + \gamma \cdot \mathbb{E}[q(r, t)(y - \hat{y})].$$

In our CauGramer, there are three hyper-parameters, i.e., $\{\alpha, \beta, \gamma\} \in \{0.0, 0.2, 0.4, 0.6, 0.8, 1.0\}$. Among them, $\alpha$ and $\beta$ jointly control representation balancing for adjusting confounding bias in the estimation of main treatment effects and peer-treatment effects. Additionally, $\beta$ serves for aggregating neighbors' representations by maximizing the predictive power of $r_{j:j \in \mathcal{P}_i}$ on self-treatment $t_i$. Through learning representations with reduced Inter-Profile Matching (IPM) distance between treated and control units, $\alpha$ and $\beta$ facilitate a bias-variance trade-off. The hyper-parameter $\gamma$ serves as an interpolation coefficient, where $\gamma = 0$ corresponds to outcome regression with representation balancing, and $\gamma = 1$ represents a conditional moment model. By adjusting $\gamma$, we integrate the Conditional Moment model into the objective function, providing dual guarantees to mitigate confounding bias effectively.

In this paper, we adopt the minimum regression error on the validation data to determine the optimal hyper-parameters $\{\alpha, \beta, \gamma\} \in \{0.0, 0.2, 0.4, 0.6, 0.8, 1.0\}$. First, We search for $\alpha, \beta \in 0.0, 0.2, 0.4, 0.6, 0.8, 1.0$ to identify the hyper-parameter combination corresponding to the minimum validation error, while fixing $\gamma = 0.2$. Subsequently, utilizing the optimal parameters $\alpha, \beta$ determined from the previous search, we further explore $\gamma \in 0.0, 0.2, 0.4, 0.6, 0.8, 1.0$ to identify the hyper-parameter combination corresponding to the minimum validation error. Considering the Flickr dataset as an example, as depicted in Figure 6, we determine the hyper-parameters that correspond to the smallest validation regression error, which also indicates the smallest $\sqrt{\epsilon_{\text{PEHE}}}$ on the Flickr dataset, marked by the red box and lines. The optimal hyper-parameters are $\alpha, \beta, \gamma = 0.8, 0.4, 0.2$ for both the BlogCatalog and Flickr datasets.

Furthermore, as shown in Table 5, we can use a grid search on the validation dataset to choose the optimal $L$ value and $M$ value in CauGramer with $L$-layer and $M$-layer. In this paper, we propose a $L$-layers (default: 2) $M$-heads (default: 3) cross-attention GCN to learn the representation $R_x = g_x(x, A)$. For interference information, we propose a $L$-layers (default: $L = 1$ for main results and $L = 2$ for unknown interference graph in Figure 3) $M$-heads (default: 3) cross-attention GCN to learn the representation $R_t = g_t(t, A)$.

Table 5: Hyper-Parameter Optimization on Flickr Dataset: CauGramer with $L$-layer and $M$-heads.

| Flickr | AME | APE | ATE | IME | IPE | ITE |
|---|---|---|---|---|---|---|
| L=1, M=3 | $0.039_{\pm0.02}$ | $0.320_{\pm0.10}$ | $0.290_{\pm0.07}$ | $0.066_{\pm0.02}$ | $0.358_{\pm0.09}$ | $0.339_{\pm0.07}$ |
| L=2, M=2 | $0.093_{\pm0.05}$ | $0.230_{\pm0.11}$ | $0.203_{\pm0.15}$ | $0.137_{\pm0.07}$ | $0.282_{\pm0.09}$ | $0.281_{\pm0.12}$ |
| L=2, M=3 | $0.074_{\pm0.05}$ | $0.223_{\pm0.06}$ | $0.203_{\pm0.07}$ | $0.114_{\pm0.07}$ | $0.222_{\pm0.06}$ | $0.270_{\pm0.08}$ |
| L=2, M=4 | $0.048_{\pm0.05}$ | $0.260_{\pm0.09}$ | $0.228_{\pm0.08}$ | $0.083_{\pm0.07}$ | $0.305_{\pm0.08}$ | $0.288_{\pm0.06}$ |
| L=3, M=3 | $0.112_{\pm0.07}$ | $0.437_{\pm0.08}$ | $0.458_{\pm0.09}$ | $0.166_{\pm0.09}$ | $0.467_{\pm0.07}$ | $0.488_{\pm0.09}$ |

Table 6: Training Time(s) of Various Methods in A Single Execution on Both Datasets.

| Method | BlogCatalog | Flickr | Method | BlogCatalog | Flickr |
|---|---|---|---|---|---|
| CFRNet | 20.9s | 35.3s | DRlearner | 29.7s | 33.9s |
| NetDeconf | 53.9s | 101.6s | G-HSIC | 508.7s | 1054.3s |
| SPNet | 460.2s | 815.8s | CAL | 108.8s | 204.1s |
| Graphormer | 788.5s | 1225.4s | RRNet | 628.4s | 1119.1s |
| CEVAE | 74.0s | 106.5s | NetEst | 1226.5 | 2125.1s |
| GraphGPS | 66.4s | 124.4s | SAT | 104.3s | 225.7s |
| GDML | 8.4s | 24.4s | UNITE | 76.3s | 111.7s |
| CNE | 66.5s | 105.9s | CauGramer | 79.7s | 155.5s |

## E.3 TIME COST ANALYSIS

For each experiment, we conduct 10 independent replications to analyze the average running time (in seconds) for the proposed model in a single execution and compare it to baselines. From the results in Table 6, we observe that among all transformer-based methods (CAL, SPNet, Graphormer), CauGramer exhibits the shortest running time (79.7s on BlogCatalog data, 155.5s on Flickr data). Unlike traditional Transformer variants, which rely on computationally expensive full graph attention mechanisms through self-attention, CauGramer adopts a more streamlined approach. By redesigning the computation graph of the Transformer, CauGramer effectively models sequential-structure interference representations using GCN with cross-attention, bypassing the need for full graph attention through self-attention. Moreover, compared to the optimal baseline NetEst, CauGramer achieves a notable speedup, being 12 times faster. This highlights CauGramer's performance and computational efficiency in graph-based modeling tasks.

## E.4 SUPPLEMENTAL EXPERIMENTS ACROSS VARIOUS INTERFERENCE FORMS

The experiments presented in Tables 2 and 3, as well as Figures 3 and 4, in the main text, have demonstrated the superiority of CauGramer over existing methods, whether in conventional networked settings or in generalized networked settings with or without unmeasured confounders, and under limited budget constraints. To simulate more complex real-world applications, we further conduct additional experiments on varying interference levels $\rho \in \{0.2, 0.4, 0.6, 0.8, 1.0, 1.5, 2.0\}$.

We report the estimation errors of Average Total Effect (ATE) and Individual Total Effect (ITE) of treatment on BlogCatalog and Flickr datasets across varying interference levels $\rho \in \{0.2, 0.4, 0.6, 0.8, 1.0, 1.5, 2.0\}$. As shown in Figure 7, as the interference level increases, the proposed CauGramer algorithm consistently outperforms existing methods. When NCs are accessible, CauGramer(NC) effectively eliminates unmeasured confounding bias, approaching the performance of CauGramer trained on full data. For causal effect estimation, we notice that all baselines' errors increase with higher interference levels. However, CauGramer and its variant CauGramer(NC) exhibit superior performance in predicting the individual total effects of treatment. Moreover, in ATE estimation, both CauGramer and CauGramer(NC) remain robust, with errors consistently below 0.1 or 0.05 on the BlogCatalog and Flickr datasets. This demonstrates that our CauGramer can learn complex intervention representations. By fully embedding peer information, the balancing constraints and minimax moment constraints provide accurate causal prediction. Consequently, even under large interference, the proposed CauGramer maintains low estimation errors, showcasing its causal robustness and effectiveness in mitigating confounding bias.

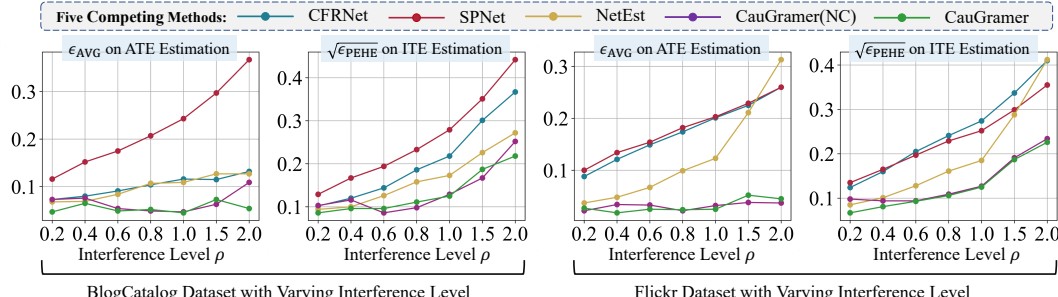

Figure 7: Results ($\epsilon_{\text{AVG}}$ and $\sqrt{\epsilon_{\text{PEHE}}}$) of Total Effect Estimation on BlogCatalog and Flickr Datasets across Varying Intervention Level $\rho \in \{0.2, 0.4, 0.8, 1.0, 1.5, 2.0\}$.

Table 7: Results of the Proposed CauGramer across Various Interferences: (1) aggregation interference Jiang & Sun (2022) (Eq. (17): $\tau_i = \frac{1}{|\mathcal{N}_i|} \sum_{j \in \mathcal{N}_i} t_j$) ; (2) interaction interference (Eq. (20): $\tau_i = \frac{1}{|\mathcal{P}_i|} \sum_{j \in \mathcal{P}_i} t_j * \|\boldsymbol{x}_i - \boldsymbol{x}_j\|_2^2$); (3) external interference (Eq. (21): $\tau_i = \frac{1}{|\mathcal{P}_i|} \sum_{j \in \mathcal{P}_i} \sin(t_j * \|\boldsymbol{x}_j\|_2^2)$).

| BC | Interference | AME | APE | ATE | IME | IPE | ITE |
|---|---|---|---|---|---|---|---|
| SPNet | Eq. (17) | $0.066_{\pm 0.04}$ | $0.223_{\pm 0.10}$ | $0.203_{\pm 0.10}$ | $0.100_{\pm 0.05}$ | $0.230_{\pm 0.10}$ | $0.213_{\pm 0.09}$ |
| NetEst | Eq. (17) | $0.076_{\pm 0.02}$ | $0.078_{\pm 0.02}$ | $0.065_{\pm 0.03}$ | $0.099_{\pm 0.03}$ | $0.092_{\pm 0.01}$ | $0.117_{\pm 0.01}$ |
| CauGramer | Eq. (17) | $0.054_{\pm 0.03}$ | $0.034_{\pm 0.03}$ | $0.039_{\pm 0.04}$ | $0.075_{\pm 0.07}$ | $0.044_{\pm 0.02}$ | $0.063_{\pm 0.04}$ |
| SPNet | Eq. (20) | $0.083_{\pm 0.06}$ | $0.212_{\pm 0.11}$ | $0.243_{\pm 0.11}$ | $0.131_{\pm 0.07}$ | $0.246_{\pm 0.10}$ | $0.279_{\pm 0.10}$ |
| NetEst | Eq. (20) | $0.089_{\pm 0.03}$ | $0.077_{\pm 0.02}$ | $0.109_{\pm 0.03}$ | $0.147_{\pm 0.03}$ | $0.145_{\pm 0.01}$ | $0.173_{\pm 0.02}$ |
| CauGramer | Eq. (20) | $0.073_{\pm 0.06}$ | $0.057_{\pm 0.04}$ | $0.045_{\pm 0.04}$ | $0.090_{\pm 0.05}$ | $0.118_{\pm 0.02}$ | $0.125_{\pm 0.01}$ |
| SPNet | Eq. (21) | $0.121_{\pm 0.06}$ | $0.273_{\pm 0.09}$ | $0.273_{\pm 0.10}$ | $0.179_{\pm 0.08}$ | $0.283_{\pm 0.09}$ | $0.284_{\pm 0.10}$ |
| NetEstimator | Eq. (21) | $0.083_{\pm 0.04}$ | $0.064_{\pm 0.01}$ | $0.072_{\pm 0.02}$ | $0.121_{\pm 0.05}$ | $0.078_{\pm 0.01}$ | $0.108_{\pm 0.02}$ |
| CauGramer | Eq. (21) | $0.063_{\pm 0.07}$ | $0.039_{\pm 0.02}$ | $0.052_{\pm 0.03}$ | $0.108_{\pm 0.09}$ | $0.051_{\pm 0.02}$ | $0.074_{\pm 0.03}$ |
| **Flickr** | **Interference** | **AME** | **APE** | **ATE** | **IME** | **IPE** | **ITE** |
| SPNet | Eq. (17) | $0.054_{\pm 0.03}$ | $0.121_{\pm 0.05}$ | $0.131_{\pm 0.05}$ | $0.079_{\pm 0.05}$ | $0.131_{\pm 0.05}$ | $0.142_{\pm 0.05}$ |
| NetEst | Eq. (17) | $0.063_{\pm 0.04}$ | $0.055_{\pm 0.03}$ | $0.073_{\pm 0.05}$ | $0.095_{\pm 0.05}$ | $0.070_{\pm 0.03}$ | $0.105_{\pm 0.05}$ |
| CauGramer | Eq. (17) | $0.028_{\pm 0.03}$ | $0.019_{\pm 0.01}$ | $0.032_{\pm 0.02}$ | $0.047_{\pm 0.04}$ | $0.032_{\pm 0.01}$ | $0.051_{\pm 0.02}$ |
| SPNet | Eq. (20) | $0.096_{\pm 0.06}$ | $0.166_{\pm 0.07}$ | $0.203_{\pm 0.08}$ | $0.147_{\pm 0.07}$ | $0.213_{\pm 0.06}$ | $0.252_{\pm 0.07}$ |
| NetEst | Eq. (20) | $0.069_{\pm 0.05}$ | $0.067_{\pm 0.05}$ | $0.123_{\pm 0.07}$ | $0.136_{\pm 0.06}$ | $0.156_{\pm 0.03}$ | $0.185_{\pm 0.05}$ |
| CauGramer | Eq. (20) | $0.058_{\pm 0.05}$ | $0.030_{\pm 0.03}$ | $0.025_{\pm 0.03}$ | $0.096_{\pm 0.06}$ | $0.111_{\pm 0.01}$ | $0.125_{\pm 0.01}$ |
| SPNet | Eq. (21) | $0.088_{\pm 0.04}$ | $0.202_{\pm 0.05}$ | $0.204_{\pm 0.04}$ | $0.136_{\pm 0.05}$ | $0.219_{\pm 0.05}$ | $0.223_{\pm 0.04}$ |
| NetEstimator | Eq. (21) | $0.060_{\pm 0.04}$ | $0.073_{\pm 0.02}$ | $0.073_{\pm 0.04}$ | $0.102_{\pm 0.04}$ | $0.093_{\pm 0.02}$ | $0.114_{\pm 0.02}$ |
| CauGramer | Eq. (21) | $0.057_{\pm 0.02}$ | $0.010_{\pm 0.01}$ | $0.046_{\pm 0.01}$ | $0.097_{\pm 0.03}$ | $0.040_{\pm 0.00}$ | $0.086_{\pm 0.02}$ |

Additionally, in Tables 2 and 3, we simulate two different forms of interference using Eqs. (17) and (20). Eq. (17) represents aggregation interference, commonly used in traditional networked interference works (Cai et al., 2023; Jiang & Sun, 2022; Ma & Tresp, 2021). Eq. (20) simulates heterogeneous interaction interference, based on the difference between self-attributes and peer attributes within the k-hop subgraph. In this section, we simulate an external interference using only peer information within the k-hop subgraph, mimicking constant external influences:

$$\tau_i^{IPE} = \frac{1}{|\mathcal{P}_i|} \sum_{j \in \mathcal{P}_i} t_j * \|\boldsymbol{x}_j\|_2^2, \tag{21}$$

We integrate experiments across various interference forms (Eqs. (17), (20), and (21)) into Table 7. Under various interference forms, the proposed CauGramer remains the best method for estimating the main effects, peer effects, and total effects of treatment.

Table 8: Results of the Proposed CauGramer under Various Unmeasured Confounders (UCs). Different subscripts on $x_{[...]}$ denote masked variables, while NULL indicates all variables are observable.

| BC | UCs | AME | APE | ATE | IME | IPE | ITE |
|---|---|---|---|---|---|---|---|
| CauGramer(UC) | $x_{[1,3,5,7,8,9]}$ | $0.128_{\pm0.10}$ | $0.073_{\pm0.04}$ | $0.094_{\pm0.04}$ | $0.182_{\pm0.14}$ | $0.131_{\pm0.02}$ | $0.166_{\pm0.04}$ |
| CauGramer(UC) | $x_{[1,3,5,7]}$ | $0.111_{\pm0.07}$ | $0.074_{\pm0.03}$ | $0.080_{\pm0.04}$ | $0.167_{\pm0.10}$ | $0.125_{\pm0.02}$ | $0.157_{\pm0.03}$ |
| CauGramer(UC) | $x_{[6,7,8,9]}$ | $0.111_{\pm0.07}$ | $0.074_{\pm0.04}$ | $0.087_{\pm0.06}$ | $0.163_{\pm0.09}$ | $0.127_{\pm0.02}$ | $0.162_{\pm0.04}$ |
| CauGramer(UC) | $x_{[8,9]}$ | $0.109_{\pm0.07}$ | $0.067_{\pm0.04}$ | $0.077_{\pm0.05}$ | $0.159_{\pm0.10}$ | $0.125_{\pm0.02}$ | $0.149_{\pm0.03}$ |
| CauGramer(NC) | $x_{[1,3,5,7,8,9]}$ | $0.092_{\pm0.06}$ | $0.068_{\pm0.02}$ | $0.057_{\pm0.04}$ | $0.134_{\pm0.08}$ | $0.122_{\pm0.02}$ | $0.134_{\pm0.03}$ |
| CauGramer(NC) | $x_{[1,3,5,7]}$ | $0.099_{\pm0.06}$ | $\mathbf{0.047_{\pm0.03}}$ | $0.061_{\pm0.04}$ | $0.148_{\pm0.08}$ | $\mathbf{0.109_{\pm0.01}}$ | $0.137_{\pm0.03}$ |
| CauGramer(NC) | $x_{[6,7,8,9]}$ | $0.085_{\pm0.07}$ | $0.057_{\pm0.03}$ | $0.062_{\pm0.05}$ | $0.138_{\pm0.10}$ | $0.123_{\pm0.02}$ | $0.137_{\pm0.03}$ |
| CauGramer(NC) | $x_{[8,9]}$ | $0.092_{\pm0.07}$ | $\underline{0.055_{\pm0.03}}$ | $\underline{0.047_{\pm0.03}}$ | $0.139_{\pm0.09}$ | $\underline{0.117_{\pm0.02}}$ | $\underline{0.129_{\pm0.02}}$ |
| SPNet | NULL | $\underline{0.083_{\pm0.06}}$ | $0.212_{\pm0.11}$ | $0.243_{\pm0.11}$ | $\underline{0.131_{\pm0.07}}$ | $0.246_{\pm0.10}$ | $0.279_{\pm0.10}$ |
| NetEst | NULL | $0.089_{\pm0.03}$ | $0.077_{\pm0.02}$ | $0.109_{\pm0.02}$ | $0.147_{\pm0.03}$ | $0.145_{\pm0.01}$ | $0.173_{\pm0.02}$ |
| CauGramer | NULL | $\mathbf{0.073_{\pm0.06}}$ | $0.057_{\pm0.04}$ | $\mathbf{0.045_{\pm0.04}}$ | $\mathbf{0.090_{\pm0.05}}$ | $0.118_{\pm0.02}$ | $\mathbf{0.125_{\pm0.01}}$ |

| Flickr | UCs | AME | APE | ATE | IME | IPE | ITE |
|---|---|---|---|---|---|---|---|
| CauGramer(UC) | $x_{[1,3,5,7,8,9]}$ | $0.082_{\pm0.04}$ | $0.055_{\pm0.03}$ | $0.080_{\pm0.04}$ | $0.125_{\pm0.06}$ | $0.124_{\pm0.01}$ | $0.153_{\pm0.01}$ |
| CauGramer(UC) | $x_{[1,3,5,7]}$ | $0.096_{\pm0.04}$ | $0.062_{\pm0.01}$ | $0.065_{\pm0.04}$ | $0.140_{\pm0.06}$ | $0.130_{\pm0.01}$ | $0.155_{\pm0.01}$ |
| CauGramer(UC) | $x_{[6,7,8,9]}$ | $0.153_{\pm0.05}$ | $0.063_{\pm0.03}$ | $0.071_{\pm0.06}$ | $0.220_{\pm0.07}$ | $0.127_{\pm0.02}$ | $0.159_{\pm0.04}$ |
| CauGramer(UC) | $x_{[8,9]}$ | $0.091_{\pm0.07}$ | $0.067_{\pm0.02}$ | $0.056_{\pm0.04}$ | $0.141_{\pm0.08}$ | $0.128_{\pm0.01}$ | $0.150_{\pm0.02}$ |
| CauGramer(NC) | $x_{[1,3,5,7,8,9]}$ | $0.081_{\pm0.03}$ | $0.035_{\pm0.02}$ | $0.051_{\pm0.02}$ | $0.124_{\pm0.04}$ | $0.114_{\pm0.01}$ | $0.140_{\pm0.01}$ |
| CauGramer(NC) | $x_{[1,3,5,7]}$ | $\underline{0.063_{\pm0.03}}$ | $0.038_{\pm0.02}$ | $0.062_{\pm0.06}$ | $\underline{0.101_{\pm0.03}}$ | $0.115_{\pm0.01}$ | $0.146_{\pm0.04}$ |
| CauGramer(NC) | $x_{[6,7,8,9]}$ | $0.067_{\pm0.03}$ | $\underline{0.034_{\pm0.01}}$ | $0.043_{\pm0.01}$ | $0.110_{\pm0.03}$ | $0.113_{\pm0.00}$ | $0.134_{\pm0.01}$ |
| CauGramer(NC) | $x_{[8,9]}$ | $0.073_{\pm0.05}$ | $0.038_{\pm0.02}$ | $\underline{0.032_{\pm0.03}}$ | $0.107_{\pm0.06}$ | $\underline{0.112_{\pm0.02}}$ | $\underline{0.127_{\pm0.01}}$ |
| SPNet | NULL | $0.096_{\pm0.06}$ | $0.166_{\pm0.07}$ | $0.203_{\pm0.08}$ | $0.147_{\pm0.07}$ | $0.213_{\pm0.06}$ | $0.252_{\pm0.07}$ |
| NetEst | NULL | $0.069_{\pm0.05}$ | $0.067_{\pm0.05}$ | $0.123_{\pm0.07}$ | $0.136_{\pm0.06}$ | $0.156_{\pm0.03}$ | $0.185_{\pm0.05}$ |
| CauGramer | NULL | $\mathbf{0.058_{\pm0.05}}$ | $\mathbf{0.030_{\pm0.03}}$ | $\mathbf{0.025_{\pm0.03}}$ | $\mathbf{0.096_{\pm0.06}}$ | $\mathbf{0.111_{\pm0.01}}$ | $\mathbf{0.125_{\pm0.01}}$ |

## E.5 SUPPLEMENTAL EXPERIMENTS ACROSS VARIOUS FORMS OF LATENT VARIABLES

In the presence of unmeasured confounders (UCs), we propose using bridge functions of negative control variables ($\min\max \mathbb{E}[q(\boldsymbol{z}, \boldsymbol{x}, \boldsymbol{t}) \cdot (\boldsymbol{y} - \hat{\boldsymbol{y}}(\boldsymbol{w}, \boldsymbol{x}, \boldsymbol{t}))]$ in Eq. (14)) to generalize the unconditional maximum moment optimization in Eq. (11) as $\min\max E[q(\boldsymbol{r}(\boldsymbol{x}), \boldsymbol{t}) \cdot (\boldsymbol{y} - \hat{\boldsymbol{y}}(\boldsymbol{r}(\boldsymbol{x}), \boldsymbol{t}))]$. This extends our CauGramer to CauGramer(NC) for settings with latent variables. Then, we mask different variables ($x_{[8,9]}, x_{[6,7,8,9]}, x_{[1,3,5,7]}, x_{[1,3,5,7,8,9]}$) from the observational data, where the subscripts $x_{[...]}$ denote masked variables, to simulate various unmeasured confounders settings. The results in Table 8 demonstrate the superiority of CauGramer(NC). Even with more than five latent variables, the estimation error of CauGramer(NC) remains lower than that of the two most competing methods, SPNet and NetEst.

## E.6 SUPPLEMENTAL EXPERIMENTS WITH MORE GRAPH TRANSFORMER BASELINES

We further conduct comparisons with four competing graph Transformer methods, including SAT (Chen et al., 2022), GraphGPS (Rampášek et al., 2022), CAL (Sui et al., 2022), and Graphormer (Ying et al., 2021), and reported $\epsilon_{\text{AVG}}$ of AME, APE and ATE and $\sqrt{\epsilon_{\text{PEHE}}}$ of IME, IPE and ITE on the BlogCatalog (BC) and Flickr datasets in the Table 9.

- SAT (Chen et al., 2022) incorporates structural information into self-attention by extracting a subgraph representation rooted at each node before computing attention.

- GraphGPS (Rampášek et al., 2022) proposes a hybrid MPNN+Transformer architecture, in which we use GCNs to support the MPNNs.

- CAL (Sui et al., 2022) designs two soft mask graphs for node attention and edge attention and conducts implicit backdoor adjustment for graph tasks.

- Graphormer (Ying et al., 2021) integrates the encoder of Transformer (Vaswani et al., 2017) into node, edge, and graph embedding learning for graph tasks.

Table 9: More Baselines Combining Transformer and GCN. For the sake of fairness, we add a deconfounder module (+Deconf) to these frameworks for treatment effect estimation.

| BlogCatalog(BC) | AME | APE | ATE | IME | IPE | ITE |
|---|---|---|---|---|---|---|
| Graphormer+Deconf | $\mathbf{0.068}_{\pm 0.02}$ | $0.365_{\pm 0.02}$ | $0.362_{\pm 0.08}$ | $0.153_{\pm 0.03}$ | $0.394_{\pm 0.02}$ | $0.418_{\pm 0.08}$ |
| CAL+Deconf | $0.151_{\pm 0.03}$ | $0.308_{\pm 0.05}$ | $0.364_{\pm 0.07}$ | $0.230_{\pm 0.04}$ | $0.333_{\pm 0.06}$ | $0.385_{\pm 0.07}$ |
| GraphGPS+Deconf | $0.091_{\pm 0.09}$ | $0.195_{\pm 0.013}$ | $0.247_{\pm 0.17}$ | $\underline{0.142}_{\pm 0.12}$ | $0.224_{\pm 0.17}$ | $\underline{0.279}_{\pm 0.22}$ |
| SAT+Deconf | $0.176_{\pm 0.15}$ | $\underline{0.102}_{\pm 0.08}$ | $\underline{0.188}_{\pm 0.15}$ | $0.251_{\pm 0.17}$ | $\underline{0.155}_{\pm 0.07}$ | $0.305_{\pm 0.19}$ |
| CauGramer | $\underline{0.073}_{\pm 0.06}$ | $\mathbf{0.057}_{\pm \mathbf{0.04}}$ | $\mathbf{0.045}_{\pm \mathbf{0.04}}$ | $\mathbf{0.090}_{\pm \mathbf{0.09}}$ | $\mathbf{0.118}_{\pm \mathbf{0.02}}$ | $\mathbf{0.125}_{\pm \mathbf{0.01}}$ |
| **Flickr** | AME | APE | ATE | IME | IPE | ITE |
| Graphormer+Deconf | $\mathbf{0.055}_{\pm 0.03}$ | $0.421_{\pm 0.02}$ | $0.388_{\pm 0.02}$ | $\underline{0.112}_{\pm 0.03}$ | $0.482_{\pm 0.02}$ | $0.465_{\pm 0.01}$ |
| CAL+Deconf | $0.109_{\pm 0.04}$ | $0.340_{\pm 0.05}$ | $0.365_{\pm 0.05}$ | $0.177_{\pm 0.06}$ | $0.388_{\pm 0.05}$ | $0.402_{\pm 0.06}$ |
| GraphGPS+Deconf | $0.138_{\pm 0.14}$ | $0.261_{\pm 0.06}$ | $0.325_{\pm 0.26}$ | $0.199_{\pm 0.19}$ | $0.307_{\pm 0.06}$ | $0.362_{\pm 0.000}$ |
| SAT+Deconf | $0.089_{\pm 0.08}$ | $\underline{0.211}_{\pm 0.05}$ | $\underline{0.178}_{\pm 0.09}$ | $0.168_{\pm 0.13}$ | $\underline{0.259}_{\pm 0.05}$ | $\underline{0.294}_{\pm 0.18}$ |
| CauGramer | $\underline{0.058}_{\pm 0.05}$ | $\mathbf{0.030}_{\pm \mathbf{0.03}}$ | $\mathbf{0.025}_{\pm \mathbf{0.03}}$ | $\mathbf{0.096}_{\pm \mathbf{0.06}}$ | $\mathbf{0.111}_{\pm \mathbf{0.01}}$ | $\mathbf{0.125}_{\pm \mathbf{0.01}}$ |

Table 10: Recent Baselines on Networked Datasets.

| BlogCatalog(BC) | AME | APE | ATE | IME | IPE | ITE |
|---|---|---|---|---|---|---|
| DRLearner | $0.196_{\pm 0.07}$ | $\underline{0.183}_{\pm 0.03}$ | $\underline{0.099}_{\pm 0.07}$ | $0.544_{\pm 0.03}$ | $\underline{0.216}_{\pm 0.03}$ | $0.529_{\pm 0.02}$ |
| GDML | $0.166_{\pm 0.06}$ | $0.371_{\pm 0.06}$ | $0.537_{\pm 0.09}$ | $0.265_{\pm 0.08}$ | $0.411_{\pm 0.06}$ | $0.607_{\pm 0.10}$ |
| GDML w/o FSet | $\mathbf{0.069}_{\pm \mathbf{0.04}}$ | $0.344_{\pm 0.04}$ | $0.412_{\pm 0.04}$ | $\underline{0.152}_{\pm 0.04}$ | $0.384_{\pm 0.04}$ | $0.471_{\pm 0.04}$ |
| UNITE | $\mathbf{0.069}_{\pm \mathbf{0.00}}$ | $0.345_{\pm 0.01}$ | $0.263_{\pm 0.00}$ | $0.194_{\pm 0.00}$ | $0.346_{\pm 0.08}$ | $\underline{0.269}_{\pm 0.01}$ |
| CauGramer | $\underline{0.073}_{\pm 0.06}$ | $\mathbf{0.057}_{\pm \mathbf{0.04}}$ | $\mathbf{0.045}_{\pm \mathbf{0.04}}$ | $\mathbf{0.090}_{\pm \mathbf{0.09}}$ | $\mathbf{0.118}_{\pm \mathbf{0.02}}$ | $\mathbf{0.125}_{\pm \mathbf{0.01}}$ |
| **Flickr** | AME | APE | ATE | IME | IPE | ITE |
| DRLearner | $0.135_{\pm 0.08}$ | $\underline{0.216}_{\pm 0.02}$ | $\underline{0.131}_{\pm 0.07}$ | $0.529_{\pm 0.02}$ | $\underline{0.264}_{\pm 0.02}$ | $0.547_{\pm 0.02}$ |
| GDML | $0.239_{\pm 0.08}$ | $0.381_{\pm 0.02}$ | $0.620_{\pm 0.08}$ | $0.359_{\pm 0.10}$ | $0.448_{\pm 0.02}$ | $0.716_{\pm 0.09}$ |
| GDML w/o FSet | $0.067_{\pm 0.04}$ | $0.363_{\pm 0.07}$ | $0.428_{\pm 0.09}$ | $\underline{0.174}_{\pm 0.03}$ | $0.426_{\pm 0.07}$ | $0.511_{\pm 0.08}$ |
| UNITE | $\mathbf{0.043}_{\pm \mathbf{0.00}}$ | $0.335_{\pm 0.01}$ | $0.297_{\pm 0.00}$ | $0.212_{\pm 0.00}$ | $0.335_{\pm 0.01}$ | $\underline{0.287}_{\pm 0.01}$ |
| CauGramer | $\underline{0.058}_{\pm 0.05}$ | $\mathbf{0.030}_{\pm \mathbf{0.03}}$ | $\mathbf{0.025}_{\pm \mathbf{0.03}}$ | $\mathbf{0.096}_{\pm \mathbf{0.06}}$ | $\mathbf{0.111}_{\pm \mathbf{0.01}}$ | $\mathbf{0.125}_{\pm \mathbf{0.01}}$ |

For the sake of fairness, we add a deconfounder module (+Deconf) to these frameworks for treatment effect estimation. The results in Table 9 demonstrate the superiority of CauGramer, which achieves the best performance across all metrics on both datasets.

## E.7 FURTHER DISCUSSION ON RECENT BASELINES ON NETWORKED DATA

In this subsection, we further discuss the comparisons with four competing non-i.i.d. methods on networked data, including DRLearner (Leung & Loupos, 2022), GDML and GDML w/o FSet (Khatami et al., 2024), and UNITE (Lin et al., 2024). Then we reporte $\epsilon_{\text{AVG}}$ of AME, APE and ATE and $\sqrt{\epsilon_{\text{PEHE}}}$ of IME, IPE and ITE on the BlogCatalog (BC) and Flickr datasets in the Table 10.

From Table 10, we have the following observations: (1) DRLearner performs well for population-level estimates but exhibits significant errors in individual-level estimates. This is because Doubly Robust methods were originally designed to address confounding bias at the population level, rather than to capture fine-grained individual-level heterogeneity. (2) GDML relies on a focal set where all units' neighborhoods are non-overlapping. To construct the focal set, we randomly select a node, remove it along with its second-order neighbors, and repeat this process until no more nodes can be added. However, in densely connected graphs, the focal set contains very few nodes. On the two datasets we used, the focal set for GDML included only about 100 nodes, making it difficult to achieve accurate causal effect estimation. (3) To address this limitation, we provided an additional baseline, GDML w/o FS, which does not use a focal set and instead encompasses the entire dataset. While GDML w/o FS performs well in estimating the average main effect, it falls short on the other five metrics compared to our proposed CauGramer. (4) The UNITE provided by Lin et al. (2024) only supports the estimation of AME and IME. While UNITE shows a slight improvement over our method in terms of the Average Main Effect (AME), its performance significantly drops at the individual level. In contrast, our method consistently excels across all six metrics, demonstrating a more balanced capability in capturing both average and individual-level effects. These results

underscore the strengths of CauGramer, particularly its ability to handle complex individual-level heterogeneity and interference effects while maintaining superior performance across all metrics.

P.S. UNITE (Lin et al., 2024) uses all units features $X$ and treatments $T$ to learn a Graph Structure Learner (GSL). It then employs two learned outcome predictors to predict treated and control outcomes. If the inputs are changed, the learned graph structure and its associated weights will also change, leading to potential additional bias in the fixed outcome predictors without further assumptions. These results show that UNITE struggles with peer effects (APE/IPE) and total effects (ATE/ITE) due to changes in the graph structure and its weights, which likely render the fixed outcome predictors ineffective. Consistent with the original UNITE (Lin et al., 2024) paper, which focuses on average treatment effects and individual treatment effects, we report only AME and IME for UNITE in the main text.

# F    SUPPLEMENTARY RELATED WORK IN NETWORKED INTERFERENCE

**Unmeasured Confounders.** Our work is related to the unmeasured confounders. As shown in Table 1, CNE (Veitch et al., 2019), NetDeconf (Guo et al., 2020), and SPNet (Huang et al., 2023) only consider partial unmeasured confounders embedded in the structural network and unit information, and cannot address the missing key variables, which can influence both treatment and outcomes, within self-features or peer-features. In our CauGramer, we incorporate bridge moment constraints to reduce confounding effects. Moreover, when negative controls are available (Miao et al., 2018; Hu et al., 2023; Shi et al., 2020), CauGramer can extend to address unmeasured confounders by bridges.

**Partial Interference.** The partial interference, characterized by a partition of units into numerous disjoint clusters (e.g., households), with interference confined to units ("neighbors") within the same cluster, was initially explored in the works of Halloran & Struchiner (1995); Sobel (2006); Hudgens & Halloran (2008) under random treatment assignments in two-stage randomized experiments. Following this, a series of works account for nonrandom treatment assignments in observational data under partial interference (Barkley et al., 2020; Liu et al., 2016; Qu et al., 2021). Rather than utilizing the framework employing the $\alpha$-allocation strategy, Qu et al. (2021) define the estimands under the conditional exchangeability framework and derive the semi-parametric efficiency bound. In our CauGramer, we relax the partial interference assumption and study heterogeneous interference effects under an unknown interference graph.

**Exposure Mapping.** This paper is also related to exposure mapping, which is another general way to deal with network interference through experimental designs or treatment assignments (Aronow & Samii, 2017; Tortu et al., 2020). Aronow & Samii (2017) is the first to introduce the exposure mapping framework, pioneering the estimation of average unit-level causal effects from *randomized experiments* under networked interference. The exposure mapping relates treatments assigned to exposures received based on inverse probability weighting (Aronow & Samii, 2017). Unlike traditional methods relying on randomized trials (Aronow & Samii, 2017; Owusu, 2023; Yuan & Altenburger, 2023), Tortu et al. (2020) extend the exposure mapping framework about causal inference under network interference *in observational studies*, allowing for multi-valued treatments and weighted interference networks. While Tortu et al. (2020) proposes a parametric estimation approach, our CauGramer automatically model unknown interference effects.

**Unknown Network Structure.** The most related work to our paper are those by Yu et al. (2022); Cortez et al. (2022). Under unknown network structure, Yu et al. (2022) propose a simple estimator and an efficient randomized design for total treatment effect estimation under a heterogeneous linear interference model (Eckles et al., 2017). Building on this, Cortez et al. (2022) generalize their findings to polynomial models. In this paper, we propose a CauGramer algorithm to estimate the main effect, peer effect, and total effect of treatments *in observation networked data* under an unknown network.

