# OpenReview forum: "Causal Graph Transformer for Treatment Effect Estimation Under Unknown Interference"
_ICLR.cc/2025/Conference — ICLR 2025 Poster_

### Official Review · Reviewer_ii3V · 2024-10-18

**Soundness:** 2
**Presentation:** 3
**Contribution:** 2
**Rating:** 6
**Confidence:** 4

**Summary:**

This study aims to estimate treatment effects in the presence of unknown interference (including unknown network and summary mechanism). To achieve this goal, it proposes a method named CauGramer, which uses the transformer to address unknown interference.
However, some important studies and baseline methods for unknown interference are ignored.

**Strengths:**

1. The presentation is clear.
2. A new method is proposed.
3. Identifiability of total effect is given, which provides insight for studying unknown interference.
4. Experiments are conducted to verify the proposed method.

**Weaknesses:**

# Weakness
1. Some important studies are ignored. Unknown interference has been addressed in many existing works [R1-R4]. These works have excellent contributions for unknown interference, but they are ignored by authors. Authors should introduce the difference between their work and these works. Specifically, the authors add a paragraph in the related work section comparing their approach to [R1-R4], highlighting key differences in methodology, assumptions, and capabilities.

2. As introduced in Weaknesses 1, unknown interference is not a new challenge in causal inference, because there are some existing methods [R1~R4] that focus on this issue.
The novelty of this study might be limited. It might be helpful if the authors could introduce the limitations of these existing methods and how this study addresses them. Specifically, what limitations of prior methods does CauGramer address? What new capabilities does it enable?

3. In addition, existing methods [R1-R4] have a more relaxed assumption for unknown interference. They do not require any network information, whereas the proposed method requires network information, i.e., $A$.

4. For experiments, authors need to compare with the existing methods for unknown interference. In the current version, authors did not compare with [R1-R4]. The authors can add [R1], [R3] and [R4] as baselines in their experiments, following the implementation details.  [R1] can create a network (you can only take the edges among different nodes) when the input graph is a complete graph or an initial graph, i.e., $A$ in the authors' setting. Then, you can apply G-HSIC on the network created by [R1] as a baseline. [R3] and [R4] can directly work when you give or not give an input graph. [R2] can be ignored, as it is not a learning method.

5. In definitions 1 and 3, as well as equations (1) and (2), should the $t$ in the right of '=' be 1, as the treatment is a binary value, i.e., 1 or 0.





# Reference
[R1] Bhattacharya, et al.  Causal inference under interference and network uncertainty.  UAI 2019.

[R2] Sävje, et al. Average treatment effects in the presence of unknown interference. The Annals of Statistics 2021.

[R3] Mayleen, et al. Staggered rollout designs enable causal inference under interference without network knowledge. NeurIPS 2022.

[R4] Lin, et al. Treatment effect estimation under unknown interference. PAKDD 2024.

**Questions:**

# Questions:
1. Why do you not introduce the existing works [R1-R4], which have excellent contributions for unknown interference? Can you introduce the difference between your works and these works?  What limitations of prior methods does CauGramer address? What new capabilities does your method enable?

2. [R1-R4] can work when the network information is totally unknown. Can the proposed method work when the network information is totally unknown? How does the performance change in this case?

3. In definitions 1 and 3, as well as equations (1) and (2), should the $t$ in the right of '=' be 1 instead of $t$?

4. For more details, please see Weakness.



# Reference
[R1] Bhattacharya, et al.  Causal inference under interference and network uncertainty.  UAI 2019

[R2] Sävje, et al. Average treatment effects in the presence of unknown interference. The Annals of Statistics 2021

[R3] Mayleen, et al. Staggered rollout designs enable causal inference under interference without network knowledge. NeurIPS 2022.

[R4] Lin, et al. Treatment effect estimation under unknown interference. PAKDD 2024.

---

> ### Author Response · Authors · 2024-11-21
> **Responses by Authors (Part 1)**
>
> **Dear Reviewer #ii3V,**
>
> We sincerely appreciate your constructive comments to improve our paper. You may find our corresponding explanations below for your concerns. We have uploaded the revised paper **(with track changes marked in blue)**. We would really appreciate it if you could let us know if you have any further concerns or suggestions.
>
> **In the revised manuscript, we include discussions on four related works (listed in Appendix C):**
>
> - **Bhattacharya et al., (2020)**: Under interference and network uncertainty, this paper proposes methods that integrate structure learning and causal inference techniques to estimate causal effects. Their algorithms rely on structure learning to first learn graphical models and then estimate the Population Average Overall Effect (PAOE), a variant of the Average Total Effect. They address network uncertainty under partial interference through structure learning and then use the auto-g-computation algorithm to estimate the PAOE. **This method rely on the assumption that the true data-generating process corresponds perfectly (satisfying Markov and faithfulness conditions) to some unknown chain graph, with two restrictions.** This paper does not release its code.
> - **Savje et al., (2021)**: This paper study average treatment effects estimation in randomized experiments with unknown and arbitrary interference among units. The authors propose the Expected Average Treatment Effect (EATE) as a generalized estimand that incorporates potential spillover effects by marginalizing over all possible treatment assignments. While their study provides a theoretical and experimental analysis using existing methods, it does not introduce new algorithms to address unknown interference. Furthermore, **their analysis is based on randomized controlled trials (RCTs). In contrast, our paper focuses on addressing the challenge of unknown interference on observational data.**
> - **Cortez et al. (2022)**: Under unknown network structures, Cortez et al. (2022) extend the work of Yu et al. (2022) by generalizing causal inference methods from linear to polynomial models, enabling the estimation of total treatment effects (TTE) under more complex and non-linear network interactions. They introduce a staggered rollout experimental design, which incrementally applies treatments and collects observations, eliminating the need for prior knowledge of the network structure. **However, a limitation of their approach is its reliance on multiple randomized trials (i.e., the staggered rollout design) to gather observations at different stages.**
> - **Lin et al., (2024)**: Under unknown interference, the UNITE algorithm uses a **Graph Structure Learner (GSL)** to infer the hidden interference structure by constructing a complete graph and imposing L0-norm regularization to identify significant connections. With the inferred structure, **Graph Convolutional Networks (GCNs)** are employed to learn an **aggregation function** and model the interference from neighboring units, effectively capturing the interference patterns and enabling accurate treatment effect estimation.
>
> (Bhattacharya et al., 2020) Rohit Bhattacharya, Daniel Malinsky, and Ilya Shpitser. Causal inference under interference and
> network uncertainty. In Uncertainty in Artificial Intelligence, pp. 1028–1038. PMLR, 2020.
>
> (Savje et al., 2021) Fredrik Savje, Peter Aronow, and Michael Hudgens. Average treatment effects in the presence of
> unknown interference. Annals of statistics, 49(2):673, 2021.
>
> (Cortez et al., 2022) Mayleen Cortez, Matthew Eichhorn, and Christina Yu. Staggered rollout designs enable causal inference under interference without network knowledge. Advances in Neural Information Processing Systems, 35:7437–7449, 2022.
>
> (Lin et al., 2024) Xiaofeng Lin, Guoxi Zhang, Xiaotian Lu, and Hisashi Kashima. Treatment effect estimation under unknown interference. In Pacific-Asia Conference on Knowledge Discovery and Data Mining, pp. 28–42. Springer, 2024.

---

> > ### Author Response · Authors · 2024-11-21
> > **Responses by Authors (Part 2)**
> >
> > > **[W1] Some important studies are ignored. Unknown interference has been addressed in many existing works [R1-R4]. These works have excellent contributions for unknown interference, but they are ignored by authors. Authors should introduce the difference between their work and these works. Specifically, the authors add a paragraph in the related work section comparing their approach to [R1-R4], highlighting key differences in methodology, assumptions, and capabilities.**
> >
> > **Response:** We sincerely thank the reviewers for providing four related work on non-iid data. We add a paragraph in the related work section comparing our approach to [R1-R4]:
> >
> > **Bhattacharya et al. (2020)** rely on structure learning under strict assumptions of Markov and faithfulness conditions, and the estimation of the causal effect is done by the auto-g-computation algorithm. They address causal inference under network uncertainty through structure learning and auto-g-computation algorithm. **Savje et al., (2021)** study average treatment effects estimation in randomized experiments with unknown and arbitrary interference among units by introducing EATE, but their analysis is limited to experimental data and does not propose new algorithms for unknown interference. **Mayleen et al. (2022)** extend the work of Yu et al. (2022) by generalizing causal inference methods from linear to polynomial models, and employ staggered rollout designs to estimate total treatment effects under unknown network structures. The most relevant work is **Lin et al., (2024)**, which addresses unknown interference graphs and learnable aggregation functions through their framework, UNITE, using L0-norm regularization and GCNs for accurate treatment effect estimation.
> >
> > > **[W2] As introduced in Weaknesses 1, unknown interference is not a new challenge in causal inference, because there are some existing methods [R1~R4] that focus on this issue. The novelty of this study might be limited. It might be helpful if the authors could introduce the limitations of these existing methods and how this study addresses them. Specifically, what limitations of prior methods does CauGramer address? What new capabilities does it enable?**
> >
> > **Response:** Thanks for your advice. Among these works, **Savje et al. (2021)** and **Hoshino et al. (2022)** rely on randomized controlled trials (RCTs). **Bhattacharya et al. (2020)** assumes that the true data-generating process perfectly corresponds (satisfying Markov and faithfulness conditions) to some unknown chain graph with two restrictions. However, in practical scenarios, structure learning algorithms often face **implementation issues** and produce **unreliable results** due to the size of the graph and unmet assumptions. Although **Lin et al. (2024)** proposes the UNITE framework, which uses L0-norm regularization and GCNs for accurate treatment effect estimation, it only supports the estimation of **AME** and **IME**. Besides, they does not consider that **peer treatments can influence the treatment choices of others in real applications**.
> >
> > Unlike these works, **our CauGramer** studies causal inference problems under unknown interference in **observational data**, learning interference representations and infers causal effects without relying on structural assumptions. By integrating **confounder balancing** and **minimax moment constraints**, CauGramer fully incorporates peer information, enabling robust treatment effect estimation. Furthermore, to address the challenge of unmeasured confounders, we extend the CauGramer algorithms with **negative controls**, introducing **CauGramer(NC)** to effectively handle unmeasured confounding issues.

---

> ### Author Response · Authors · 2024-11-21
> **Responses by Authors (Part 3)**
>
> **[W3] In addition, existing methods [R1-R4] have a more relaxed assumption for unknown interference. They do not require any network information, whereas the proposed method requires network information, i.e., A. [R1-R4] can work when the network information is totally unknown. Can the proposed method work when the network information is totally unknown? How does the performance change in this case?**
>
> **Response:** Thanks for your comment. Our CauGramer can work when the network information is totally unknown. In such cases, **CauGramer can use any known connected graph (even one that is randomly generated) as a substitute for the social network to aggregate interference information**. As demonstrated in Figure 3, CauGramer remains robust even when the provided network is entirely unrelated to the 'Random' interference graph. It still learns causal effects effectively and outperforms existing methods, suggesting that any connected graph—even one that is randomly generated and independent of the interference graph—can help aggregate interference information. **Exploring how to construct an efficient connected graph would be an interesting directions, but it is beyond the scope of this paper and is left for future work. In Appendix B, we added a new section 'Limitations and Future Directions' to further clarify this.**
>
> Although [R1-R4] can work when the network information is totally unknown, **Savje et al. (2021)** and **Hoshino et al. (2022)** rely on randomized controlled trials (RCTs). **Lin et al. (2024)** supports only the estimation of **AME** and **IME**, limiting its applicability. **Bhattacharya et al. (2020)** assume that the true data-generating process perfectly aligns with (satisfying Markov and faithfulness conditions) an unknown chain graph under two strict restrictions. However, in practical scenarios, structure learning algorithms often encounter **implementation challenges** and yield **unreliable results** due to the graph’s size and unmet assumptions.
>
> In observational network data, we focus on real-world scenarios where individuals closely connected in a social network are more likely to interfere with each other. Based on this observation, we proposed the **CauGramer** algorithm to infer unknown interference information using a known social network. **Social networks are typically easier to obtain than true interference graphs, as they can often be derived from sources such as co-authorship networks, public community relationships, organizational affiliations, or online interactions like comments and messages. As a result, most existing studies also tend to rely on social networks for research purposes.**

---

> ### Author Response · Authors · 2024-11-21
> **Responses by Authors (Part 4)**
>
> > **[W4] For experiments, authors need to compare with the existing methods for unknown interference. In the current version, authors did not compare with [R1-R4]. The authors can add [R1], [R3] and [R4] as baselines in their experiments, following the implementation details. [R1] can create a network (you can only take the edges among different nodes) when the input graph is a complete graph or an initial graph, i.e., A in the authors' setting. Then, you can apply G-HSIC on the network created by [R1] as a baseline. [R3] and [R4] can directly work when you give or not give an input graph. [R2] can be ignored, as it is not a learning method.**
>
> **Response:** We appreciate the reviewer’s suggestion to incorporate comparisons with the existing methods for unknown interference. However, **Savje et al. (2021)** and **Hoshino et al. (2022)** rely on randomized controlled trials (RCTs), and thus their approaches are not applicable to our observational network data. While **Bhattacharya et al. (2020)** proposes constructing a network from a complete or initial graph, the lack of released code poses challenges for reproducibility. Furthermore, structure learning algorithms often face **implementation challenges** and produce **unreliable results** due to the large graph sizes and unmet assumptions in real-world data. The most relevant work is **[Lin et al., 2024]**, which addresses **unknown interference graphs** and **learnable aggregation functions** through their framework, **UNITE**, using L0-norm regularization and GCNs for accurate treatment effect estimation. **We include comparisons with UNITE in our experiments. We have added the comparison experiments in Table 3 in main text and in Table 10 in Appendix E.7 (with track changes marked in blue)**.
>
> | BC | **AME** | **APE** | **ATE** | **IME** | **IPE** | **ITE** |
> | - | - | - | - | - | - | - |
> | UNITE | **0.069 ± 0.00** | - | - | 0.194 ± 0.00 | - | - |
> | CauGramer | 0.073 ± 0.06 | **0.057 ± 0.04** | **0.045 ± 0.04** | **0.090 ± 0.09** | **0.118 ± 0.02** | **0.125 ± 0.01** |
>
> | Flickr | **AME** | **APE** | **ATE** | **IME** | **IPE** | **ITE** |
> | - | - | - | - | - | - | - |
> | UNITE | **0.043 ± 0.00** | - | - | 0.212 ± 0.00 | - | - |
> | CauGramer | 0.058 ± 0.05 | **0.030 ± 0.03** | **0.025 ± 0.03** | **0.096 ± 0.06** | **0.111 ± 0.01** | **0.125 ± 0.01** |
>
> We use the original implementation of **UNITE** provided by **[Lin et al., 2024]** and observe that it only supports the estimation of **AME** and **IME**. While UNITE demonstrates a slight improvement over our method in terms of the **Average Main Effect (AME)**, its performance drops significantly at the individual level. In contrast, our method consistently performs strongly across all six metrics, highlighting its comprehensive and balanced ability to handle both average and individual-level effects effectively.
>
> > **[W5] In definitions 1 and 3, as well as equations (1) and (2), should the t in the right of '=' be 1, as the treatment is a binary value, i.e., 1 or 0.**
>
> **Response:** Thank you for pointing out these typos. We will revise them accordingly.
>
> ----
>
> **Thanks for your detailed advice and comments. We welcome any further technical advice or questions on this work and we will do our best to address your concerns.**

---

> > ### Comment · Reviewer_ii3V · 2024-11-25
> >
> > Thank you for your responses. W3 and 5 are addressed. However, others are not fully addressed. I will keep my score due to the following reasons:
> >
> > 1. Authors acknowledge that many important studies that focus on unknown interference have been ignored.  It would be understandable if only one or two recent works on unknown interference were left out, but all studies on unknown interference were ignored in the first submission. Why did this happen?
> > 2. As unknown interference is not a new issue, and is addressed by [R1 ~ 4]. The technological contribution is limited, as the proposed method only combines many existing methods.
> > 3.  The baseline is still incomplete. [R1] is not compared in the current experiments.
> > 4. In Table 3 of the revision, there are some typos. Please check them carefully.

---

> ### Author Response · Authors · 2024-11-25
> **It is not accurate to state that “all studies on unknown interference were ignored in the first submission” and “the proposed method only combines many existing methods”**
>
> **Dear Reviewer #ii3V,**
>
> Thank you for your further questions. In your initial comments under weaknesses, four points focused on our lack of discussion and comparison with [R1–R4], while one for typos. However, **[R3] had already been included in Appendix D of our original manuscript.** In response to your suggestions, we have expanded our discussion of these four works in Appendix C, which has significantly improved our paper. We are sincerely grateful for bringing [R1, R2, R4] to our attention.
>
> **[It is not accurate to state that “all studies on unknown interference were ignored in the first submission”]**  There may be a misunderstanding. This paper is related to seven domains. In our original submission, due to the limited space, we only listed representative algorithms (which served as comparison baselines in the Experiments section) in the Related Work section, and **deferred discussions on “Partial Interference”, “Exposure Mapping”, and “Unknown Network Structure” to Appendix D, titled “Supplementary Related Work in Networked Interference” in original manuscript.** In Unknown Network Structure, we discussed contributions by Eckles et al. (2017), Yu et al. (2022), and Cortez et al. (2022), with **Cortez et al. (2022) corresponding to your mentioned [R3].**
>
> We appreciate your insightful comment, which highlighted the need for better organization in our Related Work Section. In response, we have restructured this section and expanded our discussion of the four related works in Appendix C. **While we acknowledge that some studies on unknown interference were overlooked, it is not accurate to state that “all studies on unknown interference were ignored in the first submission”.**
>
> **[Although unknown interference is not a new issue, it has not been well-addressed by [R1–R4]. As a response to your [W2]**:
>
> - **[R1] assumes the true data-generating process aligns perfectly with an unknown chain graph (satisfying Markov and faithfulness conditions), but this approach comes with two restrictions.** In practical scenarios, structure learning algorithms often face challenges such as implementation issues and unreliable results due to the graph’s size and unmet assumptions.
> - **[R2] and [R3] rely on randomized controlled trials (RCTs).**
> - **[R4] introduces the UNITE framework,** which employs L0-norm regularization and GCNs for treatment effect estimation but only supports the estimation of AME and IME.
>
> We hope this clarifies our position in related work.
>
> **[It is not accurate to claim that “the proposed method only combines many existing methods”].** We would like to emphasize the following:
>
> 1. Our paper proposes a novel **Causal Graph Transformer (CauGramer) with cross-attention.** Unlike existing methods that use GNNs solely for information aggregation and feed learned embeddings into a transformer, our approach fundamentally differs in structure. **In CauGramer, the Q, K, and V modules have distinct receptive fields, and the input is no longer limited to self-information, combining the strengths of GNNs in a new architecture.**
> 2. **Our CauGramer framework relaxes the assumption that peers’ treatments are independent, allowing peer treatments to influence others’ treatments.** In this setting, the balanced representations employed by traditional methods are limited to identifying the main effects of self-treatment and cannot capture peer effects. To address this limitation, we propose a novel transformation formula (lines 345–360) to address this challenge.
> 3. We construct conditional moment constraints to ensure residuals approach zero given $x$ and $t$, thereby mitigating model misspecification issue. In the presence of unmeasured confounders, we reformulate Eq. (11) as Eq. (16) by leveraging a negative control variable to identify causal effects. While Eq. (16) is not our novel contribution, **Eq. (11) is proposed by us and is not derived from Shi et al. (2020) or Kallus et al. (2021).**
> 4. In our work, we provide a **causal identification theorem (Theorem 1)** and an **interference proposition (Proposition 1)**, which guide the model in capturing interference information and learning balanced representations for causal effect estimation. Furthermore, in Appendix D.4, **we include Theorem 2, which utilizes negative controls to identify causal effects in the presence of unmeasured confounders**. We hope these contributions establish a theoretical foundation for our method.
>
> **[Regarding the comparison baselines [R1]]:**
>
> 1. **The assumptions required for structure learning in [R1] do not hold in our setting.**
> 2. **The G-HSIC, mentioned in your [W4], is already included as one of our baselines.**
> 3. **[R1], preprint in 2019 and published in UAI 2020, is no longer the state-of-the-art.**
> 4. **[R1] did not release code, which poses challenges for reproducibility.**
>
> Therefore, [R1] was not included in our experiments.
>
> Thank you so much for your time and efforts.
>
> **Authors of Paper #2331**

---

> > ### Comment · Reviewer_ii3V · 2024-11-26
> >
> > Thank you for your further explanation. I now understand your technical contribution. I rechecked related works and your new and original version of the paper. I summarized my new comments as follows:
> >
> > 1. I rechecked your original manuscript. In the original version, you only mentioned existing studies (including [R3]) for unknown interference in appendices instead of introduction/related work.  This might confuse readers.  Since you aim to address unknown interference, these studies for unknown interference should be introduced in the main content of the paper instead of appendices, which are usually ignored by readers. In addition, why can [R4] only be used for AME and IME? It seems the differences between IME and ITE/IPE are only the inputs of the network that predict Y^1 and Y^0. Changing inputs of the existing method is not a challenge.
> >
> > 2. In Table 3 of the new version,  there are some typos in the table of results on the Flickr dataset. Please check the corresponding methods and results carefully (two and column).
> >
> > I think your contribution relies on a new idea of identifying causal effects under unknown interference. This is different from proof of [R4], which seems to need an assumption that a network can be discovered from given information. You design a new framework including Eq.(11) to estimate various causal effects. [R4] uses a L_0 regularization, which might make the model sensitive to hyperparameters. Compared with [R4], your method seems to achieve better performance and robustness.
> >  Based on these contributions, I increased my score.

---

> ### Author Response · Authors · 2024-11-26
> **Thank you for raising your score!**
>
> **Dear Reviewer #ii3V,**
>
> We sincerely thank you for your valuable suggestions, which have significantly improved our paper, especially the **Related Work Section**. We apologize for the confusion regarding the related work in our original manuscript. **Due to limited space, in our first submission, existing studies (including [R3]) on unknown interference were previously placed in the Appendix. Based on your comments, we have reorganized the Related Work Section to discuss these studies in the main text. Additionally, we also provided detailed descriptions of these works in Appendix C**. We deeply appreciate your insightful feedback and have incorporated these changes and revised typos in our revised manuscript.
>
> > **[Question] Why can [R4] only be used for AME and IME? It seems the differences between IME and ITE/IPE are only the inputs of the network that predict $Y^1$ and $Y^0$. Changing inputs of the existing method is not a challenge.**
>
> **Response:** Thanks for your question. While changing inputs of the existing method is not a challenge, UNITE [R4] uses all unit features $X$ and treatments $T$ to learn a Graph Structure Learner (GSL). It then employs two learned outcome predictors to predict treated and control outcomes. **If the inputs are changed, the learned graph structure and its associated weights may also change, leading to potential additional bias in the fixed outcome predictors without further assumptions. This limitation can result in unreliable results**. Therefore, in the original **UNITE [R4]** paper, the authors focus on average treatment effects and individual treatment effects, but do not conduct experiments or provide discussions on peer effects.
>
> To address your concern, we conducted additional experiments by modifying the inputs of UNITE to estimate various causal effects (AME, APE, ATE, IME, IPE, ITE). The results are as follows:
>
> | BC | **AME** | **APE** | **ATE** | **IME** | **IPE** | **ITE** |
> | - | - | - | - | - | - | - |
> | UNITE | **0.069 ± 0.00** | 0.345 ± 0.01 | 0.263 ± 0.00 | 0.194 ± 0.00 | 0.346 ± 0.08 | 0.269 ± 0.01 |
> | CauGramer | 0.073 ± 0.06 | **0.057 ± 0.04** | **0.045 ± 0.04** | **0.090 ± 0.09** | **0.118 ± 0.02** | **0.125 ± 0.01** |
>
> | Flickr | **AME** | **APE** | **ATE** | **IME** | **IPE** | **ITE** |
> | - | - | - | - | - | - | - |
> | UNITE | **0.043 ± 0.00** | 0.335 ± 0.01 | 0.297 ± 0.00 | 0.212 ± 0.00 | 0.335 ± 0.01 | 0.287 ± 0.01 |
> | CauGramer | 0.058 ± 0.05 | **0.030 ± 0.03** | **0.025 ± 0.03** | **0.096 ± 0.06** | **0.111 ± 0.01** | **0.125 ± 0.01** |
>
> These results show that **UNITE struggles with peer effects (APE/IPE) and total effects (ATE/ITE) due to changes in the graph structure and its weights, which likely render the fixed outcome predictors ineffective**. Consistent with the original **UNITE [R4] paper, which focuses on average treatment effects and individual treatment effects,** we report only AME and IME for UNITE in the main text. **We include the above discussion about the experiments on UNITE in Appendix E.7.**
>
> **We deeply appreciate your time and effort in reviewing our paper and your constructive feedback.**
>
> **Authors of Paper #2331**

---

### Official Review · Reviewer_qEMS · 2024-10-30

**Soundness:** 2
**Presentation:** 3
**Contribution:** 3
**Rating:** 6
**Confidence:** 4

**Summary:**

The paper "Causal Graph Transformer for Treatment Effect Estimation under Unknown Interference" introduces an innovative Interference-Agnostic Causal Graph Transformer (CauGramer) framework for causal effect estimation under unknown network interference. This approach captures complex interference information across nodes through an L-order graph attention mechanism combined with cross-attention, without relying on a known interference graph. The model incorporates confounder balancing and minimax moment constraints to achieve robust causal effect estimation.

**Strengths:**

1.The study introduces a novel approach by addressing causal inference from the perspective of an unknown interference graph. The combination of L-order graph attention with cross-attention for modeling interference information.
2.The paper is well-structured.
3.Causal inference under unknown interference graphs is a significant challenge in network data analysis, and the outcomes of this study provide a reference for future research.

**Weaknesses:**

1.In Table 1, which includes i.i.d. data, the title "Causal Networked Data" may not be fully accurate. Perhaps "Observational Data" would be more accurate.
2.On line 129, "Representation" could be more accurate formatted in lowercase as "representation."
3.The method appears to rely on an unconfoundedness assumption, yet latent confounders are mentioned later, violating this assumption.
4.In Section 4.1, Ma & Tresp’s work is not necessarily limited to first-order interference.
5.There are multiple instances of quotation mark formatting errors in lines 375, 449, 487, and 516, among others.
6.How did you address latent confounders in their approach?
7.When network interference is unknown, how does the proposed method differ from existing work (e.g., Rf1-Rf3)?
8.The work lacks a description of the number of layers (L) and attention heads (M) used.
9.While the paper compares CauGramer with several mainstream methods, it lacks a detailed discussion of specific methods for unknown interference graphs (e.g., Rf1-Rf3).
Rf1: Sävje, Fredrik, Peter Aronow, and Michael Hudgens. "Average treatment effects in the presence of unknown interference." Annals of Statistics 49.2 (2021): 673.
Rf2: Hoshino, Tadao, and Takahide Yanagi. "Causal inference with noncompliance and unknown interference." Journal of the American Statistical Association (2023): 1-12.
Rf3: Lin, Xiaofeng, et al. "Treatment Effect Estimation Under Unknown Interference." Pacific-Asia Conference on Knowledge Discovery and Data Mining. Springer Nature Singapore, 2024.

**Questions:**

1.In the COVID case study, where transmission primarily occurs through contact, how can interference occur in the absence of a direct link? Could you clarify potential mechanisms of transmission under such conditions?
2.Could you clarify why the final output \( r_x^{(L)} \) is used as the input to the first layer of the peer-treatment representation network, specifically \( r_t^{(1)} = r_x^{(L)} \)?
3. Please explain the statement: "we simulate unmeasured confounding by removing features."
4. The paper states that the relationship between peer-treatment \( t_{Pi} \) and outcome \( y_i \) remains confounded by the common causes \( r_i \), leading to biased estimates of peer and total effects. Moreover, the peer-treatment \( t_{Pi} \) is unknown. In the main text, it is mentioned that \( r = r_x \oplus r_t \). Could you clarify why \( r_i \) acts as a confounder for both \( t_{Pi} \) and \( y_i \)?

---

> ### Author Response · Authors · 2024-11-21
> **Responses by Authors (Part 1)**
>
> **Dear Reviewer #qEMS,**
>
> We sincerely appreciate your constructive comments to improve our paper. You may find our corresponding explanations below for your concerns. We have uploaded the revised paper **(with track changes marked in blue)**, and we would greatly appreciate it if you could let us know of any further concerns.
>
> > **[W1] Typos:**
> >
> > **In Table 1, which includes i.i.d. data, the title "Causal Networked Data" may not be fully accurate. Perhaps "Observational Data" would be more accurate.**
> >
> > **On line 129, "Representation" could be more accurate formatted in lowercase as "representation."**
> >
> > **In Section 4.1, Ma & Tresp’s work is not necessarily limited to first-order interference.**
> >
> > **There are multiple instances of quotation mark formatting errors in lines 375, 449, 487, and 516, among others.**
>
> **Response:** Thank you for your thoughtful suggestions. Regarding Table 1, we have revised the title to **“Characteristics of Representative Algorithms on Observational Networked Data”** and removed the i.i.d. methods, **CFRNet** and the **Doubly Robust Estimator**. Additionally, we have incorporated more non-i.i.d. methods suggested by the reviewers, including **Savje et al., (2021)**, **Hoshino et al., (2023)**, **Uncertainty (Bhattacharya et al., 2020)**, and **UNITE (Lin et al., 2024)**. For the other issues, we have revised the related statements and corrected all quotation mark formatting errors throughout the manuscript.
>
> > **[W2] The method appears to rely on an unconfoundedness assumption, yet latent confounders are mentioned later, violating this assumption. How did you address latent confounders in their approach?**
>
> **Response:** Thank you for pointing this out. In **Appendix D.4**, we introduce **CauGramer(NC)**, an extension of **CauGramer** that addresses latent confounders using negative control variables (NCs). As illustrated in the causal diagram of NCs (Figure 5), **negative control exposures (NCEs)** $Z$ are proxy variables linked to the treatment $T$ through the same unmeasured confounders $U$, while **negative control outcomes (NCOs)** $W$ are influenced by the same unmeasured confounders $U$ as the outcome $Y$. The bridge functions in **CauGramer(NC)** are designed to learn these transformation links, effectively **eliminating the confounding effects of unmeasured confounders** and enabling accurate causal identification.
>
> > **[W3] The work lacks a description of the number of layers (L) and attention heads (M) used.**
>
> **Response:** The choice of the L-order neighborhood parameter would impacts the CauGramer's performance. **A small $L$ value can limit the model’s ability to capture complex relationships between long-distance nodes, reducing its representation power. Conversely, a large L value may introduce excessive noise and irrelevant information, leading to increased computational complexity and a higher risk of overfitting**.We add an analysis of the hyper-parameters $\{L, M\}$ in Table 5 in Appendix E.2.
>
> **Table: Hyper-Parameter Optimization on Flickr Dataset: CauGramer with $L$-layer and $M$-heads.**
>
> | **Flickr** | **AME** | **APE** | **ATE** | **IME** | **IPE** | **ITE** |
> | - | - | - | - | - | - | - |
> | L=1, M=3 | $0.039_{\pm 0.02}$ | $0.320_{\pm 0.10}$ | $0.290_{\pm 0.07}$ | $0.066_{\pm 0.02}$ | $0.358_{\pm 0.09}$ | $0.339_{\pm 0.07}$ |
> | L=2, M=2 | $0.093_{\pm 0.05}$ | $0.230_{\pm 0.11}$ | $0.203_{\pm 0.15}$ | $0.137_{\pm 0.07}$ | $0.282_{\pm 0.09}$ | $0.281_{\pm 0.12}$ |
> | L=2, M=3 | $0.074_{\pm 0.05}$ | $0.223_{\pm 0.06}$ | $0.203_{\pm 0.07}$ | $0.114_{\pm 0.07}$ | $0.222_{\pm 0.06}$ | $0.270_{\pm 0.08}$ |
> | L=2, M=4 | $0.048_{\pm 0.05}$ | $0.260_{\pm 0.09}$ | $0.228_{\pm 0.08}$ | $0.083_{\pm 0.07}$ | $0.305_{\pm 0.08}$ | $0.288_{\pm 0.06}$ |
> | L=3, M=3 | $0.112_{\pm 0.07}$ | $0.437_{\pm 0.08}$ | $0.458_{\pm 0.09}$ | $0.166_{\pm 0.09}$ | $0.467_{\pm 0.07}$ | $0.488_{\pm 0.09}$ |
>
> In this paper, we propose a $L$-layers (default: 2) $M$-heads (default: 3) cross-attention GCN to learn the representation $\boldsymbol{R}_x=g_x(\boldsymbol{x}, \boldsymbol{A})$. Then we propose a $L$-layers (default: $L=1$ for main results and $L=2$ for unknown interference graph in Figure 3) $M$-heads (default: 3) cross-attention GCN to learn the representation $\boldsymbol{R}_t=g_t(\boldsymbol{t}, \boldsymbol{A})$. In real applications, we can use a grid search on the validation dataset to choose the optimal $L$ value and $M$ value.

---

> ### Author Response · Authors · 2024-11-21
> **Responses by Authors (Part 2)**
>
> > **[W4] When network interference is unknown, how does the proposed method differ from existing work (e.g., Rf1-Rf3)? While the paper compares CauGramer with several mainstream methods, it lacks a detailed discussion of specific methods for unknown interference graphs (e.g., Rf1-Rf3).**
> >
> > **(Savje et al., 2021)**: Savje, Fredrik, Peter Aronow, and Michael Hudgens. "Average treatment effects in the presence of unknown interference." Annals of Statistics 49.2 (2021): 673.
> >
> > **(Hoshino et al., 2023)**: Hoshino, Tadao, and Takahide Yanagi. "Causal inference with noncompliance and unknown interference." Journal of the American Statistical Association (2023): 1-12.
> >
> > **(Lin et al., 2024)**: Lin, Xiaofeng, et al. "Treatment Effect Estimation Under Unknown Interference." Pacific-Asia Conference on Knowledge Discovery and Data Mining. Springer Nature Singapore, 2024.
>
> **Response:** We sincerely thank the reviewers for highlighting three related works on non-i.i.d. data. **In the revised manuscript, we have included discussions on these works (listed in Appendix C):**
>
> - **Savje et al., (2021)**: This paper study average treatment effects estimation in randomized experiments with unknown and arbitrary interference among units. The authors propose the Expected Average Treatment Effect (EATE) as a generalized estimand that incorporates potential spillover effects by marginalizing over all possible treatment assignments. While their study provides a theoretical and experimental analysis using existing methods, it does not introduce new algorithms to address unknown interference. Furthermore, **their analysis is based on randomized controlled trials (RCTs). In contrast, our paper focuses on addressing the challenge of unknown interference on observational data.**
> - **Hoshino et al., (2023)**: This paper studies causal inference models in which individuals interact within a social network and may not comply with the assigned treatments. It potentially assumes that the interference arises through the social network, where the structure of interference may depend on local or approximate neighborhood interactions. Based on this, the paper introduces Instrumental Exposure Mapping (IEM), a low-dimensional representation of potentially complex spillover effects, to estimate Intention-to-Treat (ITT) and Local Average Treatment Effects (LATE) for compliers, accounting for both direct and indirect effects while explicitly handling the challenges of noncompliance and unknown interference. However, **this paper assumes that the interference graph is a subgraph of the social network and relies on the random assignment of instrumental variables (IVs), as well as the independence of IVs across individuals**, to ensure valid causal inference.
> - **Lin et al., (2024)**: Under unknown interference, the UNITE algorithm uses a **Graph Structure Learner (GSL)** to infer the hidden interference structure by constructing a complete graph and imposing L0-norm regularization to identify significant connections. With the inferred structure, **Graph Convolutional Networks (GCNs)** are employed to learn an **aggregation function** and model the interference from neighboring units, effectively capturing the interference patterns and enabling accurate treatment effect estimation.
>
> **[Savje et al., 2021]** and **[Hoshino et al., 2023]** rely on randomized instrumental variables (IVs) or experiments, which are infeasible in our observational data setting. The most relevant work is **[Lin et al., 2024]**, which addresses **unknown interference graphs** and **learnable aggregation functions** through their framework, **UNITE**, using L0-norm regularization and GCNs for accurate treatment effect estimation. **We include comparisons with UNITE in our experiments. We have added the comparison experiments in Table 3 in main text and in Table 10 in Appendix E.7 (with track changes marked in blue)**.
>
> | BC | **AME** | **APE** | **ATE** | **IME** | **IPE** | **ITE** |
> | - | - | - | - | - | - | - |
> | UNITE | **0.069 ± 0.00** | - | - | 0.194 ± 0.00 | - | - |
> | CauGramer | 0.073 ± 0.06 | **0.057 ± 0.04** | **0.045 ± 0.04** | **0.090 ± 0.09** | **0.118 ± 0.02** | **0.125 ± 0.01** |
>
> | Flickr | **AME** | **APE** | **ATE** | **IME** | **IPE** | **ITE** |
> | - | - | - | - | - | - | - |
> | UNITE | **0.043 ± 0.00** | - | - | 0.212 ± 0.00 | - | - |
> | CauGramer | 0.058 ± 0.05 | **0.030 ± 0.03** | **0.025 ± 0.03** | **0.096 ± 0.06** | **0.111 ± 0.01** | **0.125 ± 0.01** |
>
> We use the original implementation of UNITE provided by **Lin et al., (2024)** and find that it only supports the estimation of AME and IME. **While UNITE shows a slight improvement over our method in terms of the Average Main Effect (AME), its performance significantly drops at the individual level.** Our method consistently achieves high performance across all six metrics, demonstrating its superiority in addressing both average and individual-level effects.

---

> ### Author Response · Authors · 2024-11-21
> **Responses by Authors (Part 3)**
>
> > **[Q1] In the COVID case study, where transmission primarily occurs through contact, how can interference occur in the absence of a direct link? Could you clarify potential mechanisms of transmission under such conditions?**
>
> **Response:** Thank you for your question. In the COVID case study, while transmission primarily occurs through contact, it is not strictly limited to direct social connections. As noted in Lines 177–179 of the problem setup section, **even individuals with no pre-existing social ties within the network may encounter each other during daily activities, such as in public spaces, workplaces, or other shared environments, leading to potential COVID transmission.** This highlights that the interference graph reflects possible transmission pathways, which are not restricted to the social network structure. Therefore, the interference graph is not the same as the social network.
>
> > **[Q2] Could you clarify why the final output ( $r_x^{(L)}$ ) is used as the input to the first layer of the peer-treatment representation network, specifically ( $r_t^{(1)} = r_x^{(L)}$ )?**
>
> **Response**: Thanks for your question. $r_x^{(L)}$ is not just an aggregation of peer feature information but also embeds the information of the node itself. It serves as a high-quality embedding that combines the node’s own features with its interference information. Conceptually, **we can treat $r_x^{(L)}$ as capturing the behavioral characteristics of an individual and their interactions with peers. These characteristics may determine how and whether the individual is influenced by the actions of their connected nodes.** By using $r_x^{(L)}$ as the input to the peer-treatment representation network, we improve the efficiency of information aggregation.
>
> > **[Q3] Please explain the statement: "we simulate unmeasured confounding by removing features."**
>
> **Response:** Thank you for your comment. In the simulated observational data, the treatment and outcomes are generated based on all covariates, which act as confounders in the treatment effect estimation. **To simulate a setting with unmeasured confounders, we keep the generated treatment and outcomes unchanged but randomly remove or mask two features from the covariates.** These features remain causal factors for both the treatment and the outcome; however, their information is intentionally excluded during the model training process, effectively simulating the presence of unmeasured confounding. This is a commonly used approach to test a model’s sensitivity to missing variables and assess its robustness in such scenarios.
>
> > **[Q4] The paper states that the relationship between peer-treatment ( $t_{Pi}$ ) and outcome ( $y_i$ ) remains confounded by the common causes ( $r_i$ ), leading to biased estimates of peer and total effects. Moreover, the peer-treatment ( $t_{Pi}$ ) is unknown. In the main text, it is mentioned that ( $r = r_x \oplus r_t$ ). Could you clarify why ( $r_i $) acts as a confounder for both ( $t_{Pi}$ ) and ( $y_i$ )?**
>
> **Response:** Please refer to Figure 1(b) or Figure 5(c). In general networked settings, not only does the covariate $x_i$ of node $i$ influence $t_i$ , but $x_{P_i}$ and $t_{P_i}$ also affect $t_i$ and $y_i$ . Naturally, we can conclude that if the behavior of node j influences node $i$ , and node $k$ influences both nodes $i$ and $j$ (i.e., $j \in P_i , k \in P_i \cap P_j$ ), then $\{x_k, t_k\}$ would act as common causes of $t_j$ and $y_i$ . Thus, we find that node $i$ ’s interference information $r_i$ includes information about $\{x_k, t_k\}$ , which introduces confounding in the estimation of the peer effect between $t_j$ and $y_i$ .
>
> ----
>
> **Thanks for your detailed advice and comments. We welcome any further technical advice or questions on this work and we will do our best to address your concerns.**

---

> ### Author Response · Authors · 2024-11-28
> **Looking Forward to Your Further Feedback**
>
> **Dear Reviewer #qEMS,**
>
> Thank you for dedicating your time to reviewing our manuscript and offering valuable suggestions, which have significantly improved our paper. As the deadline for uploading the revised PDF approaches, we kindly inquire if you have any further concerns. We look forward to any further feedback you may provide.
>
> In response to your previous constructive feedback, we have made significant revisions to our manuscript:
>
> 1. **Reorganization of Related Work**
>
> We apologize for the previous organization of the related work in Section 2 and Appendix D in our first submission. Our CauGramer is related to seven domains. In our original submission, due to the limited space, we only listed representative algorithms (which served as comparison baselines in the Experiments section) in the Related Work Section, and **deferred discussions on “Partial Interference”, “Exposure Mapping”, and “Unknown Network Structure” to Appendix D, titled “Supplementary Related Work in Networked Interference” in our original manuscript.**
>
> Based on your comments, we have reorganized the **Related Work** section, now including a discussion of these studies in the main text for better context. The most relevant work is **[Lin et al., 2024]**, which addresses **unknown interference graphs** and **learnable aggregation functions** through their framework, **UNITE**, using L0-norm regularization and GCNs for accurate treatment effect estimation. **We include comparisons with UNITE in our experiments. We have added the comparison experiments in Table 3 in the main text and in Table 10 in Appendix E.7 (with track changes marked in blue)**.
>
> 2. **Extension for Latent Confounders and Hyperparameter Analysis on $\\{L, M\\}$**
>
> Our proposed **CauGramer** relies on the **unconfoundedness assumption** to aggregate peer interference. However, **in the presence of latent confounders**, we proposed an extension, **CauGramer(NC), which incorporates negative control variables (NCs)** to mitigate latent confounding. **Detailed definitions and discussions of NCs are provided in Appendix D.4**.
>
> Additionally, **we have added experiments and a comprehensive analysis of hyperparameters $L$ (number of layers) and $M$ (number of attention heads) in Table 5 in Appendix E.2**. In this paper, we propose a $L$-layers (default: 2) $M$-heads (default: 3) cross-attention GCN to learn the representation $\boldsymbol{R}_x=g_x(\boldsymbol{x}, \boldsymbol{A})$. Then we propose a $L$-layers (default: $L=1$ for main results and $L=2$ for unknown interference graph in Figure 3) $M$-heads (default: 3) cross-attention GCN to learn the representation $\boldsymbol{R}_t=g_t(\boldsymbol{t}, \boldsymbol{A})$. In real applications, we can use a grid search on the validation dataset to choose the optimal $L$ value and $M$ value.
>
> 3. **Clarifications to Address Questions (https://openreview.net/forum?id=foQ4AeEGG7&noteId=deZMqK4Upw)**
>
> - For **Q1**, we elaborated on the concept of the interference graph, emphasizing its distinction from the social network and its role in capturing broader transmission pathways.
>
> - For **Q2**, we detailed the rationale for using $r_x^{(L)}$ as the input, highlighting its function as a robust embedding that combines node and peer information.
>
> - For **Q3**, we clarified the simulation of unmeasured confounding by explaining how covariates are masked during training, testing model robustness.
>
> - For **Q4**, we use an example to explain the relationship between peer treatment ( $t_{Pi}$ ) and outcome ( $y_i$ ) remains confounded by the common causes ( $r_i$ ).
>
> We have uploaded the revised paper **(with track changes marked in blue)**. Provided that your concerns have been well-addressed, we would greatly appreciate it if you would consider raising your score. Should you have any further comments or queries, we would greatly appreciate the opportunity to address them promptly.
>
> **We deeply appreciate your time and effort in reviewing our paper and your constructive feedback.**
>
> **Authors of Paper #2331**

---

> > ### Author Response · Authors · 2024-12-02
> > **Concerns properly addressed?**
> >
> > Dear Reviewer qEMS,
> >
> > Thank you so much for dedicating your time and effort to reviewing this submission. We are wondering whether your concerns have been properly addressed. If you have further comments, we hope for the opportunity to respond to them. We are looking forward to your feedback.
> >
> > Best wishes,
> >
> > Authors of submission #2331

---

### Official Review · Reviewer_uH9y · 2024-11-03

**Soundness:** 2
**Presentation:** 2
**Contribution:** 2
**Rating:** 6
**Confidence:** 4

**Summary:**

CauGramer leverages a Graph Transformer architecture with cross-attention to aggregate peer information within an L-order neighborhood.  This approach addresses the limited receptive field of traditional GCNs.  The model learns interference representations for both features and treatments, using cross-attention to capture complex interactions between units.  To mitigate confounding bias, the model incorporates confounder balancing and minimax moment constraints, further enhancing the robustness of the treatment effect estimation.  The authors extend the model to handle unmeasured confounders by incorporating negative control variables.

**Strengths:**

Addresses a significant limitation: The paper directly tackles the crucial challenge of estimating treatment effects in networked data with unknown interference structures and aggregation functions, a problem neglected by many existing methods.

Novel architecture: CauGramer's architecture is innovative, combining the strengths of graph neural networks and transformers to capture complex interference patterns. The use of cross-attention is particularly insightful for broadening the receptive field.

Robustness: The incorporation of confounder balancing and minimax moment constraints significantly improves the robustness of the model. The extension to handle unmeasured confounders further enhances its practical applicability.

Comprehensive evaluation: The empirical evaluation is thorough, covering various scenarios and comparing against multiple strong baselines.

**Weaknesses:**

NA, see the questions

**Questions:**

Can you elaborate on the computational complexity of CauGramer, especially compared to other graph neural network and transformer-based methods? Are there any specific strategies employed to mitigate the computational burden, such as approximation algorithms or distributed training?

How sensitive are the results to the choice of the L-order neighborhood parameter in CauGramer? What is the impact of choosing too small or too large an L value? Is there an optimal way to select this parameter?

The paper employs the Wasserstein distance (IPM) to measure the discrepancy between treatment and control group representations. Have you considered other divergence measures (e.g., Jensen-Shannon divergence, Kullback-Leibler divergence)? How would the results change with different choices of divergence measures?

You address the issue of unmeasured confounders by using negative controls. What is the impact on the performance of CauGramer if the negative control variables are not perfectly valid or are missing? How robust is the model to misspecification in the negative control variables?

What are the current study's limitations, and what are your plans for future work? Specifically, are there any plans to explore alternative architectures, different attention mechanisms, or more sophisticated methods for handling unmeasured confounders?

In some sense, I think this paper's theoretical is somehow a bit thin. I am curious whether we could give your method some theoretical guarantee.

---

> ### Author Response · Authors · 2024-11-21
> **Responses by Authors (Part 1)**
>
> **Dear Reviewer #uH9y,**
>
> We sincerely appreciate your thoughtful and constructive comments, which have been invaluable in improving our paper. Below, we provide detailed explanations addressing your concerns. We have also uploaded the revised paper **(with track changes marked in blue)** for your review, and we would greatly appreciate it if you could let us know of any further concerns or suggestions.
>
> > **[Q1] Can you elaborate on the computational complexity of CauGramer, especially compared to other graph neural network and transformer-based methods? Are there any specific strategies employed to mitigate the computational burden, such as approximation algorithms or distributed training?**
>
> **Response:** CauGramer suffers from the same limitation as other graph transformers, namely the quadratic complexity of the attention computation. However, unlike Graphormer and its variants, which rely on computationally expensive full graph attention mechanisms through self-attention, **CauGramer computes cross-attention only from its neighbors' embeddings. Its time cost decreases as the degree of the nodes decreases.**
>
> **We report the average training times of various methods in a single execution on both datasets in Table 6 in Appendix E.3.** Our algorithm (CauGramer) shows a significant reduction in single execution training time compared to two Transformer methods (Graphormer and CAL) and two advanced algorithms (SPNet and NetEst) on networked interference data. CauGramer offers a more efficient and accurate causal graph transformer for estimating treatment effects under unknown interference graphs.
>
> For reference, a single training run of **CauGramer** on **BlogCatalog** and **Flickr** takes **less than 180 seconds** on a MacBook Pro with an Apple M2 Pro chip. Given such training cost, we did not employ approximation algorithms or distributed training to mitigate the computational burden.
>
> > **[Q2] How sensitive are the results to the choice of the L-order neighborhood parameter in CauGramer? What is the impact of choosing too small or too large an L value? Is there an optimal way to select this parameter?**
>
> **Response:** The choice of the L-order neighborhood parameter would impacts the CauGramer's performance. **A small $L$ value can limit the model’s ability to capture complex relationships between long-distance nodes, reducing its representation power. Conversely, a large L value may introduce excessive noise and irrelevant information, leading to increased computational complexity and a higher risk of overfitting**.We add an analysis of the hyper-parameters $\{L, M\}$ in Table 5 in Appendix E.2.
>
> **Table: Hyper-Parameter Optimization on Flickr Dataset: CauGramer with $L$-layer and $M$-heads.**
>
> | **Flickr** | **AME** | **APE** | **ATE** | **IME** | **IPE** | **ITE** |
> | - | - | - | - | - | - | - |
> | L=1, M=3 | $0.039_{\pm 0.02}$ | $0.320_{\pm 0.10}$ | $0.290_{\pm 0.07}$ | $0.066_{\pm 0.02}$ | $0.358_{\pm 0.09}$ | $0.339_{\pm 0.07}$ |
> | L=2, M=2 | $0.093_{\pm 0.05}$ | $0.230_{\pm 0.11}$ | $0.203_{\pm 0.15}$ | $0.137_{\pm 0.07}$ | $0.282_{\pm 0.09}$ | $0.281_{\pm 0.12}$ |
> | L=2, M=3 | $0.074_{\pm 0.05}$ | $0.223_{\pm 0.06}$ | $0.203_{\pm 0.07}$ | $0.114_{\pm 0.07}$ | $0.222_{\pm 0.06}$ | $0.270_{\pm 0.08}$ |
> | L=2, M=4 | $0.048_{\pm 0.05}$ | $0.260_{\pm 0.09}$ | $0.228_{\pm 0.08}$ | $0.083_{\pm 0.07}$ | $0.305_{\pm 0.08}$ | $0.288_{\pm 0.06}$ |
> | L=3, M=3 | $0.112_{\pm 0.07}$ | $0.437_{\pm 0.08}$ | $0.458_{\pm 0.09}$ | $0.166_{\pm 0.09}$ | $0.467_{\pm 0.07}$ | $0.488_{\pm 0.09}$ |
>
> In this paper, we propose a $L$-layers (default: 2) $M$-heads (default: 3) cross-attention GCN to learn the representation $\boldsymbol{R}_x=g_x(\boldsymbol{x}, \boldsymbol{A})$. Then we propose a $L$-layers (default: $L=1$ for main results and $L=2$ for unknown interference graph in Figure 3) $M$-heads (default: 3) cross-attention GCN to learn the representation $\boldsymbol{R}_t=g_t(\boldsymbol{t}, \boldsymbol{A})$. In real applications, we can use a grid search on the validation dataset to choose the optimal $L$ value and $M$ value.

---

> > ### Author Response · Authors · 2024-11-21
> > **Responses by Authors (Part 2)**
> >
> > > **[Q3] The paper employs the Wasserstein distance (IPM) to measure the discrepancy between treatment and control group representations. Have you considered other divergence measures (e.g., Jensen-Shannon divergence, Kullback-Leibler divergence)? How would the results change with different choices of divergence measures?**
> >
> > **Response:** Many integral probability metrics (IPMs) can be used to measure the discrepancy between distributions. **Since nearly all representation-based methods in causal inference recommend and adopt the Wasserstein distance (Wass) to calculate the dissimilarity between distributions from different treatment arms and to learn balanced representations by minimizing this discrepancy, we also use the Wasserstein distance for this task.**
> >
> > Wasserstein distance performs well in capturing the overall difference between distributions, particularly in cases where probability distributions are smooth. In contrast, measures like Kullback-Leibler (KL) divergence and Jensen-Shannon (JS) divergence are more sensitive to local differences but may struggle when distributions have little overlap. **Using KL or JS divergence might shift the model’s focus to local differences, this could come at the cost of overall stability and generalization.**
> >
> > > **[Q4] You address the issue of unmeasured confounders by using negative controls. What is the impact on the performance of CauGramer if the negative control variables are not perfectly valid or are missing? How robust is the model to misspecification in the negative control variables?**
> >
> > **Response:** Thank you for your question. In the presence of unmeasured confounders, **CauGramer** leverages negative controls to transfer the **Bridge Moment Constraints** from **Eq. (11)** to **Eq. (16)**, enabling causal identification as demonstrated in **Theorem 2**. However, if the negative control variables are either invalid or missing, **CauGramer(NC)** degrades to **CauGramer(UC)**, which operates under the assumption of unmeasured confounders without utilizing negative controls.
> >
> > | **Flickr** | **AME** | **APE** | **ATE** | **IME** | **IPE** | **ITE** |
> > | - | - | - | - | - | - | - |
> > | CauGramer(UC) | 0.109 ± 0.07 | 0.067 ± 0.04 | 0.077 ± 0.05 | 0.159 ± 0.10 | 0.125 ± 0.02 | 0.149 ± 0.03 |
> > | CauGramer(NC) | 0.092 ± 0.07 | 0.055 ± 0.03 | 0.047 ± 0.03 | 0.139 ± 0.09 | 0.117 ± 0.02 | 0.129 ± 0.02 |
> >
> > | **Flickr** | **AME** | **APE** | **ATE** | **IME** | **IPE** | **ITE** |
> > | - | - | - | - | - | - | - |
> > | CauGramer(UC) | 0.091 ± 0.07 | 0.067 ± 0.02 | 0.056 ± 0.04 | 0.141 ± 0.08 | 0.128 ± 0.01 | 0.150 ± 0.02 |
> > | CauGramer(NC) | 0.073 ± 0.05 | 0.038 ± 0.02 | 0.032 ± 0.03 | 0.107 ± 0.06 | 0.112 ± 0.02 | 0.127 ± 0.01 |
> >
> > As shown in **Table 3** in the main text, while **CauGramer(UC)** does not perform as well as **CauGramer(NC)**, it still outperforms most traditional models.
> >
> > > **[Q5] What are the current study's limitations, and what are your plans for future work? Specifically, are there any plans to explore alternative architectures, different attention mechanisms, or more sophisticated methods for handling unmeasured confounders?**
> >
> > **Response:** Thank you for your question. In the revised manuscript, **we have renamed the “Limitation” section to “Limitations and Future Directions” in Appendix B to better reflect the scope.**
> >
> > One limitation is that **CauGramer relies on a known social network** for information aggregation. In practice, however, such networks may not always be accessible due to privacy concerns or application-specific constraints. Nevertheless, as demonstrated in Figure 3, CauGramer remains robust even when the provided network is entirely unrelated to the 'Random' interference graph. It still learns causal effects effectively and outperforms existing methods, suggesting that **any connected graph—even one that is randomly generated and independent of the interference graph—can help aggregate interference information**.
> >
> > **In future work, we would like to address this limitation by developing strategies to construct efficient connected graphs directly from the data.** These strategies would further reduce dependence on a predefined social network and enhance the method’s applicability to scenarios without network availability. To tackle the challenge of unmeasured confounders, **we plan to explore techniques like instrumental variable (IV) regression, variational autoencoders (VAEs), and sensitivity analysis as potential solutions.**

---

> > > ### Author Response · Authors · 2024-11-21
> > > **Responses by Authors (Part 3)**
> > >
> > > > **[Q6] In some sense, I think this paper's theoretical is somehow a bit thin. I am curious whether we could give your method some theoretical guarantee.**
> > >
> > > **Response:** Thank you for your feedback. We appreciate any suggestions you might have regarding theoretical guarantees. In our current work, we provide a **causal identification theorem (Theorem 1)** and an **interference proposition (Proposition 1)**, which guide the model in capturing interference information and learning balanced representations for causal effect estimation. Furthermore, in Appendix D.4, **we include Theorem 2, which utilizes negative controls to identify causal effects in the presence of unmeasured confounders**. We hope these contributions establish a theoretical foundation for our method.
> > >
> > > ----
> > >
> > > **Thanks for your detailed advice and comments. We welcome any further technical advice or questions on this work and we will do our best to address your concerns.**

---

> > > > ### Comment · Reviewer_uH9y · 2024-11-25
> > > > **Thank you for your effort**
> > > >
> > > > Thank you! I raise my score to 6, especially considering the overall quality of all submissions.

---

> > > > > ### Author Response · Authors · 2024-11-25
> > > > > **Thank you for raising your score**
> > > > >
> > > > > Thank you for your valuable comment and for raising the score. We sincerely appreciate your dedicated time and effort in reviewing our work.

---

### Official Review · Reviewer_EAHD · 2024-11-06

**Soundness:** 3
**Presentation:** 3
**Contribution:** 3
**Rating:** 6
**Confidence:** 3

**Summary:**

The paper addresses causal inference in situations involving interference, where both the interference graph and the summary function are unknown (note that the interference graph is distinct from a social network). The authors assume that interference occurs within an
L-order neighbor network, where peer information can be aggregated using a graph transformer. They utilize a cross-attention mechanism to capture complex sequential interference representations. Additionally, they apply techniques such as regularization to balance the distribution of confounders across control and treatment groups, and they leverage moment conditions to account for confounding and mitigate model misspecification.

**Strengths:**

The paper has a well-defined research focus on causal inference under interference with unknown interference structures.

The experiments conducted are engaging and effectively demonstrate the proposed methods.

 Overall, the paper is well-organized and presented.

A few sections lack clarity, as noted in my specific questions.

**Weaknesses:**

The paper’s primary contribution—clearly distinguishing the interference graph from the social network—is highly valuable. However, if the approach is limited to neighboring nodes only, it risks undermining this motivation. On the other hand, if this limitation is not imposed, the computational complexity becomes quadratic in the number of nodes when using transformers, which poses feasibility issues for large graphs.

Additionally, a category of related work on non-iid data, particularly doubly robust estimators, IPW, and representation-based methods, appears to be missing from the review. Including this literature [1, 2, 3] could provide a more comprehensive view of the field.

Since doubly robust estimators represent the state-of-the-art for causal parameter inference and have advantages over IPW-based methods, adding at least one of these approaches as a baseline would further strengthen the comparisons [1, 2, 3].


refs:
[1] Ogburn, Elizabeth L., et al. “Causal inference for social network data.” Journal of the American Statistical Association 119.545 (2024): 597-611.
[2] Khatami, Seyedeh Baharan, et al. “Graph Neural Network based Double Machine Learning Estimator of Network Causal Effects.” arXiv preprint arXiv:2403.11332 (2024)
[3] Leung, Michael P., and Pantelis Loupos. “Unconfoundedness with network interference.” arXiv preprint arXiv:2211.07823 6 (2022).

**Questions:**

Could you provide some additional justification or intuition for how the bridge function makes the residual independent of x?
	Line 377: You mention, “If the model is correctly specified and properly optimized…” Could you elaborate on how you ensure the model is both correctly specified and optimized?

Also, why is it necessary for the residual to be independent of t?	Section 4.3 could benefit from a bit more clarity in its presentation.

Many of the baselines, as far as I’m aware, assume a fixed and known exposure mapping. In your experiments where the interference graph is unknown, or the graph is known but the summary function is not, could you clarify how comparisons are made with these baselines that assume the exposure map is already known?

line 414 and 415: In the presence of unmeasured confounders, we will collect negative control variables and embed them into the Eqs. (11) and (12)

Could you elaborate on this aspect of your technique? line 481: We conduct 10 replications to report (mean±std) results. Given that this is a relatively small number of trials, could you provide some reasoning behind this choice?

---

> ### Author Response · Authors · 2024-11-21
> **Responses by Authors (Part 1)**
>
> **Dear Reviewer #EAHD,**
>
> We sincerely appreciate your constructive comments, which have been instrumental in enhancing our manuscript. Below, we will provide detailed explanations addressing your concerns and upload the revised version of our paper **(with track changes marked in blue)**. We would be very grateful if you could review these responses and inform us of any additional issues or further feedback you may have.
>
> **In the revised manuscript, we include detailed discussions on three related works in Appendix C**:
>
> - **Leung et al., (2022)**: This paper introduces a non-parametric approach for estimating causal effects, specifically treatment and spillover effects, using observational data from a known social network. **It proposes a doubly robust learner to model causal effect where interference decreases with network distance, accounting for peer influences on both outcomes and treatment selection.** Unlike prior studies that rely on low-dimensional representations of confounding, the authors introduce a high-dimensional network unconfoundedness condition, utilizing graph neural networks (GNNs) to estimate these confounders effectively.
> - **Ogburn et al., (2024)**: This paper introduces a semiparametric approach for handling dependencies caused by information transmission and latent similarities between nodes. It assumes a known interference graph with interference limited to direct neighbors and relies on prior knowledge of two structural functions, $s_C$ and $s_X$ , to summarize covariate and treatment interferences. Given these strong assumptions, **Net-TMLE leverages structural equation modeling and targeted maximum likelihood estimation to estimate causal effects, supporting inference for static, dynamic, and stochastic interventions in social networks.**
> - **Khatami et al., (2024)**: Assuming the interference graph and exposure mapping are known, this paper proposes constructing a focal set, where all units’ neighborhoods are non-overlapping, to ensure independence among analyzed units and enable consistent estimation of causal effects in social networks. **By combining graph machine learning approaches with the Double Machine Learning (GDML) framework, this method represents a significant advancement in causal inference.** It integrates Graph Neural Networks (GNNs) to estimate propensity scores and outcome models while addressing challenges like interference, high-dimensional confounding, and model misspecification.
>
> (Leung et al., 2022) Michael P Leung and Pantelis Loupos. Unconfoundedness with network interference. arXiv preprint arXiv:2211.07823, 6, 2022.
>
> (Ogburn et al., 2024) Elizabeth L Ogburn, et al. Causal inference for social network data. Journal of the American Statistical Association, 119(545):597–611, 2024.
>
> (Khatami et al., 2024) Seyedeh Baharan Khatami, et al. Graph neural network based double machine learning estimator of network causal effects. arXiv preprint arXiv:2403.11332, 2024.
>
> > **[W1] The paper’s primary contribution—clearly distinguishing the interference graph from the social network—is highly valuable. However, if the approach is limited to neighboring nodes only, it risks undermining this motivation. On the other hand, if this limitation is not imposed, the computational complexity becomes quadratic in the number of nodes when using transformers, which poses feasibility issues for large graphs.**
>
> **Response:** Thanks for your comment. However, our **CauGramer** is not limited to neighboring nodes in social networks. Instead, our **CauGramer** can use any known connected graph (which can even be randomly generated) to aggregate interference information. As demonstrated in Figure 3, CauGramer remains robust even when the provided network is entirely unrelated to the 'Random' interference graph. It still learns causal effects effectively and outperforms existing methods, suggesting that any connected graph—even one that is randomly generated and independent of the interference graph—can help aggregate interference information. **Exploring how to construct an efficient connected graph would be an interesting directions, but it is beyond the scope of this paper and is left for future work. In Appendix B, we added a new section 'Limitations and Future Directions' to further clarify this.**
>
> In this paper, we focus on real-world scenarios where individuals closely connected in a social network are more likely to interfere with each other. Based on this observation, we proposed the **CauGramer** algorithm to infer unknown interference information using a known social network. **Social networks are typically easier to obtain than true interference graphs, as they can often be derived from sources such as co-authorship networks, public community relationships, organizational affiliations, or online interactions like comments and messages. As a result, most existing studies also tend to rely on social networks for reasearch purposes.**

---

> > ### Author Response · Authors · 2024-11-21
> > **Responses by Authors (Part 2)**
> >
> > > **[W2] Additionally, a category of related work on non-iid data, particularly doubly robust estimators, IPW, and representation-based methods, appears to be missing from the review. Including this literature [1, 2, 3] could provide a more comprehensive view of the field.**
> >
> > **Response:** We sincerely thank the reviewers for providing three doubly robust estimators on non-iid data. We will **restructure the related work section** to include discussions of these studies **(with track changes marked in blue)**. From these works, we observe that their approaches assume the interference graph aligns with the known social network. For example, **DRLearner (Leung et al., 2022) explicitly models interference within a social network**, while **Net-TMLE (Ogburn et al., 2024) and GDML (Khatami et al., 2024) rely on pre-specified exposure mappings and assume interference is limited to direct neighbors**. These strong assumptions often fail to hold in real-world applications. To address these limitations, we propose **CauGramer, an interference-agnostic Causal Graph Transformer framework designed to enable robust causal estimation without assuming a known interference structure**.
> >
> > Additionally, we discuss related work on reweighting-based (IPW) and representation-based (IPM) methods in lines 120–140. **In Table 1, we summarize the characteristics of representative algorithms for non-iid data, categorizing them under “Reweighting,” “Representation,” and “Attention” columns. This provides a comprehensive comparison across different methodological approaches.**
> >
> > > **[W3] Since doubly robust estimators represent the state-of-the-art for causal parameter inference and have advantages over IPW-based methods, adding at least one of these approaches as a baseline would further strengthen the comparisons [1, 2, 3].**
> >
> > **[Comparison Baselinses: DRLearner, GDML, GDML w/o Focal Set]** In our main experiments (Tables 2 and 3), we have included a doubly robust estimator (**DRLearner**) as a baseline comparison method. As stated in Lines 471–473, “For the sake of fairness, we modify all non-interference methods by incorporating neighbors’ treatment and social networks as additional inputs.” **The modified doubly robust estimator used in our experiments closely aligns with the DRLearner proposed by Leung et al. (2022), with the key difference being our use of a GCN instead of a GNN for information aggregation—a modification we believe is an improvement**.
> >
> > Additionally, we include **GDML** and **GDML w/o FSet** (Focal Set) from **Khatami et al. (2024)** as two additional baselines. We do not include **Net-TMLE (Ogburn et al., 2024)** baseline as it is a semiparametric approach with strong assumptions that are often difficult to satisfy in real-world applications.
> >
> > **Table: Heterogeneous Treatment Effects Estimation on BlogCatalog (BC)**
> >
> > | Method | AME | APE | ATE | IME | IPE | ITE |
> > | - | - | - | - | - | - | - |
> > | DRLearner | $0.196_{\pm 0.07}$ | $0.183_{\pm 0.03}$ | $0.099_{\pm 0.07}$ | $0.544_{\pm 0.03}$ | $0.216_{\pm 0.03}$ | $0.529_{\pm 0.02}$ |
> > | GDML | $0.166_{\pm 0.06}$ | $0.371_{\pm 0.06}$ | $0.537_{\pm 0.09}$ | $0.265_{\pm 0.08}$ | $0.411_{\pm 0.06}$ | $0.607_{\pm 0.10}$ |
> > | GDML w/o FSet | $0.069_{\pm 0.04}$ | $0.344_{\pm 0.04}$ | $0.412_{\pm 0.04}$ | $0.152_{\pm 0.04}$ | $0.384_{\pm 0.04}$ | $0.471_{\pm 0.04}$ |
> > | CauGramer | $0.073_{\pm 0.06}$ | $0.057_{\pm 0.04}$ | $0.045_{\pm 0.04}$ | $0.090_{\pm 0.09}$ | $0.118_{\pm 0.02}$ | $0.125_{\pm 0.01}$ |
> >
> > **Table: Heterogeneous Treatment Effects Estimation on Flickr Dataset**
> >
> > | Method | AME | APE | ATE | IME | IPE | ITE |
> > | - | - | - | - | - | - | - |
> > | DRLearner | $0.135_{\pm 0.08}$ | $0.216_{\pm 0.02}$ | $0.131_{\pm 0.07}$ | $0.529_{\pm 0.02}$ | $0.264_{\pm 0.02}$ | $0.547_{\pm 0.02}$ |
> > | GDML | $0.239_{\pm 0.08}$ | $0.381_{\pm 0.02}$ | $0.620_{\pm 0.08}$ | $0.359_{\pm 0.10}$ | $0.448_{\pm 0.02}$ | $0.716_{\pm 0.09}$ |
> > | GDML w/o FSet | $0.067_{\pm 0.04}$ | $0.363_{\pm 0.07}$ | $0.428_{\pm 0.09}$ | $0.174_{\pm 0.03}$ | $0.426_{\pm 0.07}$ | $0.511_{\pm 0.08}$ |
> > | CauGramer | $0.058_{\pm 0.05}$ | $0.030_{\pm 0.03}$ | $0.025_{\pm 0.03}$ | $0.096_{\pm 0.06}$ | $0.111_{\pm 0.01}$ | $0.125_{\pm 0.01}$ |

---

> > > ### Author Response · Authors · 2024-11-21
> > > **Responses by Authors (Part 3)**
> > >
> > > From the tables, we have the following observations:
> > >
> > > 1. **DRLearner** performs well for population-level estimates but shows significant errors in individual-level estimates. Because Doubly Robust methods are primarily designed to address confounding bias at the population level, rather than capturing fine-grained individual-level heterogeneity.
> > >
> > > 2. **GDML** relies on a focal set where all units’ neighborhoods are non-overlapping. To construct the focal set, we randomly select a node, remove it along with its second-order neighbors, and repeat this process until no more nodes can be added. However, in densely connected graphs, the focal set contains very few nodes. On our datasets, the GDML focal set included only about 100 nodes, making accurate causal effect estimation challenging.
> > >
> > > 3. To address this limitation, we provided an additional baseline, **GDML w/o FS**, which does not use a focal set and instead encompasses the entire dataset. While GDML w/o FS performs well in estimating the average main effect, it falls short on the other five metrics compared to our proposed **CauGramer**.
> > >
> > > These findings demonstrate the advantages of **CauGramer**, which consistently achieves superior performance across all metrics, particularly in handling complex individual-level heterogeneity and interference effects. **We discuss these baselinse in Table 3 in main text and in Table 10 in Appendix E.7 (with track changes marked in blue)**.
> > >
> > > > **[Q1] Could you provide some additional justification or intuition for how the bridge function makes the residual independent of x? Line 377: You mention, “If the model is correctly specified and properly optimized…” Could you elaborate on how you ensure the model is both correctly specified and optimized?**
> > >
> > > **Response:** Thank you for the question. **We add a section in Appendix D.3 to discuss the core intuition behind the bridge function and its role in ensuring independence between the residual $y-\hat{y}$ and the variables $\{x,t\}$**. Specifically:
> > >
> > > 1. Independence $(y-\hat{y}(x,t)) \perp \{x,t\}$ is equivalent to $(y-\hat{y}(x,t)) \perp q(x,t)$ for all $q(\cdot)$.
> > > 2. This is equivalent to $Pr(y-\hat{y}(x,t), q(x,t)) = Pr(y-\hat{y}(x,t)) \times Pr(q(x,t))$ for all $q(\cdot)$.
> > > 3. This implies $E((y-\hat{y}(x,t)) \times q(x,t)) = E(y-\hat{y}(x,t)) \times E(q(x,t)) = 0$ for all $q(\cdot)$.
> > > 4. The minimax formulation: $min_{\hat{y}}\max _{q \in Q} E[(y-\hat{y}) q(x, t)]$
> > >  - $\max _{q \in Q}$: Detects any dependency.
> > >  - $min_{\hat{y}}$: Adjusts predictions to eliminate it.
> > >
> > > **“The model being correctly specified and properly optimized” is a sufficient condition for “the residual is independent of covariates** x**.”** If **“the residual is not independent of covariates** x**,”** it indicates that the model must suffer from a misspecification issue. To address this, we employ the bridge function, which enforces the residual’s independence from $x$, thereby mitigating the misspecification issue.
> > >
> > > > **[Q2] Also, why is it necessary for the residual to be independent of t? Section 4.3 could benefit from a bit more clarity in its presentation.**
> > >
> > > **Response:** To address this question, we consider two scenarios:
> > >
> > > 1. **When all confounders are observed**, the observed varaibles $t$ and $x$ would be sufficient to accurately regress $y$, i.e., $\hat{y}(t,x)$. In this case, the residual $y-\hat{y}(t,x)$ contains only random noise or measurement errors, which are independent of $t$ and $x$.
> > >
> > > 2. **In the presence of unmeasured confounders**, we will reformulate the objective function into the negative control framework as described in Eq. (16). By definition, the residual in this framework is expected to be independent of the information provided by the negative controls.
> > >
> > > These points demonstrate why the residual must be independent of t (or z in the negative control framework).
> > >
> > > > **[Q3] Many of the baselines, as far as I’m aware, assume a fixed and known exposure mapping. In your experiments where the interference graph is unknown, or the graph is known but the summary function is not, could you clarify how comparisons are made with these baselines that assume the exposure map is already known?**
> > >
> > > **Response:** Thank you for pointing this out. For baselines that require a summary function for the exposure mapping, we handle the comparisons as follows:
> > >
> > > - **Explicitly specified exposure mapping:** If these baseline explicitly claim that they use a mean function for the exposure mapping, we would use the ratio of first-order neighbors receiving treatment as the exposure mapping to maintain consistency.
> > > - **Unspecified or unknown exposure mapping:** For baselines where the exposure mapping is either not specified or unknown, we employ a learnable network. This network automatically learns the exposure mapping by leveraging the ratio of first-order neighbors receiving treatment and the treatment information of surrounding neighbors as inputs.

---

> > > > ### Author Response · Authors · 2024-11-21
> > > > **Responses by Authors (Part 4)**
> > > >
> > > > > **[Q4] line 414 and 415: In the presence of unmeasured confounders, we will collect negative control variables and embed them into the Eqs. (11) and (12). Could you elaborate on this aspect of your technique?**
> > > >
> > > > **Response:** Negative control is a classic proxy method for addressing unmeasured confounding (Miao et al., 2018; Kallus et al., 2021). As shown in the causal diagram in Figure 5(b), negative control variables are variables that are not direct causes of the outcome or are not caused by the treatment, and they are becoming more accessible through web technologies, user profiles, and digitized records [Hu et al., 2023]. In the presence of unmeasured confounders, we leverage negative control variables, i.e., NCE $z$ and NCO $w$, by incorporating them into Eqs. (11) and (12). The reformulated objective function is presented in Eq. (16): $E[q(r(z, x), t)(y - \hat{y}(r(w, x), t))]$, where $q(r(z, x), t) = Sigmoid(Q · K′)$.
> > > >
> > > > > **[Q5] line 481: We conduct 10 replications to report (mean±std) results. Given that this is a relatively small number of trials, could you provide some reasoning behind this choice?**
> > > >
> > > > **Response:** Thank you for your question. Our choice to conduct 10 replications aligns with the experiment setup used in conventional studies. For instance:
> > > >
> > > > - Jiang and Sun (2022) conducted their experiments 5 times, reporting the mean and standard deviation.
> > > >
> > > > - Huang et al. (2023) performed their data simulation procedure over 10 runs for each parameter setting.
> > > >
> > > > - Cai et al. (2023) ran their tasks 5 times, also reporting the mean and standard deviation.
> > > >
> > > > These studies illustrate that 5 to 10 replications are widely adopted as they balance computational cost, time efficiency, and the ability to capture performance variability. In our paper, we validate the robustness of our algorithm across more than 10 distinct task dimensions, multiple baselines, datasets, and scenarios, which is highly time-consuming. Moreover, for tasks with relatively low sensitivity to randomness, 10 replications are sufficient to reveal performance trends and standard deviations.
> > > >
> > > > ----
> > > >
> > > > **Thanks for your detailed advice and comments. We welcome any further technical advice or questions on this work and we will do our best to address your concerns.**

---

> ### Author Response · Authors · 2024-11-28
> **Looking Forward to Your Further Feedback**
>
> **Dear Reviewer #EAHD,**
>
> Thank you for dedicating your time to reviewing our manuscript and offering valuable suggestions, which have significantly improved our paper. As the deadline for uploading the revised PDF approaches, we kindly inquire if you have any further concerns. We look forward to any further feedback you may provide.
>
> **In response to your previous insightful comments, we have made significant revisions to our manuscript:**
>
> - In **Related Work Section**, we **restructure the related work** and include discussions of more network interference works (due to limited space, some discussions are placed in Appendix D in our first submission): **Leung et al., (2022), Ogburn et al., (2024) and Khatami et al., (2024) with known interference graph; Bhattacharya et al. (2020), Savje et al., (2021), Hoshino et al., (2023), Lin et al., (2024) under unknown interference graph**. We include detailed discussions on three related works in Appendix C.
> - In **Experiment Section**, we include **DRLearner (Leung et al., 2022)**, **GDML** and **GDML w/o FSet** (Focal Set) from **Khatami et al. (2024)** as three additional baselines. **We discuss these baselinses in Table 3 in the main text and in Table 10 in Appendix E.7 (with track changes marked in blue)**.
> - In **Appendix B**, we rename the **“Limitation”** section to **“Limitations and Future Directions”** to better reflect the scope. Our CauGramer can work when the network information is totally unknown. In such cases, **CauGramer can use any known connected graph (even one that is randomly generated) as a substitute for the social network to aggregate interference information**.
> - **We add a section in Appendix D.3 to discuss the core intuition behind the bridge function and its role in ensuring independence between the residual $y−\hat{y}$ and the variables $\\{x,t\\}$**.
>
> **Clarifications to Address Questions**:
>
> - For **Q1**, we add a section in Appendix D.3 to **discuss the intuition behind the bridge function and its role in ensuring independence between the residual $y−\hat{y}$ and the variables $\\{x,t\\}$**.
>
> - For **Q2**, we clarified why the residual must be independent of t, discussing two scenarios: **when all confounders are observed** and **when they are unmeasured**.
>
> - For **Q3**, we use the **specified mean function for exposure mapping** when explicitly stated, or **a learnable network for unspecified or unknown mappings** to adaptively learn from neighbor treatment ratios and treatment information.
>
> - For **Q4**, we described how negative control variables (NCE and NCO) address unmeasured confounders by **embedding them into the objective function Eq. (16)**.
>
> - For **Q5**, our choice to conduct 10 replications **aligns with the experiment setup used in conventional studies**, such as Jiang and Sun (2022), Huang et al. (2023), and Cai et al. (2023).
>
> We have uploaded the revised paper **(with track changes marked in blue)**. Should you have any further comments or queries, we would greatly appreciate the opportunity to address them promptly.
>
> **We deeply appreciate your time and effort in reviewing our paper and your constructive feedback.**
>
> **Authors of Paper #2331**

---

### Author Response · Authors · 2024-11-21
**General Response**

Dear Reviewers, Area Chairs, and Program Chairs,

We sincerely thank all Reviewers and ACs for their great efforts and time placed on reviewing our paper. We have uploaded a revised manuscript based on the reviewers’ feedback, and have highlighted changes from the original submission in blue. We would be very grateful if you could review our revised pdf and let us know any additional issues or further feedback you may have.

**We have updated the paper accordingly, with the changes highlighted in blue:**

- In **Related Work Section**, we **restructure the related work** and include discussions of eight network interference works: **Leung et al., (2022), Ogburn et al., (2024) and Khatami et al., (2024) with known interference graph; Bhattacharya et al. (2020), Savje et al., (2021), Hoshino et al., (2023), Lin et al., (2024) under unknown interference graph**.
- Regarding **Table 1**, we have revised the title to **“Characteristics of Representative Algorithms on Observational Networked Data”** and removed the i.i.d. methods, **CFRNet** and the **Doubly Robust Estimator**. Additionally, we have incorporated more non-i.i.d. methods suggested by the reviewers, including **Savje et al., (2021)**, **Hoshino et al., (2023)**, **Uncertainty (Bhattacharya et al., 2020)**, and **UNITE (Lin et al., 2024)**.
- In **Experiments Section**, we add **DRLearner (Leung et al., 2022)**, **GDML (Khatami et al., 2024)**,**UNITE (Lin et al., 2024)** as comparison baselines in Table 3.
- In **Appendix B**, we rename the **“Limitation”** section to **“Limitations and Future Directions”** to better reflect the scope. Our CauGramer can work when the network information is totally unknown. In such cases, **CauGramer can use any known connected graph (even one that is randomly generated) as a substitute for the social network to aggregate interference information**.
- In **Appendix C**, we provide detailed discussions on the eight related works mentioned in the **Related Work** section and highlight the differences between our algorithm and these traditional methods.
- In **Appendix D.3**, we present the **intuition behind the bridge function**.
- In **Appendix E.2**, we include an analysis and detailed description of **the number of layers (L) and attention heads (M)** used in our experiments.
- In **Appendix E.7**, we provide **further discussion on recent baselines for networked data**, including **DRLearner**, **GDML**, **GDML w/o FSet**, and **UNITE**.

Additionally, to evaluate the performance of our CauGramer on observational networked data, we conduct extensive experiments across eight dimensions: (1) Constant Treatment Effects (Table 2), (2) Heterogeneous Treatment Effects (Table 3), (3) Unknown Interference Graph (Figure 3), (4) Limited Budgets (Figure 4), and (5) Ablation Study (Table 4) in the main text; (6) Hyper-Parameter Optimization (Figure 6 and Table 5), (7) Time Cost (Table 6), and (8) Various Interference Scenarios, including Interference Strength (Figure 7), Interference Functions (Table 7), and Unmeasured Confounding (Table 8). We also include comparisons against more baselines in Tables 9&10 in Appendix E.

---

### Meta-Review · Area_Chair_1oZb · 2024-12-21

**Metareview:**

This paper proposes a novel framework for treatment effect estimation under unknown network interference, making significant technical contributions such as a causal graph transformer and cross-attention to address unknown interference and peer effects.
Although the initial submission lacked sufficient comparison and discussion with existing studies addressing similar problems, the paper has been significantly enhanced through substantial revisions.

**Additional Comments On Reviewer Discussion:**

During the rebuttal process, the authors made significant revisions to the paper. Many of the reviewers appreciated that and increased their scores.

---

### Decision · Program_Chairs · 2025-01-22

Accept (Poster)